# SEEDS: Exponential SDE Solvers for Fast High-Quality Sampling from Diffusion Models

**Martin Gonzalez**[*]  
IRT SystemX

**Nelson Fernandez**  
Air Liquide

**Thuy Tran**  
IRT SystemX

**Elies Gherbi**  
IRT SystemX

**Hatem Hajri**  
IRT SystemX  
& Safran

**Nader Masmoudi**  
New York University

## Abstract

A potent class of generative models known as Diffusion Probabilistic Models (DPMs) has become prominent. A forward diffusion process adds gradually noise to data, while a model learns to gradually denoise. Sampling from pre-trained DPMs is obtained by solving differential equations (DE) defined by the learnt model, a process which has shown to be prohibitively slow. Numerous efforts on speeding-up this process have consisted on crafting powerful ODE solvers. Despite being quick, such solvers do not usually reach the optimal quality achieved by available slow SDE solvers. Our goal is to propose SDE solvers that reach optimal quality without requiring several hundreds or thousands of NFEs to achieve that goal. We propose Stochastic Explicit Exponential Derivative-free Solvers (SEEDS), improving and generalizing Exponential Integrator approaches to the stochastic case on several frameworks. After carefully analyzing the formulation of exact solutions of diffusion SDEs, we craft SEEDS to analytically compute the linear part of such solutions. Inspired by the Exponential Time-Differencing method, SEEDS use a novel treatment of the stochastic components of solutions, enabling the analytical computation of their variance, and contains high-order terms allowing to reach optimal quality sampling $\sim$ 3-5$\times$ faster than previous SDE methods. We validate our approach on several image generation benchmarks, showing that SEEDS outperform or are competitive with previous SDE solvers. Contrary to the latter, SEEDS are derivative and training free, and we fully prove strong convergence guarantees for them. Our code is publicly available in this link.

## 1 Introduction

Diffusion Probabilistic Models (DPMs) [32, 12] have emerged as a powerful category of generative models and have proven to quickly become SOTA for generative tasks such as image, video, audio generation [8, 24, 13, 3, 4], and more [29, 36]. These models employ a forward diffusion process where noise is gradually added to the data, and the model learns to remove the noise progressively. However, sampling from most pre-trained DPMs is done by simulating the trajectories of associated differential equations (DE) and has been found to be prohibitively slow [34]. Previous attempts to accelerate this process have mainly focused on developing efficient ODE solvers. On one hand, training-based methods speed-up sampling by using auxiliary training such as Progressive Distillation [31] and Fourier Neural Operators [42], learning the noise schedule, scaling, variance, or trajectories. On the other hand, training-free methods [16, 22, 15, 40, 23] are slower but are more versatile for

---

[*]Corresponding author: `martin.gonzalez@irt-systemx.fr`

37th Conference on Neural Information Processing Systems (NeurIPS 2023).

being employed on different models and achieve higher quality results than current training-based methods. Although these solvers are fast, they often fall short of achieving the optimal quality attained by slower SDE solvers [16]. The latter usually do not present theoretical convergence guarantees and, while being training-free, they often still require costly parameter optimization to achieve optimal results which might be difficult to estimate for large datasets.

Our objective is to introduce SDE solvers that can achieve optimal quality without requiring an excessively large number of function evaluations (NFEs). To accomplish this, we propose Stochastic Explicit Exponential Derivative-free Solvers (SEEDS). These are *off-the-shelf* SDE samplers: they offer promising high-quality sampling without further training or parameter optimization. SEEDS enhance and generalize existing Exponential Integrator [22, 23, 41] approaches to the stochastic case on several practical frameworks and is based on the following 4 building blocks: (1) the exponential representation of semi-linear SDE exact solutions which isolate linear terms to be computed analytically; (2) a general change-of-variables recipe to simplify the integrals involved in the solutions in order to better approximate the deterministic one; (3) a method to analytically compute the variance of the stochastic one; (4) a method to decompose the obtained stochastic components in such a way that the resulting sequences of higher-stage numerical approximations yield Markov chains.

Overall, we make the following contributions: (a) our change-of-variables method allow us to re-frame the gDDIM solver [41] as a special case of SEEDS and to craft bespoke solvers for the EDM-preconditioned DPMs in [16] which attain equivalent sampling quality twice faster than the previous SOTA sampling method [16]; (b) based on the Stochastic Exponential Time-Differencing method, we analytically compute of our solver's stochastic components in terms of the so-called $\varphi$-functions, allowing for efficient implementation; (c) our noise decomposition method (4), which is the key of success of SEEDS, is both theoretically grounded, experimentally shown to be optimal, and has no deterministic equivalent; (d) we provide full proofs of strong/weak convergence guarantees for our SDE solvers which, to our knowledge, has no precedent in the DPM literature. In particular, although the formula used for the truncated Itô-Taylor expansion might seem similar to that of [22], our convergence theorems leverage the full Itô-Taylor expansion of solutions, making our proofs not incremental and different from [22]; (e) we conduct extensive experiments demonstrating that SEEDS establishes SOTA results among available solvers on several image generation benchmarks, or is competitive with existing SDE solvers while being 2-5 times faster than the latter.

Although our solvers theoretically apply to certain non-isotropic DPMs such as Critically-damped Langevin Dynamics (CLD) [10] (see Prop. 4.5 and Rem. 4.6), we will restrict our presentation to the isotropic case for which many notations become simpler.

## 2 Background on Diffusion Probabilistic Models

**General Isotropic DE Formulation.** The evolution of a data sample $\mathbf{x}_0 \in \mathbb{R}^d$ taken from an unknown data distribution $p_{\text{data}}$ into standard Gaussian noise can be defined as a forward diffusion process $\{\mathbf{x}_t\}_{t \in [0,T]}$, with $T > 0$, which is a solution to a linear SDE:

$$\mathrm{d}\mathbf{x}_t = f(t)\mathbf{x}_t \mathrm{d}t + g(t)\mathrm{d}\boldsymbol{\omega}_t, \qquad f(t) := \frac{\mathrm{d} \log \alpha_t}{\mathrm{d}t}, \quad g(t) = \alpha_t \sqrt{\frac{\mathrm{d}[\sigma_t^2]}{\mathrm{d}t}}, \tag{1}$$

where $f(t), g(t) \in \mathbb{R}^{d \times d}$ are called the drift and diffusion coefficients respectively and $\boldsymbol{\omega}$ is a $d$-dimensional standard Wiener process, and $\alpha_t, \sigma_t \in \mathbb{R}^{>0}$ are differentiable functions with bounded derivatives. In practice, when specifying the SDE (1), $\sigma_t$ acts as a schedule controlling the noise levels of an input at time $t$, and $\alpha_t$ as a time-dependent signal scaling controlling its dynamic range.

By denoting $p_t(\mathbf{x}_t)$ the marginal distribution of $\mathbf{x}_t$ at time $t$, functions $\alpha_t$ and $\sigma_t$ are designed so that the end-time distribution of the process process is $p_T(\mathbf{x}_T) \approx \mathcal{N}(\mathbf{x}_T | \mathbf{0}, \tilde{\sigma}^2 \mathbf{I}_d)$ for some $\tilde{\sigma} > 0$. As (1) is linear, the transition probability $p_{0t}(\mathbf{x}_t | \mathbf{x}_0)$ from $\mathbf{x}_0$ to $\mathbf{x}_t$ is Gaussian whose mean and variance can be expressed in terms of $\alpha_t$ and $\sigma_t$. For simplicity, we will denote it

$$p_{0t}(\mathbf{x}_t | \mathbf{x}_0) = \mathcal{N}(\mathbf{x}_t; \mu_t \mathbf{x}_0, \Sigma_t), \qquad \mu_t, \Sigma_t \in \mathbb{R}^{d \times d}.$$

The evolution of the reverse time process of $\{\mathbf{x}_t\}_{t \in [0,T]}$ (which we will still denote $\{\mathbf{x}_t\}_{t \in [0,T]}$ for simplicity) is then driven by a backward differential equation

$$\mathrm{d}\mathbf{x}_t = [f(t)\mathbf{x}_t - \frac{1+\ell^2}{2} g^2(t) \nabla_{\mathbf{x}_t} \log p_t(\mathbf{x}_t)]\mathrm{d}t + \ell g(t)\mathrm{d}\bar{\boldsymbol{\omega}}_t, \tag{2}$$

where $\mathrm{d}t$ are negative infinitesimal time-steps and $\bar{\boldsymbol{\omega}}_t$ is now a Wiener process with variance $-\mathrm{d}t$. In this article, we will concentrate in the cases $\ell = 0, 1$, known in the literature as the Probability Flow ODE (PFO) and diffusion reverse SDE (RSDE), respectively.

**Training.**   Denoising score-matching is a technique to train a time-dependent model $D_\theta(\mathbf{x}_t, t)$ to approach the score function $\nabla_{\mathbf{x}_t} \log p_t(\mathbf{x}_t)$ at each time $t$. Intuitively, as $D_\theta$ approaches the score, it produces a sample which maximizes the log-likelihood. As such, this model is coined as a *data prediction* model. However, in practice DPMs can be more efficiently trained by reparameterizing $D_\theta$ into a different model $F_\theta(\mathbf{x}_t, t)$ whose objective is to predict the noise to be removed from a sample at time $t$. This *noise prediction* model is trained by means of the loss

$$\mathbb{E}_{t \sim \mathcal{U}[0,T], \mathbf{x}_0 \sim p_{\mathrm{data}}, \epsilon \sim \mathcal{N}(\mathbf{0}, \mathbf{I}_d)}[\|\epsilon - F_\theta(\mu_t \mathbf{x}_0 + \boldsymbol{K}_t \epsilon, t)\|^2_{\boldsymbol{K}_t^{-1} \gamma_t \boldsymbol{K}_t^{-\top}}],$$

where $\gamma_t$ is a time dependent weighting parameter and $\boldsymbol{K}_t \boldsymbol{K}_t^\top = \Sigma_t$.

## 3   Accelerating Optimal Quality Solvers for Diffusion SDEs

Once $F_\theta$ or $D_\theta$ have been trained, one can effectively solve (2) after replacing the score function by its corresponding expression involving either one of these models. For instance, taking the noise prediction model and $\ell = 1$, sampling is conducted by simulating trajectories of a SDE of the form

$$\mathrm{d}\mathbf{x}_t = [A(t)\mathbf{x}_t + b(t)F_\theta(\mathbf{x}_t, t)]\mathrm{d}t + g(t)\mathrm{d}\boldsymbol{\omega}_t, \tag{3}$$

for some functions $A(t), b(t)$ which are usually not equal to $f(t), g^2(t)$. In what follows, we consider a time discretization $\{t_i\}_{i=0}^M$ going backwards in time starting from $t_0 = T$ to $t_M = 0$ and to ease the notation we will always denote $t < s$ for two consecutive time-steps $t_{i+1} < t_i$.

The usual representation of the analytic solution $\mathbf{x}_t$ at time $t$ of (3) w.r.t. an initial condition $\mathbf{x}_s$ is:

$$\mathbf{x}_t = \mathbf{x}_s + \int_s^t [A(\tau)\mathbf{x}_\tau + b(\tau)F_\theta(\mathbf{x}_\tau, \tau)]\mathrm{d}\tau + \int_s^t g(\tau)\mathrm{d}\boldsymbol{\omega}_\tau. \tag{4}$$

The numerical schemes we propose for approaching the trajectories of (3) based on representation (4) are grounded in the 4 following principles:

1. The variation-of-parameters formula: representing analytic solutions with linear term extracted from the integrand;

2. Exponentially weighted integrals: extracting the time-varying linear coefficient attached to the network from the integrand by means of a specific choice of change of variables which allows analytic computation of the leading coefficients in the truncated Itô-Taylor expansion associated to $F_\theta(\mathbf{x}_\tau, \tau)$ up to any arbitrary order;

3. Modified Gaussian increments: after replicating such change of variables onto the stochastic integral, analytically computing its variance.

4. Markov-preserving noise decomposition: stochastic integrals need to be dependent on overlapping time intervals and independent on non-overlapping ones.

**Exponential representation of exact solutions of diffusion SDEs.**   The first key insight of this work is that, using the *variation-of-parameters* formula, we can represent the analytic solution $\mathbf{x}_t$ at time $t$ of (3) with respect to an initial condition $\mathbf{x}_s$ as follows:

$$\mathbf{x}_t = \Phi_A(t, s)\mathbf{x}_s + \int_s^t \Phi_A(t, \tau)b(\tau)F_\theta(\mathbf{x}_\tau, \tau)\mathrm{d}\tau + \int_s^t \Phi_A(t, \tau)g(\tau)\mathrm{d}\boldsymbol{\omega}_\tau, \tag{5}$$

where $\Phi_A(t, s) = \exp\left(\int_s^t A(\tau)\mathrm{d}\tau\right)$ is called the transition matrix associated with $A(t)$. The separation of the linear and nonlinear components is achieved by this formulation and also appears in [22, 23, 41]. It differs from black-box SDE solvers as it enables the exact calculation of the linear portion, thereby removing any approximation errors associated with it. However, the integration of the nonlinear portion remains complex due to the interaction of the new coefficient $\Phi_A(t, \tau)b(\tau)$ and the intricate neural network, making it challenging to approximate.

**Exponentially weighted integrals.**   Due to the regularity conditions usually imposed on the drift and diffusion coefficients of (1), one can make several choices of change-of-variables on the integral components in (5) in order to simplify it. Our second key insight is that there is a specific choice of change of variables allowing the analytic computation of the Itô-Taylor coefficients of $F_\theta(\mathbf{x}_\tau, \tau)$ with respect to $\tau$, and based at $s$ that will be used for crafting SEEDS. More specifically, this expansion reads

$$F_\theta(\mathbf{x}_\tau, \tau) = \sum_{k=0}^{n} \frac{(\tau - s)^k}{k!} F_\theta^{(k)}(\mathbf{x}_s, s) + \mathcal{R}_n,$$

where the residual $\mathcal{R}_n$ consists of deterministic iterated integrals of length greater than $n + 1$ and all iterated integrals with at least one stochastic component. As such, we obtain

$$\int_s^t \Phi_A(t, \tau) b(\tau) F_\theta(\mathbf{x}_\tau, \tau) \mathrm{d}\tau = \sum_{k=0}^{n} F_\theta^{(k)}(\mathbf{x}_s, s) \int_s^t \Phi_A(t, \tau) b(\tau) \frac{(\tau - s)^k}{k!} \mathrm{d}\tau + \tilde{\mathcal{R}}_n, \quad (6)$$

where $\tilde{\mathcal{R}}_n$ is easily obtained from $\mathcal{R}_n$ and $\int_s^t \Phi_A(t, \tau) b(\tau) \mathrm{d}\tau$. The third key contribution of our work is to rewrite, for any $k \geqslant 0$, the integral $\int_s^t \Phi_A(t, \tau) b(\tau) \frac{(\tau - s)^k}{k!} \mathrm{d}\tau$ as an integral of the form $\int_{\lambda_s}^{\lambda_t} e^\lambda \frac{(\lambda - \lambda_s)^k}{k!} \mathrm{d}\lambda$ since the latter is recursively analytically computed in terms of the $\varphi$-functions

$$\varphi_0(t) := e^t, \qquad \varphi_{k+1}(t) := \int_0^1 e^{(1-\tau)t} \frac{\tau^k}{k!} \mathrm{d}\tau = \frac{\varphi_k(t) - \varphi_k(0)}{t}, \qquad k \geqslant 0.$$

**Modified Gaussian increments.**   In order for making such change of variables to be consistent on the overall system, one needs to replicate it accordingly in the stochastic integral $\int_s^t \Phi_A(t, \tau) g(\tau) \mathrm{d}\bar{\boldsymbol{\omega}}_\tau$. As such, our last key contribution is to transform it into an exponentially weighted stochastic integral with integration endpoints $\lambda_s, \lambda_t$ and apply the Stochastic Exponential Time Differencing (SETD) method [1] to compute its variance analytically, as illustrated in (14) below.

Let us test our methodology in two key examples. As we explained in Section 2, sampling from pre-trained DPMs amounts on choosing a schedule $\sigma_t$, a scaling $\alpha_t$, and a parameterized learnt approximation of the score function $\nabla_{\mathbf{x}_t} \log p_t(\mathbf{x}_t)$. In what follows, we denote by $t_\lambda$ the inverse of a chosen change of variables $\lambda_t$ and we denote $\hat{\mathbf{x}}_\lambda := \mathbf{x}(t_\lambda(\lambda)), \hat{F}_\theta(\hat{\mathbf{x}}_\lambda, \lambda) := F_\theta(\mathbf{x}(t_\lambda(\lambda)), t_\lambda(\lambda))$.

**The VPSDE case.**   Let $\tilde{\alpha}_t := \frac{1}{2}\beta_d t^2 + \beta_m t$, where $\beta_d, \beta_m > 0$ and $t \in [0, 1]$. Then, by denoting

$$\sigma_t := \sqrt{e^{\tilde{\alpha}_t} - 1}, \qquad \alpha_t := e^{-\frac{1}{2}\tilde{\alpha}_t}, \qquad \bar{\sigma}_t := \alpha_t \sigma_t, \qquad \nabla_{\mathbf{x}_t} \log p_t(\mathbf{x}_t) \simeq \bar{\sigma}_t^{-1} F_\theta(\mathbf{x}_t, t), \quad (7)$$

we obtain the VP SDE framework from [34] and the following result.

**Proposition 3.1.** *Let $t < s$. The analytic solution at time $t$ of the RSDE* (2) *with coefficients* (7) *and initial value $\mathbf{x}_s$ is*

$$\mathbf{x}_t = \frac{\alpha_t}{\alpha_s} \mathbf{x}_s - 2\alpha_t \int_{\lambda_s}^{\lambda_t} e^{-\lambda} \hat{F}_\theta(\hat{\mathbf{x}}_\lambda, \lambda) \mathrm{d}\lambda - \sqrt{2}\alpha_t \int_{\lambda_s}^{\lambda_t} e^{-\lambda} \mathrm{d}\bar{\boldsymbol{\omega}}_\lambda, \qquad \lambda_t := -\log(\sigma_t). \quad (8)$$

The change of variables of (8) is interesting as it allows to compute analytically the Itô-Taylor coefficients in (6) by using, for $h = \lambda_t - \lambda_s$, the following key result which will be used in Prop. 4.2:

$$\int_{\lambda_s}^{\lambda_t} e^{-\lambda} \frac{(\lambda - \lambda_s)^k}{k!} \mathrm{d}\lambda = \sigma_t h^{k+1} \varphi_{k+1}(h). \quad (9)$$

For instance, in the case when $k = 0$, it is easy to see that $\int_{\lambda_s}^{\lambda_t} e^{-\lambda} \mathrm{d}\lambda = \sigma_t(e^h - 1)$ and $\int_{\lambda_s}^{\lambda_t} e^{-\lambda} \mathrm{d}\bar{\boldsymbol{\omega}}_\lambda$ obeys a normal distribution with zero mean, and one can analytically compute its variance:

$$\int_{\lambda_s}^{\lambda_t} e^{-2\lambda} \mathrm{d}\lambda = \frac{\sigma_t^2}{2}(e^{2h} - 1). \quad (10)$$

**The EDM case.** Denote $\sigma_d^2$ the variance of the considered initial dataset and set

$$\sigma_t := t, \alpha_t := 1, \nabla_{\mathbf{x}_t} \log p_t(\mathbf{x}_t) \simeq \frac{1}{t^2} \left[ \frac{\sigma_d^2 \mathbf{x}_t}{t^2 + \sigma_d^2} + \frac{t\sigma_d}{\sqrt{t^2 + \sigma_d^2}} F_\theta \left( \frac{\mathbf{x}_t}{\sqrt{t^2 + \sigma_d^2}}, \frac{\log(t)}{4} \right) \right]. \quad (11)$$

These parameters correspond to the preconditioned EDM framework introduced in [16, Sec. 5, App. B.6]. The following result is the basis for constructing our customized SEEDS in this case, and for which we report experimental results in Table 1. For simplicity, we will write $F_\theta(\mathbf{x}_t, t)$ for the preconditioned model in (11) and we refer to Appendix B for details.

**Proposition 3.2.** *Let $t < s$. The analytic solution at time $t$ of* (2) *with coefficients* (11) *and initial value $\mathbf{x}_s$ is, for $\ell = 1$,*

$$\mathbf{x}_t = \frac{t^2 + \sigma_d^2}{s^2 + \sigma_d^2} \mathbf{x}_s + 2(t^2 + \sigma_d^2) \int_{\lambda_s}^{\lambda_t} e^{-\lambda} \hat{F}_\theta(\hat{\mathbf{x}}_\lambda, \lambda) \mathrm{d}\lambda - \sqrt{2}(t^2 + \sigma_d^2) \int_{\lambda_s}^{\lambda_t} e^{-\lambda} \mathrm{d}\overline{\boldsymbol{\omega}}_\lambda, \quad (12)$$

*where $\lambda_t := -\log \left[ \frac{t}{\sigma_d \sqrt{t^2 + \sigma_d^2}} \right]$. In the case when $\ell = 0$, it is given by*

$$\mathbf{x}_t = \sqrt{\frac{t^2 + \sigma_d^2}{s^2 + \sigma_d^2}} \mathbf{x}_s + \sqrt{t^2 + \sigma_d^2} \int_{\lambda_s}^{\lambda_t} e^{-\lambda} \hat{F}_\theta(\hat{\mathbf{x}}_\lambda, \lambda) \mathrm{d}\lambda, \quad \lambda_t := -\log \left[ \arctan \left[ \frac{t}{\sigma_d} \right] \right]. \quad (13)$$

*Remark* 3.3. One can wonder about the generality of such change of variables. Our method is very general in that one can always make such change of variables with very mild regularity conditions: for $c : [0, T] \longrightarrow \mathbb{R}^{>0}$ integrable, with primitive $C(t) > 0$, we have $c(t) = e^{\log(c(t))}$. This means we can write $c(t) = \dot{C}(t) = e^{\lambda_t} \dot{\lambda}_t$ with $\lambda_t = \log(C(t))$. In other words, for such $c$, we have

$$\int_s^t c(\tau) \mathrm{d}\tau = \int_s^t e^{\lambda_\tau} \dot{\lambda}_\tau \mathrm{d}\tau = \int_{\lambda_s}^{\lambda_t} e^\lambda \mathrm{d}\lambda.$$

## 4 Higher Stage SEEDS for DPMs

In this section we present our SEEDS algorithms by putting together all the ingredients presented in the previous section. Let $t < s$. In all what follows, we consider the analytic solution at time $t$ of the RSDE (2) with coefficients (7), $h = \lambda_t - \lambda_s$ and initial value $\mathbf{x}_s$. Plugging (9) with $k = 0$ and (10) into the exact solution (8) allow us to infer the first SEEDS scheme, given by iterations of the form

$$\tilde{\mathbf{x}}_t = \frac{\alpha_t}{\alpha_s} \tilde{\mathbf{x}}_s - 2\bar{\sigma}_t(e^h - 1)\hat{F}_\theta(\hat{\mathbf{x}}_{\lambda_s}, \lambda_s) - \bar{\sigma}_t \sqrt{e^{2h} - 1}\epsilon, \qquad \epsilon \sim \mathcal{N}(\mathbf{0}, \mathbf{I}_d). \quad (14)$$

The following Theorem gives strong order convergence guarantees for this method, which we call SEEDS-1, under mild conditions which apply to all our experiments. We stress out that its proof (App. C) is a non-trivial result, it is fundamentally different in nature from [22] and involves mathematical tools which have no deterministic counterparts.

**Theorem 4.1.** *Under Assumption C.1, the numerical solution $\tilde{\mathbf{x}}_t$ produced by the SEEDS-1 method* (14) *converges to the exact solution $\mathbf{x}_t$ of*

$$\mathrm{d}\mathbf{x}_t = [f(t)\mathbf{x}_t + g^2(t)\bar{\sigma}_t^{-1} F_\theta(\mathbf{x}_t, t)]\mathrm{d}t + g(t)\mathrm{d}\boldsymbol{\omega}_t, \qquad (\bar{\sigma}_t^{-1} := 1/\bar{\sigma}_t) \quad (15)$$

*with coefficients* (7) *in Mean-Square sense with strong order 1.0: there is a constant $C > 0$ such that*

$$\sqrt{\mathbb{E} \left[ \sup_{0 \leqslant t \leqslant 1} |\tilde{\mathbf{x}}_t - \mathbf{x}_t|^2 \right]} \leqslant Ch, \qquad as \ h \longrightarrow 0.$$

**Higher stage SEEDS.** As announced, by fully exploiting the analytic computations enabled by the expansion (9) we now turn into crafting our multi-step SEEDS. Usually, SDE solvers are constructed by using the full Itô-Taylor expansion of the SDE solutions and usually need a big number of evaluations of the network $\hat{F}_\theta$ to achieve higher order of convergence. As our main concern is to present stochastic solvers with a minimal amount of NFE, we choose to truncate such Itô-Taylor expansion so that the neural networks only appear in the deterministic contributions.

| **Algorithm 1** Iterative procedure | **Algorithm 2** SEEDS-1$(F_\theta, \tilde{\mathbf{x}}_s, s, t)$ |
|---|---|
| **Input:** initial value $\mathbf{x}_T$, steps $\{t_i\}_{i=0}^M$, model $F_\theta$
Initialize $\tilde{\mathbf{x}}_{t_0} \leftarrow \mathbf{x}_T$
**for** $i = 1$ **to** $M - 1$ **do**
$\quad (t, s) \leftarrow (t_i, t_{i-1}), \quad h \leftarrow \lambda_t - \lambda_s$
$\quad \tilde{\mathbf{x}}_t = \text{SEEDS-k}(F_\theta, \tilde{\mathbf{x}}_s, s, t)$
**end for**
Return $\tilde{\mathbf{x}}_{t_M} \leftarrow \text{last-step}(\tilde{\mathbf{x}}_{t_{M-1}}, t_{M-1}, t_M)$ | $z \leftarrow \mathcal{N}(0, 1)$
$\tilde{\mathbf{x}}_t \leftarrow \frac{\alpha_t}{\alpha_s}\tilde{\mathbf{x}}_s - 2\bar{\sigma}_t\left(e^h - 1\right)F_\theta(\tilde{\mathbf{x}}_s, s) - \bar{\sigma}_t\sqrt{e^{2h} - 1}z$ |

---

**Algorithm 3** SEEDS-2$(F_\theta, \tilde{\mathbf{x}}_s, s, t)$

$s_1 \leftarrow t_\lambda(\lambda_s + \frac{h}{2}), \quad (z^1, z^2) \leftarrow \mathcal{N}(0, \text{Id}) \otimes \mathcal{N}(0, \text{Id})$

$\mathbf{u} \leftarrow \frac{\alpha_{s_1}}{\alpha_s}\tilde{\mathbf{x}}_s - 2\bar{\sigma}_{s_1}\left(e^{\frac{h}{2}} - 1\right)F_\theta(\tilde{\mathbf{x}}_s, s) - \boldsymbol{A}, \qquad \boldsymbol{A} := \bar{\sigma}_{s_1}\sqrt{e^h - 1}z^1$

$\tilde{\mathbf{x}}_t \leftarrow \frac{\alpha_t}{\alpha_s}\tilde{\mathbf{x}}_s - 2\bar{\sigma}_t(e^h - 1)F_\theta(\mathbf{u}, s_1) - \boldsymbol{B}, \qquad \boldsymbol{B} := \bar{\sigma}_t(\sqrt{e^{2h} - e^h}z^1 + \sqrt{e^h - 1}z^2)$

---

**Algorithm 4** SEEDS-3$(F_\theta, \tilde{\mathbf{x}}_s, s, t)$ with $0 < r_1 < r_2 < 1$

$s_1 \leftarrow t_\lambda\left(\lambda_s + r_1 h\right), \quad s_2 \leftarrow t_\lambda\left(\lambda_s + r_2 h\right), \quad (z^1, z^2, z^3) \leftarrow \mathcal{N}(0, \text{Id})^{\otimes 3}$

$\mathbf{u}_1 \leftarrow \frac{\alpha_{s_1}}{\alpha_s}\tilde{\mathbf{x}}_s - 2\bar{\sigma}_{s_1}\left(e^{r_1 h} - 1\right)F_\theta(\tilde{\mathbf{x}}_s, s) - \bar{\sigma}_{s_1}\sqrt{e^{2r_1 h} - 1}z^1$

$\boldsymbol{A} \leftarrow \bar{\sigma}_{s_2}(\sqrt{e^{2r_2 h} - e^{2r_1 h}}z^1 + \sqrt{e^{2r_1 h} - 1}z^2)$

$\mathbf{u}_2 \leftarrow \frac{\alpha_{s_2}}{\alpha_s}\tilde{\mathbf{x}}_s - 2\bar{\sigma}_{s_2}\left(e^{r_2 h} - 1\right)F_\theta(\tilde{\mathbf{x}}_s, s) - 2\frac{\bar{\sigma}_{s_2}r_2}{r_1}\left(\frac{e^{r_2 h} - 1}{r_2 h} - 1\right)(F_\theta(\mathbf{u}_1, s_1) - F_\theta(\tilde{\mathbf{x}}_s, s))) - \boldsymbol{A}$

$\boldsymbol{B} \leftarrow \bar{\sigma}_t(\sqrt{e^{2h} - e^{2r_2 h}}z^1 + \sqrt{e^{2r_2 h} - e^{2r_1 h}}z^2 + \sqrt{e^{2r_1 h} - 1}z^3)$

$\tilde{\mathbf{x}}_t \leftarrow \frac{\alpha_t}{\alpha_s}\tilde{\mathbf{x}}_s - 2\bar{\sigma}_t\left(e^h - 1\right)F_\theta(\tilde{\mathbf{x}}_s, s) - 2\frac{\bar{\sigma}_t}{r_2}\left(\frac{e^h - 1}{h} - 1\right)(F_\theta(\mathbf{u}_2, s_2) - F_\theta(\tilde{\mathbf{x}}_s, s))) - \boldsymbol{B}$

---

**Proposition 4.2.** *Assume that $\hat{F}_\theta$ is a $\mathcal{C}^{2n+1}$-function with respect to $\lambda$. Then the truncated Itô-Taylor expansion of* (8) *reads, for $\epsilon \sim \mathcal{N}(\mathbf{0}, \mathbf{I}_d)$,*

$$\mathbf{x}_t = \frac{\alpha_t}{\alpha_s}\mathbf{x}_s - 2\bar{\sigma}_t \sum_{k=0}^n h^{k+1}\varphi_{k+1}(h)\hat{F}_\theta^{(k)}(\hat{\mathbf{x}}_{\lambda_s}, \lambda_s) - \bar{\sigma}_t\sqrt{e^{2h} - 1}\epsilon + \mathcal{R}_{n+1}, \qquad (16)$$

*with $\hat{F}_\theta^{(k)}(\hat{\mathbf{x}}_\lambda, \lambda) = L_\lambda^k \hat{F}_\theta(\hat{\mathbf{x}}_\lambda, \lambda)$, with $L_\lambda$ is an infinitesimal operator defined in Appendix E.2.2 and $\mathcal{R}_{n+1}$ consists on the usual deterministic residue and all iterated integrals of length at greater or equal to 2 in which there is at least one stochastic component among them.*

Our approach for constructing derivative-free 2-stage and 3-stage SEEDS schemes consists on exploiting the analytic computation of the Itô-Taylor coefficients in Proposition 4.2 and replace the $\hat{F}_\theta^{(k)}(\hat{\mathbf{x}}_\lambda, \lambda)$ terms by well-adapted correction terms which *do not need any derivative evaluation* and dropping the $\mathcal{R}_{n+1}$ contribution as in the Runge-Kutta approach.

**Markov-preserving noise decomposition.** We use collocation methods for constructing higher-stage derivative-free solvers. Although the chosen truncated Itô-Taylor expansion produces approximations for the deterministic integral similar to [22], adding the corresponding noise contribution found by the SETD method at each step does not yield Markov chains in general. The reason is that stochastic integrals on overlapping time intervals need to be dependent, a phenomenon that has no deterministic counterpart. As such, our last and key element to construct SEEDS consists on a novel decomposition of stochastic integrals which enforces the Markov property for multi-stage SEEDS.

Algorithms 1 to 4 prescribe all SEEDS schemes obtained by this procedure in the VP case. We now show (see App. C for the proofs) that all methods yield Markov chains and are weakly convergent.

**Proposition 4.3.** *The sequences $\{\tilde{\mathbf{x}}_t\}_t$ induced by the choice of stochastic noise contributions presented in Algorithms.3 and 4 satisfy the Markov property.*

**Corollary 4.4.** *Under Assumption C.2, the numerical solutions $\tilde{\mathbf{x}}_t$ produced by the SEEDS methods* (3) *and* (4) *converge to the exact solution $\mathbf{x}_t$ of* (15) *with coefficients* (7) *in weak sense with global order 1 in both cases: there is a constant $C > 0$ such that, for any continuous bounded function $G$:*

$$|\mathbb{E}[G(\tilde{\mathbf{x}}_{t_M})] - \mathbb{E}[G(\mathbf{x}_{t_M})]| \leqslant Ch.$$

**Comparison with existing sampling methods.** Let us now examine the connection between SEEDS and existing sampling techniques used for DPMs, emphasizing the contrasts between them.

The main distinctive feature of SEEDS is that they are *off-the-shelf* solvers. This means that, not only they are *training-free*, contrary to [9], but they do not require any kind of optimization procedure to achieve their optimal results. This is in contrast to methods such as: gDDIM, which is training-free but not off-the-shelf as one needs to make preliminary optimization procedures such as simulating the transition matrix of their method in the CLD case; Heun-Like method from EDM (for all baseline models and the EDM-optimized models for ImageNet) since they need preliminary optimization procedures on 4 parameters which actually break the convergence criteria. Moreover, neither gDDIM, EDM nor the SSCS method in [10] present full proofs of convergence for their solvers. Also, both DEIS and gDDIM identify their methods with stochastic DDIM theoretically, but the poor results obtained by their stochastic solvers do not yield to further experimentation in their works. In a way, SEEDS can be thought as improved and generalized DPM-Solver to SDEs. Nevertheless, such generalization is not incremental as the tools for proving convergence in our methods involve concepts which are exclusive to SDEs. We now make rigorous statements of the above discussion.

**Proposition 4.5.** *Consider the SEEDS approximation of* (15) *with coefficients* (7). *Then*

1. *If we set $g = 0$ in* (15), *the resulting SEEDS do not yield DPM-Solver.*

2. *If we parameterize* (15) *in terms of the data prediction model $D_\theta$, the resulting SEEDS are not equivalent to their noise prediction counterparts defined in Alg. 1 to 4.*

3. *The gDDIM solver [41, Th. 1] equals to SEEDS-1 in the data prediction mode, for $\ell = 1$.*

The first point makes it explicit that SEEDS are not incremental based on DPM-Solver. The second point in Prop. 4.5 is analog to the result in Appendix B of [23], where the authors compare DPM-Solver2 and DPM-Solver++(2S), that is the noise and data prediction approaches, and find that they do not equate. The last point exhibits gDDIM as a special case of SEEDS-1 for isotropic DPMs.

*Remark* 4.6. Building solvers from the representation of the exact solution in (5) requires computing the transition matrix $\Phi_A(t, s)$, which cannot be analytically computed for non-isotropic DPMs such as CLD [10]. Nevertheless, the SEEDS approach can be applied in this scenario in at least two different ways. On the one hand, the SSCS method from [10] resides splitting $\Phi_A(t, s)$ into two separate terms. The first can be analytically computed. The second describes the evolution of a semi-linear differential equation [10, Eq. 92]. While [10] approximates the latter by the Euler method, crafting exponential integrators for approximating such DE may yield an acceleration of the SSCS method. On the other hand, gDDIM [41] proposes an extension of DEIS sampling [40] to CLD by setting a pre-sampling phase [41, App. C.4] in which they compute an approximation of $\Phi_A(t, s)$ in order to apply their method, and the latter was shown in Prop. 4.5. to be a special case of our method. Unfortunately, the authors did not release pre-trained models in [41], and the latter are not the same as those in [10]. Sampling in this scenario may also benefit from our approach.

## 5 Experiments

We compare SEEDS with several previous methods on discretely and continuously pre-trained DPMs. We report results of many available sources, such as DDPM [12], Analytic DDPM [2], PNDM [20], GGF [15], DDIM [33], gDDIM [41], DEIS [40] and DPM-Solver [22]. Although we do not include training-based schemes here, we still included GENIE [9], which trains a small additional network but still solves the correct generative ODE at higher-order. For each experiment, we compute the FID score for 50K sampled images on multiple runs and report the minimum along different solvers. Details on model specifications and experiment illustrations are shown in Appendix F.

**Practical considerations.** For continuously trained models, SEEDS use the EDM discretization [16, Eq. 5] with default parameters and does *not* use the *last-step iteration trick*, meaning that the last iteration of SEEDS is trivial. For discretely trained models, SEEDS use the linear step schedule in the interval $[\lambda_{t_0}, \lambda_{t_N}]$ interval following [22, Sec. 3.3, 3.4]. All the reported SEEDS results were obtained using the noise prediction mode. We conducted comparative experiments on SEEDS for both the data and noise prediction modes and found better results with the latter (see Tab. 3 for details). EDM solvers [16, Alg. 2] depend on four parameters controlling the amount of noise to

Table 1: Sample quality measured by FID↓ on pre-trained DPMs. We report the minimum FID obtained by each model and the NFE at which it was obtained. For CIFAR, CelebA and FFHQ, we use baseline pre-trained models [34, 16]. For ImageNet, we use the optimized pre-trained model from [16]. *discrete-time model, ⋆continuous-time model, †recomputed FID for the non-deep model.

| SAMPLING METHOD | FID↓ | NFE |
|---|---|---|
| CIFAR-10* VP-UNCOND. | | |
| DDIM [33] | 3.95 | 1000 |
| ANALYTIC-DDPM [2] | 3.84 | 1000 |
| GENIE [9] | 3.64 | 25 |
| ANALYTIC-DDIM [2] | 3.60 | 200 |
| F-PNDM (LINEAR) [20] | 3.60 | 250 |
| DPM-SOLVER† [22] | 3.48 | 44 |
| F-PNDM (COSINE) [20] | 3.26 | 1000 |
| DDPM [12] | 3.16 | 1000 |
| SEEDS-3 (OURS) | **3.08** | **201** |
| CIFAR-10⋆ VP-COND. | | |
| DPM-SOLVER† [22] | 3.57 | 195 |
| EDM ($S_{\text{churn}} = 0$) [16] | 2.48 | 35 |
| SEEDS-3 (OURS) | **2.08** | **129** |
| CIFAR-10⋆ VP-UNCOND. | | |
| DPM-SOLVER† [22] | 2.59 | 51 |
| GGF [15] | 2.59 | 180 |
| GDDIM [41] | 2.56 | 100 |
| DEIS $\rho 3$KUTTA [40] | 2.55 | 50 |
| EULER-MARUYAMA [34] | 2.54 | 1024 |
| STOCHASTIC EDM [16] | 2.54 | 1534 |
| SEEDS-3 (OURS) | 2.39 | 165 |
| EDM (OPTIMIZED) [16] | **2.27** | **511** |

| SAMPLING METHOD | FID↓ | NFE |
|---|---|---|
| CELEBA-64* VP-UNCOND. | | |
| ANALYTIC-DDPM [2] | 5.21 | 1000 |
| DDIM [33] | 4.78 | 200 |
| DDPM [12] | 3.50 | 1000 |
| GDDIM [41] | 3.85 | 50 |
| ANALYTIC-DDIM [2] | 3.13 | 1000 |
| 3-DEIS [40] | 2.95 | 50 |
| F-PNDM (LINEAR) [20] | 2.71 | 250 |
| DPM-SOLVER [22] | 2.71 | 36 |
| SEEDS-3 (OURS) | **1.88** | **90** |
| FFHQ-64⋆ VP-UNCOND. | | |
| DPM-SOLVER† [22] | 3.52 | 90 |
| SEEDS-3 (OURS) | 3.40 | 150 |
| EDM ($S_{\text{churn}} = 0$) [16] | **3.39** | **79** |
| IMAGENET-64 EDM-COND. | | |
| DPM-SOLVER† [22] | 3.01 | 270 |
| EDM ($S_{\text{churn}} = 0$) [16] | 2.22 | 511 |
| SEEDS-3 (OURS) | 1.38 | 270 |
| EDM (OPTIMIZED) [16] | **1.36** | **511** |

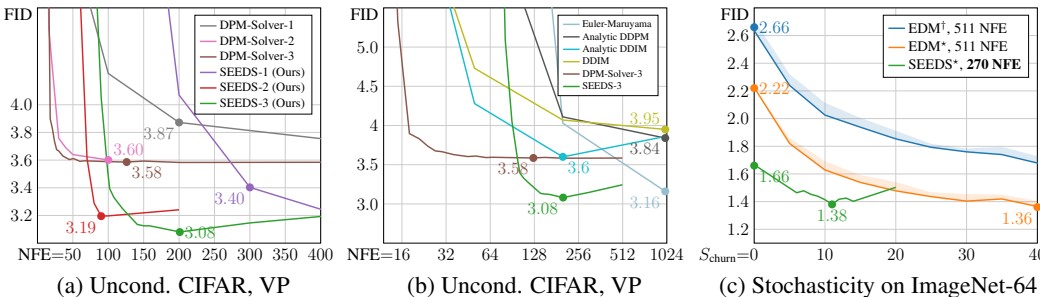

(a) Uncond. CIFAR, VP    (b) Uncond. CIFAR, VP    (c) Stochasticity on ImageNet-64

Figure 1: (a-b) Comparison of sample quality measured by FID ↓ of SEEDS, DPM-Solver and other methods for discretely trained DPMs on CIFAR-10 with varying number of function evaluations. (c) Effect of $S_{\text{churn}}$ on SEEDS-3 (at NFE = 270) and EDM method (at NFE = 511) on class-conditional ImageNet-64. †baseline ADM model. ⋆EDM preconditioned model.

be injected in a specific sub-interval of the iterative procedure. We consider three scenarios: when stochasticity is injected along all the iterative procedure, we denote it stochastic EDM, when no stochasticity is injected we denote it EDM ($S_{\text{churn}} = 0$) and we denote EDM (Optimized) the case where such parameters were subject to an optimization procedure. To better evaluate sampling quality along the sample pre-trained DPM, we recalculate DPM-Solver for sampling from the *non-deep* VP continuous model on the CIFAR-10 dataset. All implementation details can be found in Appendix D.

**Comparison with previous works.** In Table 1 we compare SEEDS with other sampling methods for pre-trained DPMs, and report the minimum FID obtained and their respective NFE. For each of the reported pre-trained models in CIFAR-10, CelebA-64 and ImageNet-64, SEEDS outperform all off-the-shelf methods in terms of quality with relatively low NFEs. For the discrete pre-trained DPM on CIFAR-10 (VP Uncond.) it is $\sim 5\times$ faster than the second best performant solver. Additionally, SEEDS remain competitive with the optimized EDM sampler. For ImageNet-64, it is nearly as good

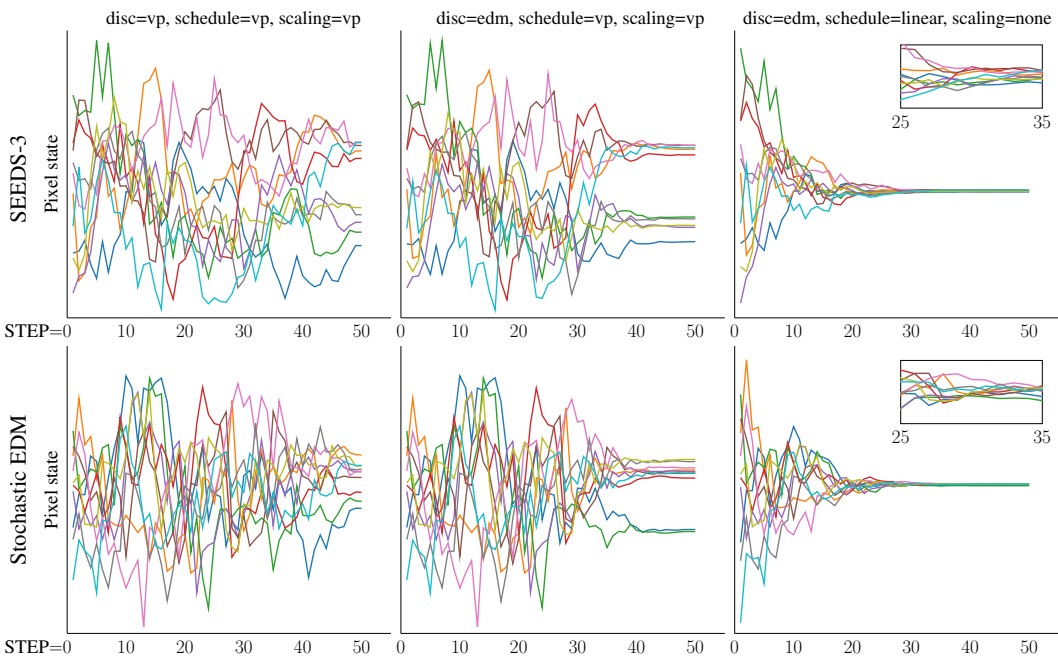

Figure 2: Trajectories of 10 pixels (R channel) sampled from SEEDS (1st line) and Stochastic EDM (2nd line) on the optimized pre-trained model [16] on ImageNet64. Schedule=scaling=vp corresponds to the VP coefficients in (7) and schedule=linear, scaling=none to the EDM coefficients (11). We use the time discretizations disc=vp (linear) and disc=edm given in [16, Tab.1].

as the optimized EDM sampler while being almost twice faster than the latter. Figure 1 (a) compares the FID score of SEEDS and DPM-Solver with varying NFEs. While DPM-Solver methods stabilize faster in a very low NFE regime, our methods eventually surpass them. Interestingly, after reaching their minimum, SEEDS methods tend to become worse at higher NFEs, a fact that is also visible in Figure 1 (b), where we notice that such phenomenon is also present on other SDE methods. We report in Appendix F, the results of our SEEDS methods in the low NFE regime and connect their behavior with their proven convergence rate.

**Combining SEEDS with other methods.**    While being an off-the-shelf method, SEEDS can be combined with the Churn-like method used in EDM incurring into SDE solvers with an additional source of stochasticity. As done in [16], we evaluate the effect of this second kind of stochasticity, measured by a parameter denoted $S_{\text{churn}}$. Figure 1 (c) shows that SEEDS and EDM show similar behavior, although SEEDS-3 is twice faster, more sensitive to $S_{\text{churn}}$, and quickly achieves comparable performance to EDM. This indicates that SEEDS could possibly outperform EDM after a proper parameter optimization that will be left for future works. Nevertheless, we highlight the fact that obtaining such optimal parameters is costly and might scale poorly.

**Stiffness reduction with SEEDS.**    In Fig. 2, we illustrate the impact of different choices of discretization steps, noise schedule and dynamic scaling on SEEDS and stochastic EDM. We see that choosing the EDM discretization over the linear one has the effect of flattening the pixel trajectories at latest stages of the simulation procedure. Also, choosing the parameters (11) over those in (7) has the effect of greatly changing the distribution variances as the trajectories evolve. Notice that all the SEEDS trajectories seem perceptually more stable than those from EDM. It would be interesting to relate this to the *stiffness* of the semi-linear DE describing these trajectories, and to the magnitude of the parameters involved in the noise injection for EDM solver amplifying this phenomenon.

**Ablation studies.**    As said earlier, our principled use of the Chasles rule to enforce independence only on non-overlapping paths for SEEDS-2/3 ensures that the set of resulting iterations of our solvers satisfy the Markov property, is new and is the central key of success of SEEDS. To highlight this,

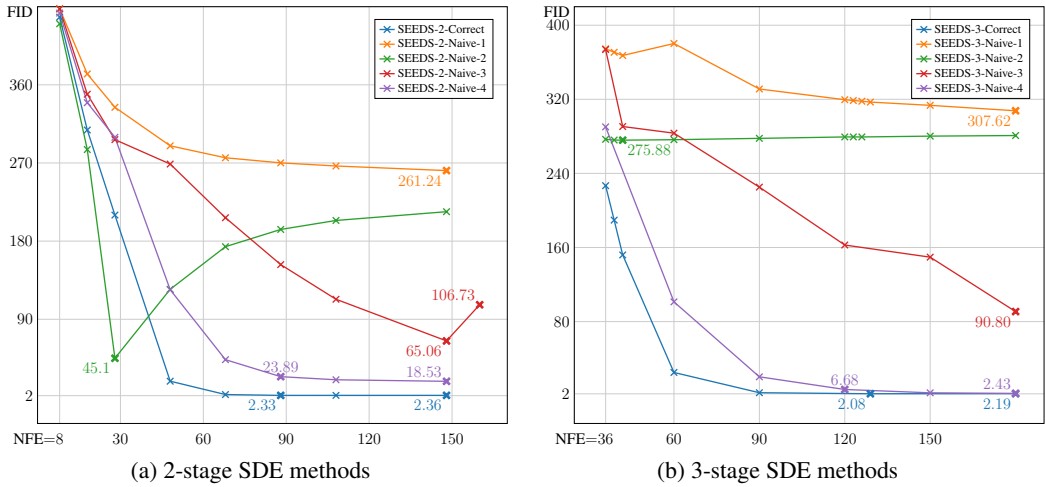

(a) 2-stage SDE methods          (b) 3-stage SDE methods

Figure 3: Quality (measured by FID at increasing NFEs) comparison of SEEDS-2/3 with the enumerated ablation versions of it on CIFAR-10 in the (baseline) VP conditional framework.

we conduct an ablation study on how 4 different combinations of the noise components $\boldsymbol{A}$ and $\boldsymbol{B}$ in Algorithms 3 and 4 have an impact on the sampling quality of SEEDS.

For simplicity, we explain this for SEEDS-2 (Alg. 3). Set $(z^1, z^2, z^3)$ three independent standard Gaussian random variables. Denote $\boldsymbol{A} = \bar{\sigma}_{s_1}\sqrt{e^h - 1}z^1$ for the noise contribution in the $\mathbf{u}$ term. We have the following choices for the noise contribution $\boldsymbol{B}$ in $\tilde{\mathbf{x}}_t$:

- SEEDS-2-Correct: our noise combination $\boldsymbol{B} = \bar{\sigma}_t\left(\sqrt{e^{2h} - e^h}z^1 + \sqrt{e^h - 1}z^2\right)$

- SEEDS-2-Naive-1: one noise per stage $\boldsymbol{B} = \bar{\sigma}_t\left(\sqrt{e^{2h} - e^h} + \sqrt{e^h - 1}\right)z^2$

- SEEDS-2-Naive-2: one noise per step $\boldsymbol{B} = \bar{\sigma}_t\left(\sqrt{e^{2h} - e^h} + \sqrt{e^h - 1}\right)z^1$

- SEEDS-2-Naive-3: one noise per integral evaluation $\boldsymbol{B} = \bar{\sigma}_t\left(\sqrt{e^{2h} - e^h}z^3 + \sqrt{e^h - 1}z^2\right)$

- SEEDS-2-Naive-4: noises in inverse position $\boldsymbol{B} = \bar{\sigma}_t\left(\sqrt{e^{2h} - e^h}z^2 + \sqrt{e^h - 1}z^1\right)$.

Figure 3 experimentally shows that all naive combinations of noises for SEEDS-2 and SEEDS-3 lead to FID/NFE curves with different behavior and a sharp drop of performance both in quality and speed in all of them, a phenomenon having no deterministic parallel.

## 6 Conclusions

Our focus is on addressing the challenge of training-free sampling from DPMs without compromising sampling quality. To achieve this, we introduce SEEDS, an off-the-shelf solution for solving diffusion SDEs. SEEDS capitalize on the semi-linearity of diffusion SDEs by approximating a simplified formulation of their exact solutions. Inspired by numerical methods for stochastic exponential integrators, we propose three SEEDS schemes with proven convergence order. They transform the integrals involved in the exact solution into exponentially weighted integrals, and estimate the deterministic one while analytically computing the variance of the stochastic integral. We extend our approach to handle other isotropic DPMs, and evaluate its performance on various benchmark tests. Our experiments demonstrate that SEEDS can generate images of optimal quality, outperforming existing SDE solvers while being $3 \sim 5\times$ faster. **Limitations and broader impact.** While SEEDS prioritize optimal quality sampling, they may require substantial computational resources and energy consumption, making them less suitable for scenarios where speed is the primary concern. In such cases, alternative ODE methods may be more appropriate. Additionally, as with other generative models, DPMs can be employed to create misleading or harmful content, and our proposed solver could inadvertently amplify the negative impact of generative AI for malicious purposes.

**Acknowledgments.** This work has been supported by the French government under the "France 2030" program, as part of the SystemX Technological Research Institute.

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

# Contents

## A   Discussion

**Why do SEEDS exhibit high FID scores in the low NFE regime?**    In Appendix F.4, we provide all tables with FID values at increasing NFEs corresponding to Fig. 1. Notice that SEEDS exhibit high FID scores in the low NFE regime. Let us discuss some theoretical facts that might explain this phenomenon. In [38, Theorem 1.], there is a formal explanation on why ODE samplers outperform SDE samplers in the small NFE regime and fall short in the large NFE regime. In particular, it is theoretically shown that, at large step sizes, it is the discretization error that dominates sampling errors while, at small step sizes, it is the approximation error that dominates it. One can infer that in the large NFE regime, SDE methods (with proven convergence orders) will outperform ODE methods in terms of sampling quality. It would be interesting to see if combining SEEDS with the new ideas in [38] might improve SEEDS in the low NFE regime.

Additionally, SDE solvers and ODE solvers may have different sample quality, even if the score models are optimal. Indeed, as proven in [21, Appendix B.], even if the score model has been trained to the optimal score function, the distributions between SDEs and ODEs are still different. This is because the distribution at time $T$ in the forward process is always not exactly a standard normal distribution as in the reverse SDE/ODE process, and thus the distributions at time 0 are different.

Finally, [38, 5] show that the overall generalization error divides into discretisation and approximation errors, indicating the theoretical role of the sampling error as a contribution to a DPM's performance. This is also consistent with the suggestion in [16, Section 5], stating that more diverse datasets continue to benefit from stochastic sampling rather than deterministic sampling.

**Do SEEDS maintain good performances for higher resolution image generation?**    As training-free and optimization-free SDE solvers, SEEDS naturally maintain good performances in higher

resolution images in the realm of unguided image generation (unconditional and conditional). For unconditional generation using Latent Diffusion Model, SEEDS are able to generate good quality images already at 100 NFEs. Now, for guided image generation, higher-stage SEEDS will, as expected, see their performance sharply drop as the guidance scale grows for the same reasons DPM-solver do (see [23] for details). Yet, SEEDS-1 still maintains high quality sampling in this scenario. In Appendix F.8, Fig. 15 exhibits a $512^2$ image generated with SEEDS-1 at 90 NFEs on Stable Diffusion with default guidance scale.

**Do SEEDS exhibit other distinguishing features beyond generation optimal quality?** As SDE solvers, SEEDS have a distinctive (although indirect) capability in the realm of adversarial robustness compared to DPM-Solver:

1. For more than 2 years, the leader-board of RobustBench has been dominated by Diffusion-based data augmentation techniques on top of Adversarial Training. The current SOTA [28, 37] uses EDM-preconditioned DPMs and need to generate as many as 50M images to achieve SOTA robustness results. For ImageNet-64, [37] use the EDM pretrained model and optimization-based stochastic EDM sampler for data augmentation, leading to a 5% robust accuracy improvement compared to doing so for the baseline ADM pretrained model. Since SEEDS reach same FID quality as EDM, but twice faster, we believe it will have a positive impact in this domain, making diffusion- based data augmentation schemes more affordable with limited computational capacity.

2. The work [27] uses DPM-based adversarial purification as a test-time adversarial defense. The idea is to use off-the-shelf DPMs to annihilate the adversarial content in inputs in test-time before feeding it to a pretrained classifier. In [27, Table 6], one can see that SDE solvers show substantial robustness capabilities compared to ODE solvers.

We'd like to stress out that, although indirect, such capabilities are on the side of the sampling methods rather than on the side of intrinsic robustness properties of score-based learning method. More broadly, robustness properties of ML models determined as neural differential equations (DPMs being in this scope) has been studied in [39, 11].

**How do SEEDS compare to Stochastic Runge-Kutta methods?** Contrary to the ODE case, there are many stochastic Runge-Kutta approaches, usually tailored for SDEs of a specific form. Nonetheless, a common way of distinguishing solvers with same strong order is to assign them couples $(p_d, p_s)$, where $p_s$ is the (stochastic) strong convergence order and $p_d$ is the resulting (deterministic) order determined if setting $g = 0$ in the considered SDE i.e. when they are deterministic. For instance, [30, Tab. 6.2] determines solvers with orders (1,1.0) and (2,1.0) respectively and [30, Tab. 6.3] determines solvers SRA1 and SRA3 with orders (2,1.5) and (3,1.5) respectively. Many of these solvers' speed was already tested in [15, Table 3] on CIFAR-10 (VP). Experimentally, SEEDS-1 shows to be 6.83x faster than the baseline Euler-Maruyama scheme.

To our knowledge, the only available strong order SERKs method for SDEs with in-homogeneous diffusion coefficients are the exponential Euler-Maruyama (EEM) method [18] and the stochastic RK Lawson (SRKL) schemes [7]. In short, the SRKL schemes only compute analytically the linear coefficient and use the Integrating Factor (IF) method to approximate the integrals in the representation of the exact solution given by the variation-of-parameters formula. This way, by a special change of variables (see [7, Alg. 1]), one can create exponential integrator versions of many SDE methods. We implemented our own version of the SRKL schemes, which we denote SRKL-1/2/3, to take into account that $\sigma, \alpha$ are not constant and used the Integrating Factor method to approximate the integrals in the representation of the exact solution given by the variation-of-parameters formula and under the VP $\lambda$-change-of-variables. Interestingly, the SRKL schemes stabilize at increasing NFEs but at much higher FID values than their SETD counterparts.

In Table 2 below, we draw a comparison of SEEDS with current Stochastic RKL methods (on the $\lambda$ temporal parameter space) on CIFAR-10 for the discretely trained DPM in the VP unconditional regime.

**Is the proven convergence order of each of the SEEDS methods optimal?** The proposed convergence orders for each SEEDS-1/2/3 is optimal: this is a consequence of the general result from [6] about maximum convergence rates for SDE schemes with uncorrelated Gaussian increments.

Table 2: Comparison of SEEDS with adapted Stochastic RKL methods on CIFAR-10 in the VP unconditional discrete framework.

| METHOD \ NFE | 10 | 20 | 50 | 100 | BEST KNOWN |
|---|---|---|---|---|---|
| SRKL-1($\lambda$) | 332.52 | 282.96 | 33.42 | 8.62 | - |
| SEEDS-1 | 303.48 | 153.21 | 22.70 | 7.97 | (500 NFE) 3.13 |
| SRKL-2($\lambda$) | 475.20 | 469.64 | 134.82 | 7.74 | - |
| SEEDS-2 | 476.90 | 226.70 | 7.17 | 3.23 | (90 NFE) 3.21 |
| SRKL-3($\lambda$) | 462.24 | 376.15 | 8.36 | 7.46 | - |
| SEEDS-3 | 483.00 | 428.60 | 43.30 | 3.41 | (201 NFE) 3.08 |

The underlying idea, fully detailed in Appendix E, is that any solver with strong order $\geq 1.5$ has to account for double stochastic integrals in the non-truncated Itô-Taylor expansion, ultimately forcing any SRK-like solver to use correlated random variables (see [30, Tab. 6.3] and [17] more generally). SEEDS avoid this additional complexity but an interesting future avenue would be to extend SEEDS to the higher strong order case (and not in the IF approach but the SETD approach) as long as it doesn't incur into an explosion of needed NFEs per step to craft such solvers. Another interesting path would be to craft weak second order SERK methods for DPMs (the work [19] addressed this only for homogeneous semi-linear SDEs with constant linear coefficients).

## B    Detailed Derivation of the SEEDS Design Space

In Sections 2 and 3 we proposed a simplified presentation of the design space of diffusion models and of the ingredients that constitute our proposed SEEDS methodology. In this section, we further develop our presentation in a technical manner, making explicit the formalization of our design choices.

### B.1    The Isotropic General SDE framework

In Section 2, we presented a parametric family of differential equations (2) driving the generative process for DPMs, based on time-reversing the forward noising diffusion process (1). While doing so, we presented two parameters - the noise schedule $\sigma_t$ and the scaling $\alpha_t$ - for which the effects on DPMs have been widely studied in [16].

As the shape of the trajectories of (1) and (2) (for $\ell = 0, 1$) are defined by $\alpha_t$ and $\sigma_t$, we start by writing down, for the scaling $\mathbf{x}_t = \alpha_t \widehat{\mathbf{x}}_t$, the scaled generalization of the proposed SDEs in [16, Eq. 103] which unifies in a single framework the forward and reverse trajectories:

$$d\mathbf{x}_t^\pm = \left[ \frac{\dot{\alpha}_t}{\alpha_t} \mathbf{x}_t^\pm - \alpha_t^2 \dot{\sigma}_t \sigma_t \nabla_{\mathbf{x}_t^\pm} \log p(\widehat{\mathbf{x}}_t^\pm; \sigma_t) \pm \alpha_t^2 \dot{\sigma}_t \sigma_t \nabla_{\mathbf{x}_t^\pm} \log p(\widehat{\mathbf{x}}_t^\pm; \sigma_t) \right] dt + \alpha_t \sigma_t \sqrt{2\frac{\dot{\sigma}_t}{\sigma_t}} d\boldsymbol{\omega}_t^\pm. \quad (17)$$

Following [16], the previous VP, VE, iDDPM, DDIM and EDM frameworks all are unified as different choices of $\alpha_t, \sigma_t$, among other choices presented in [16, Tab. 1] and we will use this as a basis for all the proofs contained in this Appendix. In particular, forward time means taking $\mathbf{x}_t^+$ for which the score vanishes in this context. Now set $\mathbf{x}_t = \mathbf{x}_t^-$.

The formulation in (2) involves a family of backwards differential equations controlled by a parameter $\ell \in [0, 1]$ which all yield reverse-time processes for (1), a fact that can be obtained by studying the Fokker-Planck equation for marginals $p(\widehat{\mathbf{x}}_t^+; \sigma_t)$ of (17).

When $\ell = 1$, (2) the obtained SDE is known as the reverse SDE (RSDE) and, when $\ell = 0$, we obtain an ODE that is known as the Probability Flow ODE (PFO):

$$d\mathbf{x}_t = \left[ \frac{\dot{\alpha}_t}{\alpha_t} \mathbf{x}_t - \alpha_t^2 \dot{\sigma}_t \sigma_t \nabla_{\mathbf{x}_t} \log p(\widehat{\mathbf{x}}_t; \sigma_t) \right] dt. \quad (18)$$

Now, finding a minimum for the loss function in [16, Eq. 51] is formulated as a convex optimization problem. As such, for the ideal model $D(\mathbf{x}_t; \sigma_t) = \arg\min_D \mathcal{L}(D; \mathbf{x}_t, \sigma_t)$, the score function with

scaled input is expressed as

$$\nabla_{\mathbf{x}_t} \log p(\widehat{\mathbf{x}}_t; \sigma_t) \quad = \quad \frac{D(\widehat{\mathbf{x}}_t; \sigma_t) - \widehat{\mathbf{x}}_t}{\alpha_t \sigma_t^2}.$$

This ideal model is usually subtracted by a *raw* network $F$ in the form of a time-dependent preconditioning:

$$D(\mathbf{x}_t; \sigma_t) := c_1(t)\mathbf{x}_t + c_2(t)F(c_3(t)\mathbf{x}_t; c_4(t)), \qquad c_i(t) \in \mathbb{R}^d, \quad i = 1, \ldots, 4.$$

As such, we can express the score function as two parameterizations involving $D$ or $F$ as follows:

$$\nabla_{\mathbf{x}_t} \log p(\widehat{\mathbf{x}}_t; \sigma_t) \quad = \quad \frac{D(\widehat{\mathbf{x}}_t; \sigma_t) - \widehat{\mathbf{x}}_t}{\alpha_t \sigma_t^2} = \frac{(c_1(t) - 1)\widehat{\mathbf{x}}_t + c_2(t)F(c_3(t)\widehat{\mathbf{x}}_t; c_4(t))}{\alpha_t \sigma_t^2}. \quad (19)$$

Let us now denote $D_{\theta,t}^1 := D_\theta(\widehat{\mathbf{x}}_t; \sigma_t)$ for a pre-trained network approximating the ideal denoiser and let $D_{\theta,t}^2 := F_\theta(c_3(t)\widehat{\mathbf{x}}_t; c_4(t))$ be the corresponding raw pre-trained network. Substituting the score function in the RSDE and the PFO with each of these models yields four *different* differential equations with a neural network as one of their components. These are given, for $i = 1, 2$, by

$$d\mathbf{x}_t \quad = \quad [A^i(t)\mathbf{x}_t + B^i(t)D_{\theta,t}^i]dt + g(t)d\bar{\boldsymbol{\omega}}_t, \quad (20)$$

$$d\mathbf{x}_t \quad = \quad [A^{i+2}(t)\mathbf{x}_t + B^{i+2}(t)D_{\theta,t}^i]dt. \quad (21)$$

When $D_{\theta,t}^1$ (resp. $D_{\theta,t}^2$) is employed to replace the score function in (17) using (19), the resulting SDE (20) will be called *data (resp. noise) prediction neural SDE*. Proceeding analogously for the PFO (18) yield two ODEs (21) which will be called *data (resp. noise) prediction neural PFO*. The general form of the $A^i$ and $B^i$ coefficients determining each of these DEs is as follows:

$$A^1(t) = \frac{\dot{\alpha}_t}{\alpha_t} + 2\frac{\dot{\sigma}_t}{\sigma_t} \qquad\qquad B^1(t) = -2\alpha_t \frac{\dot{\sigma}_t}{\sigma_t} \qquad\qquad \text{(DP NRSDE)}$$

$$A^2(t) = \frac{\dot{\alpha}_t}{\alpha_t} + 2\frac{\dot{\sigma}_t}{\sigma_t}(1 - c_1(t)) \qquad B^2(t) = -2\alpha_t \frac{\dot{\sigma}_t}{\sigma_t}c_2(t) \qquad \text{(NP NRSDE)}$$

$$A^3(t) = \frac{\dot{\alpha}_t}{\alpha_t} + \frac{\dot{\sigma}_t}{\sigma_t} \qquad\qquad B^3(t) = -\alpha_t \frac{\dot{\sigma}_t}{\sigma_t} \qquad\qquad \text{(DP NPFO)}$$

$$A^4(t) = \frac{\dot{\alpha}_t}{\alpha_t} + \frac{\dot{\sigma}_t}{\sigma_t}(1 - c_1(t)) \qquad B^4(t) = -\alpha_t \frac{\dot{\sigma}_t}{\sigma_t}c_2(t) \qquad \text{(NP NPFO)}$$

*Remark* B.1. At first glance, it would seem misleading to differentiate four DEs as these essentially correspond to different choices of $\alpha_t, \sigma_t, c_1(t), \ldots, c_4(t)$. But the reason why we do so is that, after applying the *variation of constants* formula, each of these DEs will yield a different representation of their exact solutions (see (22) and (23) below). As we will see below, constructing exponential integrators heavily depends on such representation and will show to lead to four different modes of SEEDS solvers, each one showing different behavior and performance for sampling from pre-trained DPMs. As such, we will articulate this difference already at the DE formulation.

For $t < s$, the variation of constants formulae for NSDEs allows to represent the exact solutions of (20) as

$$\mathbf{x}_t = \Phi^i(t, s)\mathbf{x}_s + \int_s^t \Phi^i(t, \tau)B^i(\tau)D_{\theta,\tau}^i d\tau + \int_s^t \Phi^i(t, \tau)g(\tau)d\bar{\boldsymbol{\omega}}_t, \qquad i = 1, 2 \quad (22)$$

and those for NPFOs (21) as

$$\mathbf{x}_t = \Phi^i(t, s)\mathbf{x}_s + \int_s^t \Phi^i(t, \tau)B^i(\tau)D_{\theta,\tau}^{i-2} d\tau, \qquad i = 3, 4 \quad (23)$$

where

$$\Phi_{A^i}(t, s) = \exp\left(\int_s^t A^i(\tau)d\tau\right) \quad (24)$$

is called the transition matrix associated with $A^i(t)$ and is defined as the solution to

$$\frac{\partial}{\partial t}\Phi_{A^i}(t, s) = A^i(t)\Phi_{A^i}(t, s), \qquad \Phi_{A^i}(s, s) = \boldsymbol{I}_d.$$

When $A^i(t)$ is constant and $B^i(t) = 1$, there is a well-established literature on exponential ODE and SDE solvers with explicit *stiff order conditions* and prescribed by different forms of Butcher tableaux. When $A^i(t)$ is not constant, for the expression in (24) to make sense in the usual sense (in terms of exponential series expansion) instead of having to make use of time-ordered exponentials/ Magnus expansions, the $f(t) := A^i(t)$ coefficients must satisfy $[f^{(k)}(t), f^{(l)}(s)] = 0$. This condition is trivially satisfied here as the $A^i(t)$ considered here are $d$-dimensional diagonal matrices.

Notice that, if $A^i \neq A^j$ for some $i \neq j$, their associated transition matrices will not be equal. In particular, if $A^1 \neq A^2$, then the variances of the stochastic integrals in (22) are different for $i = 1$ and $i = 2$. This is the first step in explaining the statement in Rem. B.1, and we refer the reader to the proof of Proposition 4.5 where we put into evidence its validity.

## B.2 Re-framing and Generalizing Previous Exponential Solvers

### B.2.1 The VP case

Let $\tilde{\alpha}_t := \int_0^t (\beta_d \tau - \beta_m) \mathrm{d}\tau = \frac{1}{2}\beta_d t^2 + \beta_m t$, where $\beta_d > \beta_m > 0$. Set

$$f(t) := \frac{\mathrm{d}\log\alpha_t}{\mathrm{d}t}, \quad g(t) = \alpha_t \sqrt{\frac{\mathrm{d}[\sigma_t^2]}{\mathrm{d}t}}, \quad \sigma_t = \sqrt{e^{\tilde{\alpha}_t} - 1}, \quad \alpha_t = e^{-\frac{1}{2}\tilde{\alpha}_t} = \frac{1}{\sqrt{\sigma_t^2 + 1}}.$$

Recall that in the VP case, and the *noise prediction mode*, [22] construct exponential solvers on the base of the following ODE

$$\mathrm{d}\mathbf{x}_t = \left[ f(t)\mathbf{x}_t + \frac{g^2(t)}{2\bar{\sigma}_t} \epsilon_\theta(\mathbf{x}_t; t) \right] \mathrm{d}t, \quad t \in [T, 0], \tag{25}$$

where $\bar{\sigma}_t := \alpha_t \sigma_t$. The ODE (25) identifies with that in [16] for the VP case for which the authors identify the preconditioning

$$c_1(t) = 1, \quad c_2(t) = -\sigma_t, \quad c_3(t) = \frac{1}{\sqrt{\sigma_t^2 + 1}}, \quad c_4(t) = (M-1)\sigma^{-1}(\sigma_t) = (M-1)t.$$

As such, we obtain the following coefficients for the NP NPFO:

$$A^4(t) = \frac{\dot{\alpha}_t}{\alpha_t}, \quad B^4(t) = \alpha_t \dot{\sigma}_t, \quad \Phi^4(t, s) = \frac{\alpha_t}{\alpha_s}$$

and

$$\nabla_{\mathbf{x}_t} \log p(\hat{\mathbf{x}}_t; \sigma_t) = \frac{D_\theta(\hat{\mathbf{x}}_t; \sigma_t) - \hat{\mathbf{x}}_t}{\alpha_t \sigma_t^2} = \frac{(c_1(t) - 1)\hat{\mathbf{x}}_t + c_2(t)F(c_3(t)\hat{\mathbf{x}}_t; c_4(t))}{\alpha_t \sigma_t^2}$$
$$= \frac{-\sigma_t F_\theta(\mathbf{x}_t, (M-1)t)}{\alpha_t \sigma_t^2}.$$

### B.2.2 Proof of Proposition 3.1

First of all, denote $F_\theta(\mathbf{x}_t, (M-1)t) = \epsilon_\theta(\mathbf{x}_t, t)$. We have

$$f(t) = \frac{\mathrm{d}\log\alpha_t}{\mathrm{d}t}, \quad g^2(t) = 2\bar{\sigma}_t^2 \left( \frac{\mathrm{d}\log\bar{\sigma}_t}{\mathrm{d}t} - \frac{\mathrm{d}\log\alpha_t}{\mathrm{d}t} \right) = -2\bar{\sigma}_t^2 \frac{\mathrm{d}\lambda_t}{\mathrm{d}t}, \quad \frac{\bar{\sigma}_t}{\alpha_t} = e^{-\lambda_t}.$$

This way, one can directly relate $\lambda_t$ with the *signal-to-noise ratio* $\mathrm{SNR}(t) = \alpha_t^2/\bar{\sigma}_t^2$, also being used in [22]. As such, $\mathrm{SNR}(t)$ is strictly monotonically decreasing in time. Thus, the analytic solution to

(2) yields

$$\begin{aligned}
\mathbf{x}_t &= e^{\int_s^t f(\tau)\mathrm{d}\tau}\mathbf{x}_s + \int_s^t \left(e^{\int_\tau^t f(r)\mathrm{d}r}\frac{g^2(\tau)}{\bar\sigma_\tau}\boldsymbol{\epsilon}_\theta(\mathbf{x}_\tau,\tau)\right)\mathrm{d}\tau + \int_s^t \left(e^{\int_\tau^t f(r)\mathrm{d}r}g(\tau)\right)\mathrm{d}\bar{\boldsymbol{\omega}}(\tau) \\
&= \frac{\alpha_t}{\alpha_s}\mathbf{x}_s + \alpha_t\int_s^t \frac{g^2(\tau)}{\alpha_\tau\bar\sigma_\tau}\boldsymbol{\epsilon}_\theta(\mathbf{x}_\tau,\tau)\mathrm{d}\tau + \alpha_t\int_s^t \frac{g(\tau)}{\alpha_\tau}\mathrm{d}\bar{\boldsymbol{\omega}}(\tau) \\
&= \frac{\alpha_t}{\alpha_s}\mathbf{x}_s - \alpha_t\int_s^t \frac{2\sigma_\tau^2}{\alpha_\tau\bar\sigma_\tau}\frac{\mathrm{d}\lambda_\tau}{\mathrm{d}\tau}\boldsymbol{\epsilon}_\theta(\mathbf{x}_\tau,\tau)\mathrm{d}\tau + \alpha_t\int_s^t \frac{g(\tau)}{\alpha_\tau}\mathrm{d}\bar{\boldsymbol{\omega}}(\tau) \\
&= \frac{\alpha_t}{\alpha_s}\mathbf{x}_s - 2\alpha_t\int_s^t \frac{\bar\sigma_\tau}{\alpha_\tau}\frac{\mathrm{d}\lambda_\tau}{\mathrm{d}\tau}\boldsymbol{\epsilon}_\theta(\mathbf{x}_\tau,\tau)\mathrm{d}\tau + \alpha_t\int_s^t \frac{g(\tau)}{\alpha_\tau}\mathrm{d}\bar{\boldsymbol{\omega}}(\tau) \\
&= \frac{\alpha_t}{\alpha_s}\mathbf{x}_s - 2\alpha_t\int_s^t e^{-\lambda_\tau}\frac{\mathrm{d}\lambda_\tau}{\mathrm{d}\tau}\boldsymbol{\epsilon}_\theta(\mathbf{x}_\tau,\tau)\mathrm{d}\tau - \sqrt{2}\alpha_t\int_s^t e^{-\lambda_\tau}\sqrt{\frac{\mathrm{d}\lambda_\tau}{\mathrm{d}\tau}}\mathrm{d}\bar{\boldsymbol{\omega}}(\tau).
\end{aligned}$$

By using the change of variables to $\lambda(t)$, our equation now reads

$$\mathbf{x}_t = \frac{\alpha_t}{\alpha_s}\mathbf{x}_s - 2\alpha_t\int_{\lambda_s}^{\lambda_t} e^{-\lambda}\hat{\boldsymbol{\epsilon}}_\theta(\hat{\mathbf{x}}_\lambda,\lambda)\mathrm{d}\lambda - \sqrt{2}\alpha_t\int_{\lambda_s}^{\lambda_t} e^{-\lambda}\mathrm{d}\bar{\boldsymbol{\omega}}(\lambda). \tag{26}$$

Finally, notice that $\alpha_t = \sqrt{\frac{1}{1+e^{-2\lambda_t}}}$ and $\bar\sigma_t = \sqrt{\frac{1}{1+e^{2\lambda_t}}}$ so that (26) is equivalent to

$$\hat{\mathbf{x}}_{\lambda_t} = \frac{\hat\alpha_{\lambda_t}}{\hat\alpha_{\lambda_s}}\hat{\mathbf{x}}_{\lambda_s} - 2\hat\alpha_{\lambda_t}\int_{\lambda_s}^{\lambda_t} e^{-\lambda}\hat{\boldsymbol{\epsilon}}_\theta(\hat{\mathbf{x}}_\lambda,\lambda)\mathrm{d}\lambda - \sqrt{2}\hat\alpha_{\lambda_t}\int_{\lambda_s}^{\lambda_t} e^{-\lambda}\mathrm{d}\bar{\boldsymbol{\omega}}(\lambda).$$

This finishes the proof.

### B.2.3  Proof of Proposition 4.2

Recall that the functions $\varphi_k$ are the integrals

$$\varphi_{k+1}(t) = \int_0^1 e^{(1-\delta)t}\frac{\delta^k}{k!}\mathrm{d}\delta,$$

which satisfy $\varphi_k(0) = \frac{1}{k!}$. The truncated Itô-Taylor expansion of $\hat{\boldsymbol{\epsilon}}_\theta$ with respect to $\lambda$ reads

$$\hat{\boldsymbol{\epsilon}}_\theta(\hat{\mathbf{x}}_\lambda,\lambda) = \sum_{k=0}^{n}\frac{(\lambda-\lambda_s)^k}{k!}\hat{\boldsymbol{\epsilon}}_\theta^{(k)}(\hat{\mathbf{x}}_{\lambda_s},\lambda_s) + \mathcal{R}_{n+1},$$

where here $\hat{\boldsymbol{\epsilon}}_\theta^{(k)}$ denotes the $L_t^k$ operators defined in (58) applied to $\hat{\boldsymbol{\epsilon}}_\theta$. On the one hand, since $\int_{\lambda_s}^{\lambda_t} e^{-\lambda}d\lambda = \frac{\bar\sigma_t}{\alpha_t}(e^h-1)$, we obtain by iteratively integrating by parts

$$\begin{aligned}
\int_{\lambda_s}^{\lambda_t} e^{-\lambda}\hat{\boldsymbol{\epsilon}}_\theta(\hat{\mathbf{x}}_\lambda,\lambda)\mathrm{d}\lambda &= \sum_{k=0}^{n}\hat{\boldsymbol{\epsilon}}_\theta^{(k)}(\hat{\mathbf{x}}_{\lambda_s},\lambda_s)\int_{\lambda_s}^{\lambda_t} e^{-\lambda}\frac{(\lambda-\lambda_s)^k}{k!}\mathrm{d}\lambda + \mathcal{R}_{n+2} \\
&= \frac{\bar\sigma_t}{\alpha_t}\sum_{k=0}^{n}\hat{\boldsymbol{\epsilon}}_\theta^{(k)}(\hat{\mathbf{x}}_{\lambda_s},\lambda_s)h^{k+1}\varphi_{k+1}(h) + \mathcal{R}_{n+2}.
\end{aligned}$$

On the other hand, we have $s > t, h = \lambda_t - \lambda_s > 0$. Note that since the stochastic integrals $\int_{\lambda_s}^{\lambda_t} e^{-\lambda}\mathrm{d}\bar{\boldsymbol{\omega}}(\lambda)$ are measurable with respect to $(\bar{\boldsymbol{\omega}}(\lambda) - \bar{\boldsymbol{\omega}}(\lambda_s), 0 \leq \lambda \leq \lambda_t - \lambda_s)$, they are independent on disjoint time intervals by the independence of increments property of Brownian motion. Thus the random variable $\epsilon := \epsilon_{s,t}$ in our algorithms are independent on disjoint time intervals. We then write

$$\begin{aligned}
\int_{\lambda_s}^{\lambda_t} e^{-\lambda}\mathrm{d}\bar{\boldsymbol{\omega}}(\lambda) &= \mathcal{N}\left(0, \int_{\lambda_s}^{\lambda_t} e^{-2\lambda}\mathrm{d}\lambda\right) \\
&= \frac{1}{\sqrt{2}}\sqrt{e^{-2\lambda_s} - e^{-2\lambda_t}}\epsilon, \quad \epsilon \sim \mathcal{N}(\mathbf{0},\mathbf{I}_d) \\
&= \frac{1}{\sqrt{2}}\sqrt{\left(\frac{\bar\sigma_s}{\alpha_s} - \frac{\bar\sigma_t}{\alpha_t}\right)\left(\frac{\bar\sigma_s}{\alpha_s} + \frac{\bar\sigma_t}{\alpha_t}\right)}\epsilon, \quad \epsilon \sim \mathcal{N}(\mathbf{0},\mathbf{I}_d) \\
&= \frac{1}{\sqrt{2}}\frac{\bar\sigma_t}{\alpha_t}\sqrt{e^{2h}-1}\epsilon, \quad \epsilon \sim \mathcal{N}(\mathbf{0},\mathbf{I}_d).
\end{aligned}$$

In conclusion, the truncated Itô-Taylor expansion of the analytic expression

$$\mathbf{x}_t \;=\; \frac{\alpha_t}{\alpha_s}\mathbf{x}_s - 2\alpha_t \int_{\lambda_s}^{\lambda_t} e^{-\lambda}\widehat{\boldsymbol{\epsilon}}_\theta(\widehat{\mathbf{x}}_\lambda,\lambda)\mathrm{d}\lambda - \sqrt{2}\alpha_t \int_{\lambda_s}^{\lambda_t} e^{-\lambda}\mathrm{d}\bar{\boldsymbol{\omega}}(\lambda) \tag{27}$$

simplifies to

$$\mathbf{x}_t \;=\; \frac{\alpha_t}{\alpha_s}\mathbf{x}_s - 2\bar{\sigma}_t \sum_{k=0}^{n} h^{k+1}\varphi_{k+1}(h)\widehat{\boldsymbol{\epsilon}}_\theta^{(k)}(\widehat{\mathbf{x}}_{\lambda_s},\lambda_s) - \bar{\sigma}_t\sqrt{e^{2h}-1}\,\boldsymbol{\epsilon} + \mathcal{R}_{n+2}, \quad \boldsymbol{\epsilon}\sim\mathcal{N}(\mathbf{0},\mathbf{I}_d).$$

This finishes the proof.

### B.2.4 Generalization to the remaining data prediction and deterministic modes

Propositions 3.1 and 4.2 consist on the first steps for crafting SEEDS solvers in the VP case associated to the noise prediction neural RSDE (NP NRSDE). Generalizing the above procedure for crafting SEEDS for the 4 modes associated to (22) and (23) yields the following sets of coefficients

$$A^1(t) = \frac{\dot{\alpha}_t}{\alpha_t} + 2\frac{\dot{\sigma}_t}{\sigma_t} \qquad\qquad B^1(t) = -2\alpha_t\frac{\dot{\sigma}_t}{\sigma_t} \qquad\qquad \text{(DP NRSDE)}$$

$$A^2(t) = \frac{\dot{\alpha}_t}{\alpha_t} \qquad\qquad B^2(t) = 2\alpha_t\dot{\sigma}_t \qquad\qquad \text{(NP NRSDE)}$$

$$A^3(t) = \frac{\dot{\alpha}_t}{\alpha_t} + \frac{\dot{\sigma}_t}{\sigma_t} \qquad\qquad B^3(t) = -\alpha_t\frac{\dot{\sigma}_t}{\sigma_t} \qquad\qquad \text{(DP NPFO)}$$

$$A^4(t) = \frac{\dot{\alpha}_t}{\alpha_t} \qquad\qquad B^4(t) = \alpha_t\dot{\sigma}_t \qquad\qquad \text{(NP NPFO)}$$

We readily obtain

$$\Phi^2(t,s) = \Phi^4(t,s) = \frac{\alpha_t}{\alpha_s}, \qquad \Phi^3(t,s) = \frac{\bar{\sigma}_t}{\bar{\sigma}_s}, \qquad \Phi^1(t,s) = \frac{\sigma_t^2\alpha_t}{\sigma_s^2\alpha_s}.$$

Then, by setting the simpler change of variables $\lambda_t := -\log(\sigma_t)$, we obtain

$$\int_s^t \Phi^4(t,\tau)B^4(\tau)\mathrm{d}\tau = \alpha_t\int_s^t \frac{1}{\alpha_\tau}\alpha_\tau\dot{\sigma}_\tau\mathrm{d}\tau = \alpha_t\int_s^t \dot{\sigma}_\tau\mathrm{d}\tau = -\alpha_t\int_{\lambda_s}^{\lambda_t} e^{-\lambda}\mathrm{d}\lambda = -\bar{\sigma}_t(e^h-1).$$

By recursion we obtain

$$\int_s^t \Phi^4(t,\tau)B^4(\tau)F_{\theta,\tau}\mathrm{d}\tau = -\bar{\sigma}_t\sum_{k=0}^{n-1} h^{k+1}\varphi_{k+1}(h)F_{\theta,s}^{(k)} + \mathcal{O}(h^{n+1}).$$

In the same way, we obtain

$$\int_s^t \Phi^3(t,\tau)B^3(\tau)\mathrm{d}\tau = \sigma_t\alpha_t\int_s^t \frac{-1}{\alpha_\tau\sigma_\tau}\alpha_\tau\frac{\dot{\sigma}_\tau}{\sigma_\tau}\mathrm{d}\tau = \sigma_t\alpha_t\int_s^t \frac{-\dot{\sigma}_\tau}{\sigma_\tau^2}\mathrm{d}\tau$$

$$= \sigma_t\alpha_t\int_{\lambda_s}^{\lambda_t} e^\lambda\mathrm{d}\lambda = -\alpha_t(e^{-h}-1).$$

Next,

$$\int_s^t \Phi^2(t,\tau)B^2(\tau)\mathrm{d}\tau = \alpha_t\int_s^t \frac{2}{\alpha_\tau}\alpha_\tau\dot{\sigma}_\tau\mathrm{d}\tau = -2\bar{\sigma}_t(e^h-1),$$

and finally, as already shown in Propositions 3.1 and 4.2:

$$\int_s^t \Phi^1(t,\tau)B^1(\tau)\mathrm{d}\tau = \sigma_t^2\alpha_t\int_s^t \frac{-2\dot{\sigma}_\tau}{\sigma_\tau^3}\mathrm{d}\tau = \sigma_t^2\alpha_t\int_s^t -2\frac{\mathrm{d}\lambda_\tau}{\mathrm{d}\tau}e^{2\lambda_\tau}\mathrm{d}\tau$$

$$= \sigma_t^2\alpha_t\int_{\lambda_s}^{\lambda_t} e^{2\lambda}\mathrm{d}\lambda = -\alpha_t(e^{-2h}-1).$$

Now, since $g^2(t) = 2\alpha_t^2 \dot{\sigma}_t \sigma_t$, the stochastic integrals $\int_s^t \Phi^i(t,\tau) g(\tau) \mathrm{d}\bar{\omega}_\tau$, for $i = 1, 2$, have zero mean and variances:

$$\int_s^t (\Phi^1(t,\tau))^2 g^2(\tau) \mathrm{d}\tau = \sigma_t^4 \alpha_t^2 \int_s^t \frac{1}{\sigma_\tau^4 \alpha_\tau^2} g^2(\tau) \mathrm{d}\tau = \bar{\sigma}_t^2 (1 - e^{-2h})$$

$$\int_s^t (\Phi^2(t,\tau))^2 g^2(\tau) \mathrm{d}\tau = \alpha_t^2 \int_s^t 2\dot{\sigma}_\tau \sigma_\tau \mathrm{d}\tau = -\bar{\sigma}_t^2 (e^{2h} - 1).$$

We deduce from this the SEEDS-1 schemes in all four modes as given by iterates:

$$\widetilde{\mathbf{x}}_t = \frac{\sigma_t^2 \alpha_t}{\sigma_s^2 \alpha_s} \widetilde{\mathbf{x}}_s - \alpha_t (e^{-2h} - 1) D_\theta(\widetilde{\mathbf{x}}_s, s) + \bar{\sigma}_t \sqrt{1 - e^{-2h}} \epsilon \quad \epsilon \sim \mathcal{N}(\mathbf{0}, \mathbf{I}_d) \qquad (28)$$

$$\widetilde{\mathbf{x}}_t = \frac{\alpha_t}{\alpha_s} \widetilde{\mathbf{x}}_s - 2\bar{\sigma}_t (e^h - 1) \boldsymbol{\epsilon}_\theta(\widetilde{\mathbf{x}}_s, s) - \bar{\sigma}_t \sqrt{e^{2h} - 1} \epsilon \quad \epsilon \sim \mathcal{N}(\mathbf{0}, \mathbf{I}_d) \qquad (29)$$

$$\widetilde{\mathbf{x}}_t = \frac{\bar{\sigma}_t}{\bar{\sigma}_s} \widetilde{\mathbf{x}}_s - \alpha_t (e^{-h} - 1) D_\theta(\widetilde{\mathbf{x}}_s, s) \qquad (30)$$

$$\widetilde{\mathbf{x}}_t = \frac{\alpha_t}{\alpha_s} \widetilde{\mathbf{x}}_s - \bar{\sigma}_t (e^h - 1) \boldsymbol{\epsilon}_\theta(\widetilde{\mathbf{x}}_s, s). \qquad (31)$$

Notice that the iterates (30) and (31) are exactly the iterates of the first stage solvers in [22] and [23] with $F_\theta(\mathbf{x}_t; (M-1)t) = \boldsymbol{\epsilon}_\theta(\mathbf{x}_t, t)$. The iterates (29) coincide with the SEEDS-1 method presented in (14) and (28) consist on our SEEDS-1 method in the *data prediction mode*, which we will use in the following section.

### B.2.5   Proof of Proposition 4.5

We will write in **bold** the statement to be proven.

**If we set $g = 0$ in (15), the resulting SEEDS solvers do not yield DPM-Solver.** Indeed, if we set $g = 0$ in (15), then the method (29) does not contain a noise contribution and we readily see that it cannot be equal to (31). As the latter has been shown to be DPM-Solver-1, the conclusion follows.

**If we parameterize (15) in terms of the data prediction model $D_\theta$, the resulting SEEDS solvers are not equivalent to their noise prediction counterparts defined in Alg. 1 to 4.**

One can check that the SEEDS solver are not the same between the noise and data prediction modes by simply noticing that the noise contributions in (28) and (29) do not equate.

**The gDDIM solver [41] Theorem 1, for $\ell = 1$, is equal to SEEDS-1 in the data prediction mode.**

As shown in (29), our proposed method SEEDS-1 in the data prediction mode for the VP case has iterates of the form

$$\widetilde{\mathbf{x}}_t = \frac{\sigma_t^2 \alpha_t}{\sigma_s^2 \alpha_s} \widetilde{\mathbf{x}}_s - \alpha_t (e^{-2h} - 1) D_\theta(\widetilde{\mathbf{x}}_s, s) + \bar{\sigma}_t \sqrt{1 - e^{-2h}} \epsilon, \qquad (32)$$

where $\epsilon \sim \mathcal{N}(\mathbf{0}, \mathbf{I}_d)$, $\bar{\sigma}_t = \alpha_t \sigma_t$ and $h = \log \frac{\sigma_s}{\sigma_t}$. As our notation and that of [41, Theorem 1] overlap, we will use blue color when referring to their notation.

On the one hand, gDDIM constructs iterates over a representation of the exact solution of the following family of neural differential equations:

$$\mathrm{d}\mathbf{u}_t = \left[ f(t)\mathbf{u}_t + \frac{1 + \lambda^2}{2} \frac{g^2(t)}{\sqrt{1 - \alpha_t}} \epsilon_\theta(\mathbf{u}_t; t) \right] \mathrm{d}t + \lambda g(t) \mathrm{d}\bar{\omega}_t, \qquad (33)$$

where $\alpha_t$ decreases from $\alpha_0 = 1$ to $\alpha_T = 0$, and with coefficients

$$f(t) := \frac{1}{2} \frac{\mathrm{d}\log \alpha_t}{\mathrm{d}t}, \quad g(t) = \sqrt{-\frac{\mathrm{d}\log \alpha_t}{\mathrm{d}t}}.$$

In particular, they choose an approximation, for $\tau \in [t - \Delta t, t]$, given by

$$\boldsymbol{s}_\theta(\mathbf{u}, \tau) = \frac{\epsilon_\theta(\mathbf{u}_\tau, \tau)}{\sqrt{1 - \alpha_t}} \approx \frac{1 - \alpha_t}{1 - \alpha_\tau} \sqrt{\frac{\alpha_\tau}{\alpha_t}} \boldsymbol{s}_\theta(\mathbf{u}(t), t) - \frac{1}{1 - \alpha_\tau} \left( \mathbf{u} - \sqrt{\frac{\alpha_\tau}{\alpha_t}} \mathbf{u}(t) \right).$$

The gDDIM iterates, for $\lambda = 1 = \ell$, are then written as follows:

$$\mathbf{u}(t - \Delta t) = \sqrt{\frac{\alpha_{t-\Delta t}}{\alpha_t}}\mathbf{u}(t) + \left[-\sqrt{\frac{\alpha_{t-\Delta t}}{\alpha_t}}\sqrt{1-\alpha_t} + \sqrt{1-\alpha_{t-\Delta t} - \sigma_t^2}\right]\epsilon_\theta(\mathbf{u}(t), t) + \sigma_t\epsilon,$$

(34)

where $\epsilon \sim \mathcal{N}(\mathbf{0}, \mathbf{I}_d)$ and

$$\sigma_t^2 = (1 - \alpha_{t-\Delta t})\left[1 - \left(\frac{1-\alpha_{t-\Delta t}}{1-\alpha_t}\right)\left(\frac{\alpha_t}{\alpha_{t-\Delta t}}\right)\right].$$

(35)

Now set $(s, t) \leftarrow (t, t - \Delta t)$. Then

$$\begin{aligned}
\sigma_s^2 &= (1 - \alpha_t)\left[1 - \left(\frac{1-\alpha_t}{1-\alpha_s}\right)\left(\frac{\alpha_s}{\alpha_t}\right)\right], \\
\mathbf{u}(t) &= \sqrt{\frac{\alpha_t}{\alpha_s}}\mathbf{u}(s) + \left[\sqrt{1-\alpha_t-\sigma_s^2} - \sqrt{\frac{\alpha_t}{\alpha_s}}\sqrt{1-\alpha_s}\right]\epsilon_\theta(\mathbf{u}(s), s) + \sigma_s\epsilon.
\end{aligned}$$

Next, we identify $\alpha_t = \sqrt{\alpha_t}$, and $\bar{\sigma}_t = \sqrt{1-\alpha_t}$. Then the variance of the noise in (35) is

$$\begin{aligned}
\sigma_s^2 &= \bar{\sigma}_t^2\left[1 - \left(\frac{\bar{\sigma}_t}{\bar{\sigma}_s}\right)^2\left(\frac{\alpha_s}{\alpha_t}\right)^2\right] = \bar{\sigma}_t^2\left[1 - \left(\frac{\alpha_t\sigma_t}{\alpha_s\sigma_s}\right)^2\left(\frac{\alpha_s}{\alpha_t}\right)^2\right] \\
&= \bar{\sigma}_t^2\left[1 - \left(\frac{\sigma_t}{\sigma_s}\right)^2\right] \\
&= \bar{\sigma}_t^2(1 - e^{-2h}).
\end{aligned}$$

Hence, by denoting $\widetilde{\mathbf{x}}_t = \mathbf{u}(t)$, the gDDIM iterate (34) reads, for $\epsilon \sim \mathcal{N}(\mathbf{0}, \mathbf{I}_d)$:

$$\begin{aligned}
\widetilde{\mathbf{x}}_t &= \frac{\alpha_t}{\alpha_s}\widetilde{\mathbf{x}}_s + \left[\sqrt{\bar{\sigma}_t^2 - \bar{\sigma}_t^2(1-e^{-2h})} - \frac{\alpha_t}{\alpha_s}\bar{\sigma}_s\right]\epsilon_\theta(\widetilde{\mathbf{x}}_s, s) + \sqrt{\bar{\sigma}_t^2(1-e^{-2h})}\epsilon \\
&= \frac{\alpha_t}{\alpha_s}\widetilde{\mathbf{x}}_s + \left[\bar{\sigma}_t\sqrt{e^{-2h}} - \alpha_t\sigma_s\right]\epsilon_\theta(\widetilde{\mathbf{x}}_s, s) + \bar{\sigma}_t\sqrt{1-e^{-2h}}\epsilon \\
&= \frac{\alpha_t}{\alpha_s}\widetilde{\mathbf{x}}_s + \bar{\sigma}_t\left[\frac{\sigma_t}{\sigma_s} - \frac{\sigma_s}{\sigma_t}\right]\epsilon_\theta(\widetilde{\mathbf{x}}_s, s) + \bar{\sigma}_t\sqrt{1-e^{-2h}}\epsilon.
\end{aligned}$$

(36)

On the other hand, the data and noise prediction models in our case are related by the following equation:

$$D_\theta(\widetilde{\mathbf{x}}_s, s) = c_1(s)\frac{\widetilde{\mathbf{x}}_s}{\alpha_s} + c_2(s)\boldsymbol{\epsilon}_\theta(\widetilde{\mathbf{x}}_s, s) = \frac{\widetilde{\mathbf{x}}_s}{\alpha_s} - \sigma_s\boldsymbol{\epsilon}_\theta(\widetilde{\mathbf{x}}_s, s).$$

As such, and as $h = \lambda_t - \lambda_s = \log\frac{\sigma_s}{\sigma_t}$, one can rewrite (32) in terms of $\epsilon_\theta$ as follows:

$$\begin{aligned}
\widetilde{\mathbf{x}}_t &= \frac{\sigma_t^2\alpha_t}{\sigma_s^2\alpha_s}\widetilde{\mathbf{x}}_s - \alpha_t(e^{-2h} - 1)D_\theta(\widetilde{\mathbf{x}}_s, s) + \bar{\sigma}_t\sqrt{1-e^{-2h}}\epsilon \\
&= \frac{\sigma_t^2\alpha_t}{\sigma_s^2\alpha_s}\widetilde{\mathbf{x}}_s - \alpha_t\left(\frac{\sigma_t^2}{\sigma_s^2} - 1\right)\left[\frac{\widetilde{\mathbf{x}}_s}{\alpha_s} - \sigma_s\boldsymbol{\epsilon}_\theta(\widetilde{\mathbf{x}}_s, s)\right] + \bar{\sigma}_t\sqrt{1-e^{-2h}}\epsilon \\
&= \left[\frac{\sigma_t^2\alpha_t}{\sigma_s^2\alpha_s} - \frac{\alpha_t}{\alpha_s}\left(\frac{\sigma_t^2}{\sigma_s^2} - 1\right)\right]\widetilde{\mathbf{x}}_s + \alpha_t\left(\frac{\sigma_t^2}{\sigma_s^2} - 1\right)\sigma_s\boldsymbol{\epsilon}_\theta(\widetilde{\mathbf{x}}_s, s) + \bar{\sigma}_t\sqrt{1-e^{-2h}}\epsilon \\
&= \frac{\alpha_t}{\alpha_s}\widetilde{\mathbf{x}}_s + \bar{\sigma}_t\left(\frac{\sigma_t^2}{\sigma_s^2} - 1\right)\frac{\sigma_s}{\sigma_t}\boldsymbol{\epsilon}_\theta(\widetilde{\mathbf{x}}_s, s) + \bar{\sigma}_t\sqrt{1-e^{-2h}}\epsilon \\
&= \frac{\alpha_t}{\alpha_s}\widetilde{\mathbf{x}}_s + \bar{\sigma}_t\left(\frac{\sigma_t}{\sigma_s} - \frac{\sigma_s}{\sigma_t}\right)\boldsymbol{\epsilon}_\theta(\widetilde{\mathbf{x}}_s, s) + \bar{\sigma}_t\sqrt{1-e^{-2h}}\epsilon,
\end{aligned}$$

which coincides with the gDDIM iterate in Equation (36).

### B.2.6 The VE, DDIM and iDDPM cases

Following [16, Eq. 217], here $\alpha_t = 1$ and $c_1(t) = 1$ and so the only possibilities incurring into semi-linear differential equations are

$$A^1(t) = 2\frac{\dot\sigma_t}{\sigma_t} \qquad\qquad B^1(t) = -2\frac{\dot\sigma_t}{\sigma_t} \qquad\text{(DP SDE)}$$

$$A^3(t) = \frac{\dot\sigma_t}{\sigma_t} \qquad\qquad B^3(t) = -\frac{\dot\sigma_t}{\sigma_t}. \qquad\text{(DP PFO)}$$

We obtain, again with the choice $\lambda_t = -\log(\sigma_t)$ and setting $h = \lambda_t - \lambda_s$, the following:

$$\Phi^1(t,s) = \frac{\sigma_t^2}{\sigma_s^2}, \qquad \Phi^3(t,s) = \frac{\sigma_t}{\sigma_s},$$

$$\int_s^t \Phi^1(t,\tau)B^1(\tau)\mathrm{d}\tau = -\sigma_t^2\int_{\lambda_s}^{\lambda_t} e^{2\lambda}\mathrm{d}\lambda = \frac{1}{2}(e^{-2h}-1)$$

$$\int_s^t \Phi^3(t,\tau)B^3(\tau)\mathrm{d}\tau = -\sigma_t\int_{\lambda_s}^{\lambda_t} e^{\lambda}\mathrm{d}\lambda = e^{-h}-1.$$

Next,

$$\int_s^t (\Phi^1(t,\tau))^2 g^2(\tau)\mathrm{d}\tau = \sigma_t^4\int_s^t \frac{1}{\sigma_\tau^4}\sigma_\tau^2 2\frac{\dot\sigma_\tau}{\sigma_\tau}\mathrm{d}\tau = \sigma_t^4\int_s^t \frac{2\dot\sigma_\tau}{\sigma_\tau^3}\mathrm{d}\tau = \sigma_t^2(e^{-2h}-1).$$

We readily see that these cases are identical to the VP case with $\alpha_t = 1$. In particular, the obtained SEEDS-1 iterates are

$$\widetilde{\mathbf{x}}_t = \frac{\sigma_t^2}{\sigma_s^2}\widetilde{\mathbf{x}}_s - (e^{-2h}-1)D_\theta(\widetilde{\mathbf{x}}_s;\sigma_s) + \sigma_t\sqrt{1-e^{-2h}}\epsilon.$$

$$\widetilde{\mathbf{x}}_t = \frac{\sigma_t}{\sigma_s}\widetilde{\mathbf{x}}_s - (e^{-h}-1)D_\theta(\widetilde{\mathbf{x}}_s;\sigma_s).$$

Now denote $s_1 = t_\lambda(\lambda_s + rh)$, for $0 < r \leqslant 1$, where $t_\lambda = e^{-\lambda}$. There are two families of single-step one-parameter two-stage exponential ODE schemes:

$$\widetilde{\mathbf{x}}_t = \frac{\sigma_t}{\sigma_s}\widetilde{\mathbf{x}}_s - (e^{-h}-1)\left[\left(1-\frac{1}{2r}\right)D_\theta(\widetilde{\mathbf{x}}_s;\sigma_s) + \frac{1}{2r}D_\theta(\widetilde{\mathbf{x}}_1;\sigma_{s_1})\right]$$

$$\widetilde{\mathbf{x}}_t = \frac{\sigma_t}{\sigma_s}\widetilde{\mathbf{x}}_s - (e^{-h}-1)D_\theta(\widetilde{\mathbf{x}}_s;\sigma_s) + \frac{1}{r}\left(\frac{e^{-h}-1}{h}+1\right)[D_\theta(\widetilde{\mathbf{x}}_1;\sigma_{s_1}) - D_\theta(\widetilde{\mathbf{x}}_s;\sigma_s)],$$

with same supporting value

$$\widetilde{\mathbf{x}}_1 = \frac{\sigma_{s_1}}{\sigma_s}\widetilde{\mathbf{x}}_s - (e^{-rh}-1)D_\theta(\widetilde{\mathbf{x}}_s;\sigma_s).$$

In the same vein we define a single-step two stage exponential SDE scheme:

$$\widetilde{\mathbf{x}}_1 = \frac{\sigma_{s_1}^2}{\sigma_s^2}\widetilde{\mathbf{x}}_s - (e^{-2rh}-1)D_\theta(\widetilde{\mathbf{x}}_s;\sigma_s) + \sigma_{s_1}\sqrt{e^{-2rh}-1}\epsilon_1$$

$$\widetilde{\mathbf{x}}_t = \frac{\sigma_t^2}{\sigma_s^2}\widetilde{\mathbf{x}}_s - (e^{-2h}-1)\left[\left(1-\frac{1}{2r}\right)D_\theta(\widetilde{\mathbf{x}}_s;\sigma_s) + \frac{1}{2r}D_\theta(\widetilde{\mathbf{x}}_1;\sigma_{s_1})\right]$$
$$+ \sigma_t\left[\sqrt{e^{-2h}-e^{-2rh}}\epsilon_1 + \sqrt{e^{-2rh}-1}\epsilon_2\right].$$

### B.2.7 The EDM case

In the EDM-preconditioned case [16, App. B.6], we set $\sigma_t := t$ and $\alpha_t := 1$. We denote $\sigma_d := \sigma_{\text{data}}$ the variance of the considered initial dataset and we set

$$c_1(t) = \frac{\sigma_d^2}{t^2+\sigma_d^2}, \quad c_2(t) = \frac{t\sigma_d}{\sqrt{t^2+\sigma_d^2}}, \quad c_3(t) = \frac{1}{\sqrt{t^2+\sigma_d^2}}, \quad c_4(t) = \frac{1}{4}\log(t),$$

so we obtain the following coefficients:

$$A^1(t) = \frac{2}{t} \qquad\qquad\qquad B^1(t) = -\frac{2}{t} \qquad\qquad \text{(DP NRSDE)}$$

$$A^2(t) = \frac{2}{t}\left(1 - \frac{\sigma_d^2}{t^2 + \sigma_d^2}\right) \qquad\qquad B^2(t) = \frac{-2\sigma_d}{\sqrt{t^2 + \sigma_d^2}} \qquad\qquad \text{(NP NRSDE)}$$

$$A^3(t) = \frac{1}{t} \qquad\qquad\qquad B^3(t) = -\frac{1}{t} \qquad\qquad \text{(DP NPFO)}$$

$$A^4(t) = \frac{1}{t}\left(1 - \frac{\sigma_d^2}{t^2 + \sigma_d^2}\right) \qquad\qquad B^4(t) = \frac{-\sigma_d}{\sqrt{t^2 + \sigma_d^2}} \qquad\qquad \text{(NP NPFO)}$$

In particular, the data prediction neural SDE/PFO are identical to those in the VE case with $\sigma_t = t$. So let us concentrate on the noise prediction regime, leading us to prove Proposition 3.2.

### B.2.8 Proof of Proposition 3.2

In the Noise Prediction case, we have

$$\Phi^2(t, s) = \frac{t^2 + \sigma_d^2}{s^2 + \sigma_d^2}, \quad \Phi^4(t, s) = \sqrt{\frac{t^2 + \sigma_d^2}{s^2 + \sigma_d^2}},$$

so we readily compute:

$$\int_s^t \Phi^2(t, \tau) B^2(\tau) \mathrm{d}\tau = (t^2 + \sigma_d^2) \int_s^t \frac{1}{\tau^2 + \sigma_d^2} \cdot \frac{-2\sigma_d}{\sqrt{\tau^2 + \sigma_d^2}} \mathrm{d}\tau,$$

$$\int_s^t \Phi^4(t, \tau) B^4(\tau) \mathrm{d}\tau = \sqrt{t^2 + \sigma_d^2} \int_s^t \frac{\sigma_d}{\sqrt{\tau^2 + \sigma_d^2}} \cdot \frac{-1}{\sqrt{\tau^2 + \sigma_d^2}} \mathrm{d}\tau.$$

Let us consider two different changes of variables:

$$\lambda_t := -\log\left(\arctan\left(\frac{t}{\sigma_d}\right)\right) \qquad \text{and} \qquad \lambda_t := -\log\left(\frac{t}{\sigma_d \sqrt{t^2 + \sigma_d^2}}\right), \qquad (37)$$

that will be used for the (NP NPFO) and (NP NRSDE), respectively. For the former case, we have

$$e^{-\lambda_t} \mathrm{d}\lambda_t = -\frac{\frac{1}{\sigma_d} \mathrm{d}t}{1 + \frac{t^2}{\sigma_d^2}} = -\frac{\sigma_d \mathrm{d}t}{\sigma_d^2 + t^2}.$$

Therefore, we can deduce that

$$\int_s^t \Phi^4(t; \tau) B^4(\tau) \mathrm{d}\tau = \int_s^t \sqrt{\frac{t^2 + \sigma_d^2}{\tau^2 + \sigma_d^2}} \cdot \frac{-\sigma_d}{\sqrt{\tau^2 + \sigma_d^2}} \mathrm{d}\tau$$

$$= \sqrt{t^2 + \sigma_d^2} \int_{\lambda_s}^{\lambda_t} \frac{-\sigma_d}{\tau^2 + \sigma_d^2} \mathrm{d}\tau$$

$$= \sqrt{t^2 + \sigma_d^2} \int_{\lambda_s}^{\lambda_t} e^{-\lambda} \mathrm{d}\lambda$$

$$= \sqrt{t^2 + \sigma_d^2} \arctan\left(\frac{t}{\sigma_d}\right) (e^h - 1).$$

For the latter case, we have

$$e^{-\lambda_t} \mathrm{d}\lambda_t = -\frac{\sigma_d \sqrt{t^2 + \sigma_d^2} - t\sigma_d \frac{t}{\sqrt{t^2 + \sigma_d^2}}}{\sigma_d^2 (t^2 + \sigma_d^2)}$$

$$= -\frac{\sigma_d^2}{\sigma_d (t^2 + \sigma_d^2) \sqrt{t^2 + \sigma_d^2}} = -\frac{\sigma_d}{(t^2 + \sigma_d^2) \sqrt{t^2 + \sigma_d^2}}.$$

We then obtain

$$\int_s^t \Phi^2(t;\tau)B^2(\tau)\mathrm{d}\tau = \int_s^t \frac{t^2+\sigma_d^2}{\tau^2+\sigma_d^2} \cdot \frac{-2\sigma_d}{\sqrt{\tau^2+\sigma_d^2}}\mathrm{d}\tau$$

$$= 2(t^2+\sigma_d^2)\int_s^t e^{-\lambda}\mathrm{d}\lambda$$

$$= \frac{2t\sqrt{t^2+\sigma_d^2}}{\sigma_d}(e^h-1).$$

The stochastic integral $\int_s^t \Phi^2(t,\tau)g(\tau)\mathrm{d}\bar{\boldsymbol{\omega}}_\tau$ in noise prediction case is a Gaussian random variable with zero mean and whose variance can be computed by Itô isometry as

$$\int_t^s [\Phi^2(t;\tau)g^2(\tau)]^2\mathrm{d}\tau = (t^2+\sigma_d^2)^2 \int_t^s \frac{1}{[\tau^2+\sigma_d^2]^2}2\tau\mathrm{d}\tau$$

$$= (t^2+\sigma_d^2)^2 \int_t^s \frac{1}{[\tau^2+\sigma_d^2]^2}\mathrm{d}\tau^2$$

$$= (t^2+\sigma_d^2)^2 \left(-\frac{1}{s^2+\sigma_d^2} + \frac{1}{t^2+\sigma_d^2}\right)$$

$$= \frac{(t^2+\sigma_d^2)(s^2-t^2)}{(s^2+\sigma_d^2)}.$$

Putting everything together, we obtain the analytic solution at time $t$ of (2) with coefficients (11) and initial value $\mathbf{x}_s$ for the (NP NRSDE):

$$\mathbf{x}_t = \frac{t^2+\sigma_d^2}{s^2+\sigma_d^2}\mathbf{x}_s + 2(t^2+\sigma_d^2)\int_{\lambda_s}^{\lambda_t} e^{-\lambda}\hat{F}_\theta(\hat{\mathbf{x}}_\lambda,\lambda)\mathrm{d}\lambda - \sqrt{2}(t^2+\sigma_d^2)\int_{\lambda_s}^{\lambda_t} e^{-\lambda}\mathrm{d}\bar{\boldsymbol{\omega}}_\lambda, \quad (38)$$

where $\lambda_t := -\log\left[\frac{t}{\sigma_d\sqrt{t^2+\sigma_d^2}}\right]$. For the (NP NPFO), it is given by

$$\mathbf{x}_t = \sqrt{\frac{t^2+\sigma_d^2}{s^2+\sigma_d^2}}\mathbf{x}_s + \sqrt{t^2+\sigma_d^2}\int_{\lambda_s}^{\lambda_t} e^{-\lambda}\hat{F}_\theta(\hat{\mathbf{x}}_\lambda,\lambda)\mathrm{d}\lambda, \quad \lambda_t := -\log\left[\arctan\left[\frac{t}{\sigma_d}\right]\right]. \quad (39)$$

This finishes the proof.

*Remark* B.2. From the above proof, we immediately deduce the SEEDS-1 iterates in the EDM-preconditioned noise prediction case. These are given, for the (NP NPFO) and (NP NRSDE) respectively, by

$$\widetilde{\mathbf{x}}_t = \sqrt{\frac{t^2+\sigma_d^2}{s^2+\sigma_d^2}}\widetilde{\mathbf{x}}_s + \sqrt{t^2+\sigma_d^2}\arctan\left(\frac{t}{\sigma_d}\right)(e^h-1)\epsilon_\theta(\widetilde{\mathbf{x}}_s,s), \quad (40)$$

$$\widetilde{\mathbf{x}}_t = \frac{t^2+\sigma_d^2}{s^2+\sigma_d^2}\widetilde{\mathbf{x}}_s + \frac{2t\sqrt{t^2+\sigma_d^2}}{\sigma_d}(e^h-1)\epsilon_\theta(\widetilde{\mathbf{x}}_s,s) + \sqrt{\frac{(t^2+\sigma_d^2)(s^2-t^2)}{(s^2+\sigma_d^2)}}\epsilon, \quad (41)$$

where $\epsilon \sim \mathcal{N}(\mathbf{0},\mathbf{I}_d)$.

### B.2.9 The SEEDS algorithms in the EDM Noise Prediction case

In the same way we presented the SEEDS algorithms 3 and 4 for the VP case, the change-of-variables optimized for the EDM framework in Proposition 3.2 induces algorithms 5 for SEEDS-2 and 6 for SEEDS-3 respectively, under the Noise Prediction regime. This is the version of SEEDS that we use to achieve same performance as EDM solver but with twice less NFEs than the latter for ImageNet-64 (see Table 1 and Fig. 1 (c) in the main part of this article).

---

**Algorithm 5** SEEDS-2 (EDM NP)

**Input:** initial value $\mathbf{x}_T$, steps $\{t_i\}_{i=0}^M$, data prediction model $D_\theta$, $r = 1/2$
Initialize $\tilde{\mathbf{x}}_{t_0} \leftarrow \mathbf{x}_T$
**for** $i = 1$ **to** $M - 1$ **do**
$\quad (t, s) \leftarrow (t_i, t_{i-1}), \quad (z^1, z^2) \leftarrow \mathcal{N}(0, \text{Id})^{\otimes 2}$
$\quad \lambda(t) \leftarrow -\log\left(\frac{t}{\sigma_d \sqrt{t^2 + \sigma_d^2}}\right) \qquad t_\lambda(\lambda) \leftarrow \frac{\sigma_d}{\sqrt{\frac{1}{\sigma_d^2 e^{-2\lambda}} - 1}}, \qquad \sigma_d \leftarrow \sigma_{data}$
$\quad h \leftarrow \lambda_t - \lambda_s, \quad \lambda_{s_1} \leftarrow \lambda_s + rh, \quad s_1 \leftarrow t_\lambda(\lambda_{s_1})$
$\quad \epsilon_\theta\left(\widetilde{\mathbf{x}}_s, s\right) \leftarrow \left[D_\theta\left(\frac{\tilde{\mathbf{x}}_s}{\alpha_s}; \sigma_s\right) - \frac{\sigma_d^2}{s^2 + \sigma_d^2}\frac{\tilde{\mathbf{x}}_s}{\alpha_s}\right] \cdot \frac{\sqrt{s^2 + \sigma_d^2}}{s \sigma_d},$
$\quad \widetilde{\mathbf{x}}_{s_1} \leftarrow \frac{s_1^2 + \sigma_d^2}{s^2 + \sigma_d^2}\widetilde{\mathbf{x}}_s + 2\frac{s_1\sqrt{s_1^2 + \sigma_d^2}}{\sigma_d}(e^{rh} - 1)\epsilon_\theta\left(\widetilde{\mathbf{x}}_s, s\right) + \sqrt{\frac{(s_1^2 + \sigma_d^2)(s^2 - s_1^2)}{(s^2 + \sigma_d^2)}} z^1$
$\quad \widetilde{\mathbf{x}}_t \leftarrow \frac{t^2 + \sigma_d^2}{s^2 + \sigma_d^2}\widetilde{\mathbf{x}}_s + 2\frac{t\sqrt{t^2 + \sigma_d^2}}{\sigma_d}(e^h - 1)\epsilon_\theta\left(\widetilde{\mathbf{x}}_{s_1}, s_1\right) + \sqrt{\frac{(t^2 + \sigma_d^2)(s_1^2 - t^2)}{(s_1^2 + \sigma_d^2)}}(e^{rh}z^1 + z^2)$
**end for**
Return $\tilde{\mathbf{x}}_{t_M} \leftarrow \text{last-step}(\tilde{\mathbf{x}}_{t_{M-1}}, t_{M-1}, t_M)$

---

**Algorithm 6** SEEDS-3 (EDM NP)

**Input:** initial value $\mathbf{x}_T$, steps $\{t_i\}_{i=0}^M$, data prediction model $D_\theta$, $r_1 = 1/3, r_2 = 2/3$
Initialize $\tilde{\mathbf{x}}_{t_0} \leftarrow \mathbf{x}_T$
**for** $i = 1$ **to** $M - 1$ **do**
$\quad (t, s) \leftarrow (t_i, t_{i-1}), \quad (z^1, z^2, z^3) \leftarrow \mathcal{N}(0, \text{Id})^{\otimes 3}$
$\quad \lambda(t) \leftarrow -\log\left(\frac{t}{\sigma_d \sqrt{t^2 + \sigma_d^2}}\right) \qquad t_\lambda(\lambda) \leftarrow \frac{\sigma_d}{\sqrt{\frac{1}{\sigma_d^2 e^{-2\lambda}} - 1}}, \qquad \sigma_d \leftarrow \sigma_{data}$
$\quad h \leftarrow \lambda_t - \lambda_s, \quad \lambda_{s_1} \leftarrow \lambda_s + r_1 h, \quad \lambda_{s_2} \leftarrow \lambda_s + r_2 h, \quad s_1 \leftarrow t_\lambda(\lambda_{s_1}), \quad s_2 \leftarrow t_\lambda(\lambda_{s_2})$
$\quad \epsilon_\theta\left(\widetilde{\mathbf{x}}_s, s\right) \leftarrow \left[D_\theta\left(\frac{\tilde{\mathbf{x}}_s}{\alpha_s}; \sigma_s\right) - \frac{\sigma_d^2}{s^2 + \sigma_d^2}\frac{\tilde{\mathbf{x}}_s}{\alpha_s}\right] \cdot \frac{\sqrt{s^2 + \sigma_d^2}}{s \sigma_d},$
$\quad \widetilde{\mathbf{x}}_{s_1} \leftarrow \frac{s_1^2 + \sigma_d^2}{s^2 + \sigma_d^2}\widetilde{\mathbf{x}}_s + 2\frac{s_1\sqrt{s_1^2 + \sigma_d^2}}{\sigma_d}(e^{r_1 h} - 1)\epsilon_\theta\left(\widetilde{\mathbf{x}}_s, s\right) + \sqrt{\frac{(s_1^2 + \sigma_d^2)(s^2 - s_1^2)}{(s^2 + \sigma_d^2)}} z^1$
$\quad \boldsymbol{A} \leftarrow \sqrt{\frac{(s_2^2 + \sigma_d^2)(s_1^2 - s_2^2)}{(s_1^2 + \sigma_d^2)}}(e^{r_1 h}z^1 + z^2), \quad P_1 \leftarrow \epsilon_\theta\left(\widetilde{\mathbf{x}}_{s_1}, s_1\right) - \epsilon_\theta\left(\mathbf{x}_s, s\right)$
$\quad \widetilde{\mathbf{x}}_{s_2} \leftarrow \frac{s_2^2 + \sigma_d^2}{s^2 + \sigma_d^2}\widetilde{\mathbf{x}}_s + 2\frac{s_2\sqrt{s_2^2 + \sigma_d^2}}{\sigma_d}(e^{r_2 h} - 1)\epsilon_\theta\left(\widetilde{\mathbf{x}}_s, s\right) + \frac{2r_2}{r_1}\frac{s_2\sqrt{s_2^2 + \sigma_d^2}}{\sigma_d}\left(\frac{e^{r_2 h} - 1}{r_2 h} - 1\right)P_1 + \boldsymbol{A}$
$\quad \boldsymbol{B} \leftarrow \sqrt{\frac{(t^2 + \sigma_d^2)(s_2^2 - t^2)}{(s_2^2 + \sigma_d^2)}}\left(e^{r_2 h}z^1 + e^{r_1 h}z^2 + z^3\right), \quad P_2 \leftarrow \epsilon_\theta\left(\widetilde{\mathbf{x}}_{s_2}, s_2\right) - \epsilon_\theta\left(\mathbf{x}_s, s\right)$
$\quad \widetilde{\mathbf{x}}_t \leftarrow \frac{t^2 + \sigma_d^2}{s^2 + \sigma_d^2}\widetilde{\mathbf{x}}_s + 2\frac{t\sqrt{t^2 + \sigma_d^2}}{\sigma_d}(e^h - 1)\epsilon_\theta\left(\widetilde{\mathbf{x}}_s, s\right) + \frac{2}{r_2}\frac{t\sqrt{t^2 + \sigma_d^2}}{\sigma_d}\left(\frac{e^h - 1}{h} - 1\right)P_2 + \boldsymbol{B}$
**end for**
Return $\tilde{\mathbf{x}}_{t_M} \leftarrow \text{last-step}(\tilde{\mathbf{x}}_{t_{M-1}}, t_{M-1}, t_M)$

---

## C   Convergence Proofs

In this Section, we give detailed proofs of Theorem 4.1, Proposition 4.3 and Corollary 4.4 stated in the main part of this paper. Let us start recalling its framework. We start by considering the NP NRSDE (15) with VP coefficients:

$$
\begin{aligned}
\mathrm{d}\mathbf{x}_t &= \left[f(t)\mathbf{x}_t + \frac{g^2(t)}{\bar{\sigma}_t}\boldsymbol{\epsilon}_\theta(\mathbf{x}_t, t)\right]\mathrm{d}t + g(t)\mathrm{d}\bar{\boldsymbol{\omega}}_t \\
&= \left[f(t)\mathbf{x}_t + \frac{2\alpha_t^2 \dot{\sigma}_t \sigma_t}{\bar{\sigma}_t}\boldsymbol{\epsilon}_\theta(\mathbf{x}_t, t)\right]\mathrm{d}t + g(t)\mathrm{d}\bar{\boldsymbol{\omega}}_t \\
&= \left[\frac{\mathrm{d}\log\alpha_t}{\mathrm{d}t}\mathbf{x}_t + 2\alpha_t \dot{\sigma}_t \boldsymbol{\epsilon}_\theta(\mathbf{x}_t, t)\right]\mathrm{d}t + \alpha_t \sqrt{\frac{\mathrm{d}[\sigma_t^2]}{\mathrm{d}t}}\mathrm{d}\bar{\boldsymbol{\omega}}_t.
\end{aligned}
\tag{42}
$$

Denote $t_\lambda$ the inverse of $\lambda_t := -\log(\sigma_t)$ (which is a strictly decreasing function of $t$) and denote $\hat{\mathbf{x}}_\lambda := \mathbf{x}(t_\lambda(\lambda)), \hat{\boldsymbol{\epsilon}}_\theta(\hat{\mathbf{x}}_\lambda, \lambda) := \boldsymbol{\epsilon}_\theta(\mathbf{x}(t_\lambda(\lambda)), t_\lambda(\lambda))$. We consider a time discretization $\{t_i\}_{i=0}^{M+1}$ going backwards in time starting from $t_0 = T$ to $t_{M+1} = 0$ and to ease the notation we will always

denote $t < s$ for two consecutive time-steps $t_i < t_{i-1}$. The analytic solution at time $t$ of the RSDE (2) with coefficients (7) and initial value $\mathbf{x}_s$ reads

$$\mathbf{x}_t = \frac{\alpha_t}{\alpha_s}\mathbf{x}_s - 2\alpha_t \int_{\lambda_s}^{\lambda_t} e^{-\lambda}\hat{\boldsymbol{\epsilon}}_\theta(\widehat{\mathbf{x}}_\lambda, \lambda)\mathrm{d}\lambda - \sqrt{2}\alpha_t \int_{\lambda_s}^{\lambda_t} e^{-\lambda}\mathrm{d}\bar{\boldsymbol{\omega}}(\lambda). \tag{43}$$

Given an initial condition $\widetilde{\mathbf{x}}_{t_0} = \mathbf{x}_T$, the SEEDS-1 iterates read, for $h_i = \lambda_{t_i} - \lambda_{t_{i-1}} = \lambda_t - \lambda_s$,

$$\widetilde{\mathbf{x}}_t = \frac{\alpha_t}{\alpha_s}\widetilde{\mathbf{x}}_s - 2\bar{\sigma}_t(e^{h_i} - 1)\boldsymbol{\epsilon}_\theta(\widetilde{\mathbf{x}}_s, s) - \bar{\sigma}_t\sqrt{e^{2h_i} - 1}\boldsymbol{\epsilon} \quad \boldsymbol{\epsilon} \sim \mathcal{N}(\mathbf{0}, \mathbf{I}_d).$$

**Assumption C.1.**

1. The function $\boldsymbol{\epsilon}_\theta(\mathbf{x}, t)$ is continuous (and hence bounded) on $[0, T]$, Lipschitz with respect to $\mathbf{x}$ and there is a constant $C$ such that, for $t, s \in [0, T]$ with $t < s$, we have

$$|\boldsymbol{\epsilon}_\theta(\mathbf{x}, t) - \boldsymbol{\epsilon}_\theta(\mathbf{y}, t)|^2 \leqslant L_1|\mathbf{x} - \mathbf{y}|^2 \tag{44}$$

$$|\boldsymbol{\epsilon}_\theta(\mathbf{x}, t)|^2 \vee |g(t)|^2 \leqslant L_2(1 + |\mathbf{x}|^2) \tag{45}$$

$$|\boldsymbol{\epsilon}_\theta(\mathbf{x}, t) - \boldsymbol{\epsilon}_\theta(\mathbf{x}, s)|^2 \leqslant L_3(1 + |\mathbf{x}|^2)|t - s|^2 \tag{46}$$

2. $h = \max_{1 \leq i \leq M} |h_i| \sim O(1/M)$, where $h_i = \lambda_{t_i} - \lambda_{t_{i-1}}$.

Let $\mathcal{C}_P^l(\mathbb{R}^d, \mathbb{R})$ denote the family of $l$ times continuously differentiable real-valued functions on $\mathbb{R}^d$ whose partial derivatives of order $\leqslant l$ have polynomial growth and let $\mathcal{C}_P^{k,l}(I \times \mathbb{R}^d, \mathbb{R})$ be the space of functions $g(\cdot, \cdot)$ such that, for all $(t, x) \in I \times \mathbb{R}^d$, $g(\cdot, x) \in \mathcal{C}^k(I, \mathbb{R})$ and $g(t, \cdot) \in \mathcal{C}_P^l(\mathbb{R}^d, \mathbb{R})$.

**Assumption C.2.** In addition to Assumption C.1, assume that all the components of $\boldsymbol{\epsilon}_\theta$ belong to $\mathcal{C}_P^{4,2}(\mathbb{R}^d \times [0, 1], \mathbb{R})$.

Before going into the proofs, we give some context that lead us to necessitate such assumptions.

## C.1 Preliminaries

For an interval $I = [t_0, T]$, let $\mathbf{x} = (\mathbf{x}(t))_I$ the solution of the following SDE

$$\mathrm{d}\mathbf{x}(t) = f(\mathbf{x}(t), t)\mathrm{d}t + g(t)\mathrm{d}\boldsymbol{\omega}(t), \tag{47}$$

where $g(t) = \hat{g}(t) \cdot \mathrm{Id}_d$ is considered here as a diagonal matrix with identical diagonal entries $\hat{g}(t)$. Suppose that $f, g$ are continuous, and satisfy a linear growth and Lipschitz condition so that the conditions of the Existence and Uniqueness Theorem are fulfilled for the SDE (47).

Let $I_h = \{t_0, \ldots, t_M\}$ be a time discretization of $I$ with step sizes $h_n = t_{n+1} - t_n$ for $n = 0, \ldots, M - 1$ and let $h = \max_{0 \leqslant n < M} h_n$. A time discrete approximation scheme $\widehat{\mathbf{x}} = (\widehat{\mathbf{x}}_n)_{I_h}$ will be defined as a sequence $\widehat{\mathbf{x}}_0 = \mathbf{x}(t_0)$ and

$$\widehat{\mathbf{x}}_{n+1} = \Phi(\widehat{\mathbf{x}}_n, h_n, I_n), \qquad n = 0, \ldots, M - 1,$$

where $I_0$ is independent of $\widehat{\mathbf{x}}_0$, with $I_n = \boldsymbol{\omega}(t_{n+1} - t_n)$ Wiener increments drawn from the normal distributions with zero mean and variance $h_n$ and which are independent of $\widehat{\mathbf{x}}_0, \ldots, \widehat{\mathbf{x}}_n$ and $I_0, \ldots, I_{n-1}$.

A scheme $\widehat{\mathbf{x}}$ converges in the strong (resp. weak) sense, with global order $p > 0$, to the solution $\mathbf{x}$ of the SDE (47) if there is a constant $C > 0$, independent of $h$ and $\delta > 0$, such that, for each $h \in ]0, \delta]$, we have

$$(\mathbb{E}[|\mathbf{x}(T) - \widehat{\mathbf{x}}_M|^2])^{1/2} \leq Ch^p, \quad (\text{resp.} |\mathbb{E}[G(\mathbf{x}(T))] - \mathbb{E}[G(\widehat{\mathbf{x}}_M)]| \leq Ch^p, \forall G \in \mathcal{C}_P^{2(p+1)}(\mathbb{R}^d, \mathbb{R})).$$

Notice that if $(\mathbb{E}[|\mathbf{x}(T) - \widehat{\mathbf{x}}_M|^2])^{1/2} = \mathcal{O}(h^p)$ then for every function $f$ satisfying a Lipschitz condition, we have $|\mathbb{E}[f(\mathbf{x}(T))] - \mathbb{E}[f(\widehat{\mathbf{x}}_M)]| = \mathcal{O}(h^p)$. Nevertheless, this is not enough to infer the optimal weak order of convergence of such method.

Strong convergence is concerned with the precision of the path, while the weak convergence is with the precision of the moments. As, for DPMs, the center of attention is the evolution of the probability densities rather than that of the noising process of single data samples, weak convergence is enough to guarantee the well-conditioning of our numerical schemes. Moreover, when the diffusion coefficient vanishes, then both strong and weak convergence (with the choice $G = \mathrm{id}$) reduce to the usual deterministic convergence criterion for ODEs.

Let us now state some useful results that will be used later on.

**Assumption C.3.** All moments of the initial value $\widehat{\mathbf{x}}_0$ exist, $f$ is continuous, satisfy a linear growth and globally Lipschitz condition.

In particular, since $I$ is a closed finite interval in $\mathbb{R}$, $f(\cdot, x)$ and $g$ are bounded by some constant.

**Theorem C.4** ([25])**.** *In addition to C.3, suppose that*
$$|\mathbb{E}[\mathbf{x}(t_1) - \widehat{\mathbf{x}}_1]| \leqslant Ch^{p_1}$$
$$(\mathbb{E}[|\mathbf{x}(t_1) - \widehat{\mathbf{x}}_1|^2])^{1/2} \leqslant Ch^{p_2}$$
*with $p_2 \geqslant 1/2$ and $p_1 \geqslant p_2 + 1/2$. Then $\widehat{\mathbf{x}}$ is of strong global order $p = p_2 - 1/2$.*

**Assumption C.5.** All moments of the initial value $\widehat{\mathbf{x}}_0$ exist, $f$ is continuous, satisfy a linear growth and Lipschitz condition with all their components belonging to $\mathcal{C}_P^{p+1,2(p+1)}(I \times \mathbb{R}^d, \mathbb{R})$ and $g \in \mathcal{C}^{p+1}(I, \mathbb{R})$.

**Theorem C.6** ([25])**.** *In addition to C.5, suppose that*

1. *for large enough $r \in \mathbb{N}$, the moments $\mathbb{E}[|\widehat{\mathbf{x}}_n|^{2r}]$ exist and are uniformingly bounded with respect to $M$ and $n = 0, \ldots, M$*

2. *for all $G \in \mathcal{C}_P^{2(p+1)}(\mathbb{R}^d, \mathbb{R})$, if $\widehat{\mathbf{x}}_n = \mathbf{x}(t_n)$, then for some $K \in \mathcal{C}_P^0(\mathbb{R}^d, \mathbb{R})$, we have*
$$|\mathbb{E}[G(\mathbf{x}(t_{n+1}))] - \mathbb{E}[G(\widehat{\mathbf{x}}_{n+1})]| \leq K(\widehat{\mathbf{x}}_n)h^{p+1}.$$

*Then $\widehat{\mathbf{x}}$ is of weak global order $p$.*

**Lemma C.7.** *Suppose that $\widehat{\mathbf{x}}_0$ has moments of all orders and that, for $h < 1$,*
$$|\mathbb{E}[\Phi(\widehat{\mathbf{x}}_n, h_n, I_n) - \widehat{\mathbf{x}}_n]| \leqslant K(1 + |\widehat{\mathbf{x}}_n|)h$$
$$|\Phi(\widehat{\mathbf{x}}_n, h_n, I_n) - \widehat{\mathbf{x}}_n| \leqslant X_n(1 + |\widehat{\mathbf{x}}_n|)h^{1/2}$$
*where $X_n$ has moments of all orders. Then Condition 1 in Theorem C.6 is fulfilled.*

## C.2 Convergence of SEEDS-1

In this section we will prove that that SEEDS-1 as described above is of global strong order 1.0.

### C.2.1 Strong Itô-Taylor approximation

Then the truncated Itô-Taylor expansion of the analytic solution $\mathbf{x}_t$ of the VP NP NRSDE starting from $\mathbf{x}_s$ is given, for $\epsilon \sim \mathcal{N}(0, 1)$, by

$$
\begin{aligned}
\mathbf{x}_t &= \frac{\alpha_t}{\alpha_s}\mathbf{x}_s - 2\alpha_t \int_{\lambda_s}^{\lambda_t} e^{-\lambda}\widehat{\boldsymbol{\epsilon}}_\theta(\widehat{\mathbf{x}}_\lambda, \lambda)\mathrm{d}\lambda - \sqrt{2}\alpha_t \int_{\lambda_s}^{\lambda_t} e^{-\lambda}\mathrm{d}\boldsymbol{\omega}_\lambda \\
&= \frac{\alpha_t}{\alpha_s}\mathbf{x}_s - 2\bar{\sigma}_t(e^h - 1)\boldsymbol{\epsilon}_\theta(\mathbf{x}_s, s) - \bar{\sigma}_t\sqrt{e^{2h} - 1}\epsilon + \mathcal{R}_1 \\
&= \frac{\alpha_t}{\alpha_s}\mathbf{x}_s - 2\bar{\sigma}_t(e^h - 1)\boldsymbol{\epsilon}_\theta(\mathbf{x}_s, s) - \bar{\sigma}_t\sqrt{e^{2h} - 1}\epsilon + \mathcal{O}(h),
\end{aligned}
$$

where the symbol $\mathcal{O}(h^p)$ represent terms $\mathbf{u}$ such that $\|\mathbf{u}\| \leqslant |K(\mathbf{x}_s)|h^p$, for $K \in \mathcal{C}_P^0(\mathbb{R}^d, \mathbb{R})$ and small $h > 0$. The SEEDS-1 scheme corresponds to such truncated Itô-Taylor expansion containing only the time and Wiener integrals of multiplicity one. As such, since $G_t g(t) = 0$ and assuming Lipschitz and linear growth conditions on $\epsilon_\theta$ as in Assumption C.1, $\mathbf{x}_t$ can be interpreted as an order 1.0 strong Itô-Taylor approximation [17, Theorem 10.6.3] of the solution to (42).

### C.2.2 Continuous approximation of SEEDS-1

Let $\tilde{\alpha}_t := \frac{1}{2}\beta_d t^2 + \beta_m t$, where $\beta_d = \beta_{\max} - \beta_{\min} = 19.9$, $\beta_m = \beta_{\min} = 0.1$. We have

$$\sigma_t = \sqrt{e^{\tilde{\alpha}_t} - 1}, \qquad \alpha_t = e^{-\frac{1}{2}\tilde{\alpha}_t} = \frac{1}{\sqrt{\sigma_t^2 + 1}}$$

so, in particular, as $T = 1$, we have $\tilde{\alpha}_{t_0} = \tilde{\alpha}_1 = \frac{1}{2}(\beta_{\max} - \beta_{\min}) + \beta_{\min} = \frac{1}{2}(\beta_{\max} + \beta_{\min}) \approx 10.05$ and $\tilde{\alpha}_{t_{M+1}} = \tilde{\alpha}_0 = 0$. We deduce $\alpha_{t_0} = \alpha_1 \approx e^{-\frac{10.05}{2}} < 1$, $\alpha_{t_{M+1}} = \alpha_0 = 1$. Next, $\sigma_{t_0} = \sigma_1 \approx \sqrt{e^{10.05} - 1} > 1$ and $\sigma_{t_{M+1}} = \sigma_0 = 0$. As such, $\lambda_{t_0} = -\log(\sigma_1) := -L_0 < 0$ and $\lambda_t \xrightarrow[t \to t_{M+1}]{} +\infty$.
As such, we will set $t_M = t_{M+1} + \varepsilon$ the end time so that $\lambda_{t_M} = L_0$ is finite. This implies that, for $\lambda \in [-L_0, +\infty[$, $0 < e^{-L_0} \leqslant e^\lambda < 1 < e^{\lambda_{t_M}}$ and, for $\hat{T} = \lambda_{t_M} - \lambda_{t_0}$, $e^h = e^{\lambda_t - \lambda_s} \leqslant e^{\hat{T}}$. Now set, for $\lambda \in [-L_0, +\infty[$,

$$\hat{\alpha}_\lambda := \sqrt{\frac{1}{1 + e^{-2\lambda}}}, \qquad \hat{\sigma}_\lambda := \sqrt{\frac{1}{1 + e^{+2\lambda}}}.$$

Then, as $\lambda$ increases, $\hat{\alpha}_\lambda$ increases starting from $0 < \hat{\alpha}_{\lambda_{t_0}} < 1$ and $\hat{\alpha}_\lambda \xrightarrow[\lambda \to +\infty]{} 1$ while at the same time $\hat{\sigma}_\lambda$ decreases starting from $0 < \hat{\sigma}_{\lambda_{t_0}} < 1$ and $\hat{\sigma}_\lambda \xrightarrow[\lambda \to +\infty]{} 0$. As such, we can rewrite the exact solution (43) as

$$\widehat{\mathbf{x}}_{\lambda_t} = \frac{\hat{\alpha}_{\lambda_t}}{\hat{\alpha}_{\lambda_s}}\widehat{\mathbf{x}}_{\lambda_s} - 2\hat{\alpha}_{\lambda_t}\int_{\lambda_s}^{\lambda_t} e^{-\lambda}\hat{\boldsymbol{\epsilon}}_\theta(\widehat{\mathbf{x}}_\lambda, \lambda)\mathrm{d}\lambda - \sqrt{2}\hat{\alpha}_{\lambda_t}\int_{\lambda_s}^{\lambda_t} e^{-\lambda}\mathrm{d}\overline{\boldsymbol{\omega}}(\lambda). \tag{48}$$

Notice that $\frac{\hat{\alpha}_{\lambda_t}}{\hat{\alpha}_{\lambda_s}}$ is bounded for all $t, s$ by

$$\frac{\hat{\alpha}_{\lambda_t}}{\hat{\alpha}_{\lambda_s}} \leqslant \frac{1}{\hat{\alpha}_{\lambda_{t_0}}}$$

and $0 < \hat{\alpha}_\lambda < 1$ for all $\lambda \in [-L_0, +\infty[$.

Recall that the SEEDS-1 is defined recursively as

$$\mathbf{y}_{t_0} \leftarrow \mathbf{x}_T, \mathbf{y}_{t_i} \leftarrow \frac{\alpha_{t_i}}{\alpha_{t_{i-1}}}\mathbf{y}_{t_{i-1}} - 2\bar{\sigma}_{t_i}(e^{h_i} - 1)\boldsymbol{\epsilon}_\theta(\mathbf{y}_{t_{i-1}}, t_{i-1}) - \bar{\sigma}_{t_i}\sqrt{e^{2h_i} - 1}\epsilon_i$$

and, for simplicity, we will denote $\mathbf{y}_{\lambda_{t_i}}$ for the iterates (48).

Define a continuous approximation of SEEDS-1 as follows. For $\hat{h} = t - s$, we write $\hat{s} = [s/\hat{h}]\hat{h}$ where $[x]$ denotes the largest integer lesser or equal to $x$ and $I_{[A]}$ is the indicator function associated to a set $A$. We define the step function:

$$\widehat{\mathbf{y}}(\lambda) := \sum_{k \geqslant 0} I_{[\lambda_{t_k}, \lambda_{t_{k+1}}[}\mathbf{y}_{\lambda_{t_k}}$$

and the continuous approximation

$$\mathbf{y}(t) := \frac{\alpha_t}{\alpha_{t_0}}\mathbf{y}(t_0) - 2\alpha_t\int_{\lambda_{t_0}}^{\lambda_t} e^{-\hat{\lambda}}\hat{\boldsymbol{\epsilon}}_\theta(\widehat{\mathbf{y}}(\lambda), \hat{\lambda})\mathrm{d}\lambda - \sqrt{2}\alpha_t\int_{\lambda_{t_0}}^{\lambda_t} e^{-\lambda}\mathrm{d}\boldsymbol{\omega}_\lambda.$$

**Proposition C.8.** *There are two constants $C_1, C_2$ independent of $h$ such that, for all $t \in [0, T]$, we have*

$$\mathbb{E}\left[\sup_{t_0 \leqslant t \leqslant t_M} |\mathbf{y}(t)|^2\right] \leqslant C_1$$

$$\mathbb{E}\left[|\mathbf{y}(t) - \widehat{\mathbf{y}}(t)|^2\right] \leqslant C_2 h^2.$$

*Proof.* Recall the standard inequality $(a + b + c)^2 \leqslant 3(a^2 + b^2 + c^2)$ for $a, b, c \in \mathbb{R}$. Then

$$|\mathbf{y}(t)|^2 \quad \leqslant \quad 3\left[\left|\frac{\alpha_t}{\alpha_{t_0}}\mathbf{y}_{t_0}\right|^2 + 4\alpha_t^2\left|\int_{\lambda_{t_0}}^{\lambda_t} e^{-\hat{\lambda}}\hat{\epsilon}_\theta(\widehat{\mathbf{y}}(\lambda), \hat{\lambda})\mathrm{d}\lambda\right|^2 + 2\alpha_t^2\left|\int_{\lambda_{t_0}}^{\lambda_t} e^{-\lambda}\mathrm{d}\boldsymbol{\omega}_\lambda\right|^2\right].$$

Using the fact that $\alpha_t \leqslant 1$, we have, by writing $\hat{T} = \lambda_{t_M} - \lambda_{t_0}$ and taking the expectation:

$$\mathbb{E}\left[\sup_{t_0 \leqslant t \leqslant t_M} |\mathbf{y}(t)|^2\right]$$

$$\leqslant \quad 3\left[\left|\frac{\alpha_t}{\alpha_{t_0}}\right|^2 \mathbb{E}\left[|\mathbf{y}_{t_0}|^2\right] + 4\mathbb{E}\left[\left|\int_{\lambda_{t_0}}^{\lambda_t} e^{-\hat{\lambda}}\hat{\epsilon}_\theta(\widehat{\mathbf{y}}(\lambda), \hat{\lambda})\mathrm{d}\lambda\right|^2\right] + 2\alpha_t^2\mathbb{E}\left[\left|\int_{\lambda_{t_0}}^{\lambda_t} e^{-\lambda}\mathrm{d}\boldsymbol{\omega}_\lambda\right|^2\right]\right]$$

$$\leqslant \quad 3\left[\left|\frac{\alpha_t}{\alpha_{t_0}}\right|^2 \mathbb{E}\left[|\mathbf{y}_{t_0}|^2\right] + 4\hat{T}\mathbb{E}\left[\int_{\lambda_{t_0}}^{\lambda_t} |e^{-\hat{\lambda}}|^2\left|\hat{\epsilon}_\theta(\widehat{\mathbf{y}}(\lambda), \hat{\lambda})\right|^2\mathrm{d}\lambda\right] + 2\alpha_t^2\mathbb{E}\left[\int_{\lambda_{t_0}}^{\lambda_t} |e^{-\lambda}|^2\mathrm{d}\lambda\right]\right]$$

$$\leqslant \quad 3\left[\left|\frac{\alpha_t}{\alpha_{t_0}}\right|^2 \mathbb{E}\left[|\mathbf{y}_{t_0}|^2\right] + 4\hat{T}\int_{\lambda_{t_0}}^{\lambda_t} |e^{-\hat{\lambda}}|^2\mathbb{E}\left[\left|\hat{\epsilon}_\theta(\widehat{\mathbf{y}}(\lambda), \hat{\lambda})\right|^2\right]\mathrm{d}\lambda + 2\alpha_t^2\int_{\lambda_{t_0}}^{\lambda_t} e^{-2\lambda}\mathrm{d}\lambda\right]$$

$$\leqslant \quad 3K\left[\mathbb{E}\left[|\mathbf{y}_{t_0}|^2\right] + 4\hat{T}\int_{\lambda_{t_0}}^{\lambda_t} \mathbb{E}\left[\left|\hat{\epsilon}_\theta(\widehat{\mathbf{y}}(\lambda), \hat{\lambda})\right|^2\right]\mathrm{d}\lambda + \left(e^{2(\lambda_t - \lambda_{t_0})} - 1\right)\right]$$

$$\leqslant \quad 3K\left[\mathbb{E}\left[|\mathbf{y}_{t_0}|^2\right] + 4\hat{T}L_2\int_{\lambda_{t_0}}^{\lambda_t} \left(1 + \mathbb{E}\left[|\widehat{\mathbf{y}}(\lambda)|^2\right]\right)\mathrm{d}\lambda + (e^{2\hat{T}} - 1)\right]$$

$$\leqslant \quad 3K\left[\mathbb{E}\left[|\mathbf{y}_{t_0}|^2\right] + 4\hat{T}L_2\hat{T} + e^{2\hat{T}} - 1 + 4\hat{T}L_2\int_{\lambda_{t_0}}^{\lambda_t} \mathbb{E}\left[|\widehat{\mathbf{y}}(\lambda)|^2\right]\mathrm{d}\lambda\right]$$

$$\leqslant \quad 3K\left(\mathbb{E}\left[|\mathbf{y}_{t_0}|^2\right] + 4\hat{T}L_2\hat{T} + e^{2\hat{T}} - 1\right) + 3K4\hat{T}L_2\int_{\lambda_{t_0}}^{\lambda_t} \mathbb{E}\left[|\widehat{\mathbf{y}}(\lambda)|^2\right]\mathrm{d}\lambda$$

$$\leqslant \quad 3K\left(\mathbb{E}\left[|\mathbf{x}(t_0)|^2\right] + 4\hat{T}^2L_2 + e^{2\hat{T}} - 1\right) + 12K\hat{T}L_2\int_{\lambda_{t_0}}^{\lambda_t} \mathbb{E}\left[\sup_{\lambda_0 \leqslant r \leqslant \lambda} |\mathbf{y}(r)|^2\right]\mathrm{d}\lambda,$$

where we used the linearity of expectation, Hölder's inequality, Doob's martingale inequality, Itô isometry, the linear growth condition of $\hat{\epsilon}_\theta$, and we set $K = \max\left(\left|\frac{\alpha_t}{\alpha_{t_0}}\right|^2, |e^{L_0}|^2, \bar{\sigma}_t^2, 1\right)$. As we know that $\mathbb{E}\left[|\mathbf{x}(t_0)|^2\right] < \infty$, we apply Grönwall's inequality in the last line to obtain

$$\mathbb{E}\left[\sup_{t_0 \leqslant t \leqslant t_M} |\mathbf{y}(t)|^2\right] \leqslant C_1, \qquad C_1 := 3K\left[\mathbb{E}\left[|\mathbf{x}(t_0)|^2\right] + 4\hat{T}^2L_2 + e^{2\hat{T}} - 1\right]e^{12K\hat{T}L_2}.$$

Second, we have, for $s = t_i$, $u = t_{i+1}$ and $t \in [t_\lambda(u), t_\lambda(s)[$,

$$\mathbf{y}(t) - \widehat{\mathbf{y}}(t) = \left(\frac{\alpha_t}{\alpha_s} - 1\right)\mathbf{y}_s - 2\alpha_t\int_{\lambda_s}^{\lambda_t} e^{-\hat{\lambda}}\hat{\epsilon}_\theta(\mathbf{y}_{\lambda_s}, \lambda_s)\mathrm{d}\lambda,$$

so that, using Hölder's inequality, we get

$$|\mathbf{y}(t) - \widehat{\mathbf{y}}(t)|^2 \quad \leqslant \quad 3\left[\left|\frac{\alpha_t}{\alpha_s} - 1\right|^2|\mathbf{y}_s|^2 + 4h\int_{\lambda_s}^{\lambda_t} |e^{-\hat{\lambda}}|^2|\hat{\epsilon}_\theta(\mathbf{y}_{\lambda_s}, \lambda_s)|^2\mathrm{d}\lambda\right].$$

Now using Itô isometry we obtain:

$$\mathbb{E}\left[|\mathbf{y}(t) - \widehat{\mathbf{y}}(t)|^2\right] \quad \leqslant \quad 3\left[\left|\frac{\alpha_t}{\alpha_s} - 1\right|^2\mathbb{E}\left[|\mathbf{y}_s|^2\right] + 4h\mathbb{E}\left[\int_{\lambda_s}^{\lambda_t} |e^{-\hat{\lambda}}|^2|\hat{\epsilon}_\theta(\mathbf{y}_{\lambda_s}, \lambda_s)|^2\mathrm{d}\lambda\right]\right].$$

Now, using the bound $\mathbb{E}\left[\max |\mathbf{y}_t|^2\right] \leqslant C_1$, the fact that $\epsilon_\theta$ is bounded and the same arguments as above, we get

$$
\begin{aligned}
\mathbb{E}\left[|\mathbf{y}(t) - \widehat{\mathbf{y}}(t)|^2\right] &\leqslant 3\left[\left|\frac{\alpha_t}{\alpha_s} - 1\right|^2 C_1 + 4Kh\mathbb{E}\left[\int_{\lambda_s}^{\lambda_t} |\hat{\epsilon}_\theta(\mathbf{y}_{\lambda_s}, \lambda_s)|^2\, \mathrm{d}\lambda\right]\right] \\
&\leqslant 3\left[\left|\frac{\alpha_t}{\alpha_s} - 1\right|^2 C_1 + 4KhL_2\int_{\lambda_s}^{\lambda_t} (1 + C_1)\mathrm{d}\lambda\right] \\
&\leqslant 3\left[\left|\frac{\alpha_t}{\alpha_s} - 1\right|^2 C_1 + 4Kh^2 L_2(1 + C_1)\right].
\end{aligned}
$$

Finally, as we have

$$
e^h = \frac{\sigma_s}{\sigma_t} = \frac{\alpha_t}{\alpha_s}\frac{\sqrt{1 - \alpha_s^2}}{\sqrt{1 - \alpha_t^2}},
$$

and $1 < \frac{\sqrt{1-\alpha_s^2}}{\sqrt{1-\alpha_t^2}} \xrightarrow[M\to\infty]{} 1$, we obtain $\left|\frac{\alpha_t}{\alpha_s} - 1\right|^2 \sim |e^h - 1|^2 \sim \mathcal{O}(h^2)$. Now, by denoting $|e^h - 1|^2 \leqslant K_2 h^2$, we conclude that

$$
\mathbb{E}\left[|\mathbf{y}(t) - \widehat{\mathbf{y}}(t)|^2\right] \leqslant C_2 h^2,
$$

with $C_2 := 3K_2 C_1 + 4KL_2(1 + C_1)$. This finishes the proof. $\qquad\square$

### C.2.3 Proof of Theorem 4.1

Let's now take a look at the approximation given $\mathbf{y}_{t_0} = \mathbf{x}_{t_0}$. We have

$$
\mathbf{y}_t - \mathbf{x}_t = 2\alpha_t \int_{\lambda_{t_0}}^{\lambda_t} \left[e^{-\lambda}\widehat{\epsilon}_\theta(\widehat{\mathbf{x}}_\lambda, \lambda) - e^{-\hat{\lambda}}\widehat{\epsilon}_\theta(\widehat{\mathbf{y}}_\lambda, \hat{\lambda})\right] \mathrm{d}\lambda.
$$

Using the inequality $\alpha_t \leq 1$, the Lipschitz property of $\widehat{\epsilon}_\theta$, Assumption (C.1), and Hölder's inequality we deduce the bound:

$$
\begin{aligned}
|\mathbf{y}_t - \mathbf{x}_t|^2 &\leqslant 2\left[4\left|\int_{\lambda_{t_0}}^{\lambda_t} \left[e^{-\lambda}\widehat{\epsilon}_\theta(\widehat{\mathbf{x}}_\lambda, \lambda) - e^{-\hat{\lambda}}\widehat{\epsilon}_\theta(\widehat{\mathbf{y}}_\lambda, \hat{\lambda})\right] \mathrm{d}\lambda\right|^2\right] \\
&\leqslant 8\hat{T}\int_{\lambda_{t_0}}^{\lambda_t} \left|e^{-\lambda}\widehat{\epsilon}_\theta(\widehat{\mathbf{x}}_\lambda, \lambda) - e^{-\hat{\lambda}}\widehat{\epsilon}_\theta(\widehat{\mathbf{y}}_\lambda, \hat{\lambda})\right|^2 \mathrm{d}\lambda.
\end{aligned}
$$

Taking the expectation yields

$$
\begin{aligned}
&\mathbb{E}\left[\sup_{t_0 \leqslant s \leqslant t} |\mathbf{y}_s - \mathbf{x}_s|^2\right] \\
&\leqslant 8\hat{T}\mathbb{E}\left[\int_{\lambda_{t_0}}^{\lambda_t} \left|e^{-\lambda}\widehat{\epsilon}_\theta(\widehat{\mathbf{x}}_\lambda, \lambda) - e^{-\hat{\lambda}}\widehat{\epsilon}_\theta(\widehat{\mathbf{y}}_\lambda, \hat{\lambda})\right|^2 \mathrm{d}\lambda\right].
\end{aligned}
$$

Now, for the first integral, by writing:

$$
\begin{aligned}
&e^{-\lambda}\widehat{\epsilon}_\theta(\widehat{\mathbf{x}}_\lambda, \lambda) - e^{-\hat{\lambda}}\widehat{\epsilon}_\theta(\widehat{\mathbf{y}}_\lambda, \hat{\lambda}) \\
&= e^{-\lambda}\widehat{\epsilon}_\theta(\widehat{\mathbf{x}}_\lambda, \lambda) - e^{-\hat{\lambda}}\widehat{\epsilon}_\theta(\widehat{\mathbf{x}}_\lambda, \lambda) + e^{-\hat{\lambda}}\widehat{\epsilon}_\theta(\widehat{\mathbf{x}}_\lambda, \lambda) - e^{-\hat{\lambda}}\widehat{\epsilon}_\theta(\widehat{\mathbf{x}}_\lambda, \hat{\lambda}) \\
&\quad + e^{-\hat{\lambda}}\widehat{\epsilon}_\theta(\widehat{\mathbf{x}}_\lambda, \hat{\lambda}) - e^{-\hat{\lambda}}\widehat{\epsilon}_\theta(\mathbf{y}_\lambda, \hat{\lambda}) + e^{-\hat{\lambda}}\widehat{\epsilon}_\theta(\mathbf{y}_\lambda, \hat{\lambda}) - e^{-\hat{\lambda}}\widehat{\epsilon}_\theta(\widehat{\mathbf{y}}_\lambda, \hat{\lambda}) \\
&= (e^{-\lambda} - e^{-\hat{\lambda}})\widehat{\epsilon}_\theta(\widehat{\mathbf{x}}_\lambda, \lambda) + e^{-\hat{\lambda}}\left(\widehat{\epsilon}_\theta(\widehat{\mathbf{x}}_\lambda, \lambda) - \widehat{\epsilon}_\theta(\widehat{\mathbf{x}}_\lambda, \hat{\lambda})\right) \\
&\quad + e^{-\hat{\lambda}}\left(\widehat{\epsilon}_\theta(\widehat{\mathbf{x}}_\lambda, \hat{\lambda}) - \widehat{\epsilon}_\theta(\mathbf{y}_\lambda, \hat{\lambda}) + \widehat{\epsilon}_\theta(\mathbf{y}_\lambda, \hat{\lambda}) - \widehat{\epsilon}_\theta(\widehat{\mathbf{y}}_\lambda, \hat{\lambda})\right),
\end{aligned}
$$

we can state the following inequalities:

$$\mathbb{E}\left[\int_{\lambda_{t_0}}^{\lambda_t}\left|e^{-\lambda}\widehat{\boldsymbol{\epsilon}}_\theta(\widehat{\mathbf{x}}_\lambda,\lambda)-e^{-\hat\lambda}\widehat{\boldsymbol{\epsilon}}_\theta(\widehat{\mathbf{y}}_\lambda,\hat\lambda)\right|^2\mathrm{d}\lambda\right]$$

$$\leqslant\ 3\int_{\lambda_{t_0}}^{\lambda_t}|e^{-\lambda}-e^{-\hat\lambda}|^2\mathbb{E}\left[|\widehat{\boldsymbol{\epsilon}}_\theta(\widehat{\mathbf{x}}_\lambda,\lambda)|^2\right]\mathrm{d}\lambda+3\mathbb{E}\left[\int_{\lambda_{t_0}}^{\lambda_t}|e^{-\hat\lambda}|^2\left|\widehat{\boldsymbol{\epsilon}}_\theta(\widehat{\mathbf{x}}_\lambda,\lambda)-\widehat{\boldsymbol{\epsilon}}_\theta(\widehat{\mathbf{x}}_\lambda,\hat\lambda)\right|^2\mathrm{d}\lambda\right]$$

$$+3\mathbb{E}\left[\int_{\lambda_{t_0}}^{\lambda_t}|e^{-\hat\lambda}|^2\left|\widehat{\boldsymbol{\epsilon}}_\theta(\widehat{\mathbf{x}}_\lambda,\hat\lambda)-\widehat{\boldsymbol{\epsilon}}_\theta(\mathbf{y}_\lambda,\hat\lambda)+\widehat{\boldsymbol{\epsilon}}_\theta(\mathbf{y}_\lambda,\hat\lambda)-\widehat{\boldsymbol{\epsilon}}_\theta(\widehat{\mathbf{y}}_\lambda,\hat\lambda)\right|^2\mathrm{d}\lambda\right]$$

$$\leqslant\ 3\int_{\lambda_{t_0}}^{\lambda_t}|e^{-\lambda}-e^{-\hat\lambda}|^2\mathbb{E}\left[|\widehat{\boldsymbol{\epsilon}}_\theta(\widehat{\mathbf{x}}_\lambda,\lambda)|^2\right]\mathrm{d}\lambda+3KL_3\mathbb{E}\left[\int_{\lambda_{t_0}}^{\lambda_t}(1+|\widehat{\mathbf{x}}_\lambda|^2)|(\lambda-\hat\lambda)|^2\mathrm{d}\lambda\right]$$

$$+3\mathbb{E}\left[\int_{\lambda_{t_0}}^{\lambda_t}|e^{-\hat\lambda}|^2\left|\widehat{\boldsymbol{\epsilon}}_\theta(\widehat{\mathbf{x}}_\lambda,\hat\lambda)-\widehat{\boldsymbol{\epsilon}}_\theta(\mathbf{y}_\lambda,\hat\lambda)+\widehat{\boldsymbol{\epsilon}}_\theta(\mathbf{y}_\lambda,\hat\lambda)-\widehat{\boldsymbol{\epsilon}}_\theta(\widehat{\mathbf{y}}_\lambda,\hat\lambda)\right|^2\mathrm{d}\lambda\right]$$

$$\leqslant\ 3L_2\int_{\lambda_{t_0}}^{\lambda_t}|e^{-\hat\lambda}|^2|e^{\lambda-\hat\lambda}-1|^2(1+\mathbb{E}[|\widehat{\mathbf{x}}_\lambda|^2])\mathrm{d}\lambda+3KL_3h^2\mathbb{E}\int_{\lambda_{t_0}}^{\lambda_t}(1+|\widehat{\mathbf{x}}_\lambda|^2)\mathrm{d}\lambda$$

$$+3KL_1\int_{\lambda_{t_0}}^{\lambda_t}\mathbb{E}\left[|\widehat{\mathbf{x}}_\lambda-\mathbf{y}_\lambda+\mathbf{y}_\lambda-\widehat{\mathbf{y}}_\lambda|^2\right]\mathrm{d}\lambda.$$

Thus, we obtain

$$\mathbb{E}\left[\int_{\lambda_{t_0}}^{\lambda_t}\left|e^{-\lambda}\widehat{\boldsymbol{\epsilon}}_\theta(\widehat{\mathbf{x}}_\lambda,\lambda)-e^{-\hat\lambda}\widehat{\boldsymbol{\epsilon}}_\theta(\widehat{\mathbf{y}}_\lambda,\hat\lambda)\right|^2\mathrm{d}\lambda\right]$$

$$\leqslant\ 3L_2K\hat{T}|e^h-1|^2(1+C_1)+3L_3h^2K(1+C_1)\hat{T}$$

$$+3KL_1\int_{\lambda_{t_0}}^{\lambda_t}\mathbb{E}\left[|\widehat{\mathbf{x}}_\lambda-\mathbf{y}_\lambda+\mathbf{y}_\lambda-\widehat{\mathbf{y}}_\lambda|^2\right]\mathrm{d}\lambda$$

$$\leqslant\ 3L_2K\hat{T}|e^h-1|^2(1+C_1)+3L_3h^2K(1+C_1)\hat{T}$$

$$+6KL_1\int_{\lambda_{t_0}}^{\lambda_t}\mathbb{E}\left[|\widehat{\mathbf{x}}_\lambda-\mathbf{y}_\lambda|^2+|\mathbf{y}_\lambda-\widehat{\mathbf{y}}_\lambda|^2\right]\mathrm{d}\lambda$$

$$\leqslant\ 3L_2K\hat{T}|e^h-1|^2(1+C_1)+3L_3h^2K(1+C_1)\hat{T}$$

$$+6KL_1\hat{T}C_2h^2+6KL_1\int_{\lambda_{t_0}}^{\lambda_t}\mathbb{E}\left[|\widehat{\mathbf{x}}_\lambda-\mathbf{y}_\lambda|^2\right]\mathrm{d}\lambda$$

$$\leqslant\ 3L_2K\hat{T}|e^h-1|^2(1+C_1)+3L_3h^2K(1+C_1)\hat{T}$$

$$+6KL_1\hat{T}C_2h^2+6KL_1\int_{\lambda_{t_0}}^{\lambda_t}\mathbb{E}\left[\sup_{\lambda_0\leqslant r\leqslant\lambda}|\widehat{\mathbf{x}}(r)-\mathbf{y}(r)|^2\right]\mathrm{d}\lambda$$

$$\leqslant\ 3K\hat{T}(1+C_1)[L_2|e^h-1|^2+L_3h^2]+6KL_1\hat{T}C_2h^2$$

$$+6KL_1\int_{\lambda_{t_0}}^{\lambda_t}\mathbb{E}\left[\sup_{\lambda_0\leqslant r\leqslant\lambda}|\widehat{\mathbf{x}}(r)-\mathbf{y}(r)|^2\right]\mathrm{d}\lambda.$$

Putting everything together yields

$$\mathbb{E}\left[\sup_{t_0 \leqslant s \leqslant t} |\mathbf{y}_s - \mathbf{x}_s|^2\right]$$

$$\leqslant 8\hat{T}[3K\hat{T}(1+C_1)[L_2|e^h - 1|^2 + L_3 h] + 6KL_1\hat{T}C_2 h^2]$$

$$+8\hat{T}6KL_1\int_{\lambda_{t_0}}^{\lambda_t}\mathbb{E}\left[\sup_{\lambda_0 \leqslant r \leqslant \lambda}|\widehat{\mathbf{x}}(r) - \mathbf{y}(r)|^2\right]\mathrm{d}\lambda$$

$$\leqslant 24K\hat{T}^2(1+C_1)[L_2|e^h - 1|^2 + L_3 h^2] + 48KL_1\hat{T}^2 C_2 h^2$$

$$+48\hat{T}KL_1\int_{\lambda_{t_0}}^{\lambda_t}\mathbb{E}\left[\sup_{\lambda_0 \leqslant r \leqslant \lambda}|\widehat{\mathbf{x}}(r) - \mathbf{y}(r)|^2\right]\mathrm{d}\lambda.$$

Now, by denoting $|e^h - 1|^2 \leqslant K_2 h^2$, we apply the continuous version of the Grönwall Lemma to obtain:

$$\mathbb{E}\left[\sup_{t_0 \leqslant s \leqslant t} |\mathbf{y}_t - \mathbf{x}_t|^2\right] \leqslant C_0 h^2,$$

where

$$C_0 = [24K\hat{T}^2(1+C_1)[L_2 K_2 + L_3] + 48K\hat{T}^2 L_1 C_2]e^{48\hat{T}KL_1}.$$

Finally, using Lyapunov's inequality we obtain, for $C = \sqrt{C_0}$

$$\mathbb{E}[|\mathbf{y}_T - \mathbf{x}_T|] \leqslant \left(\mathbb{E}[|\mathbf{y}_T - \mathbf{x}_T|^2]\right)^{1/2} \leqslant Ch.$$

In other words, for $C$ as stated above, we have the following inequality

$$\sqrt{\mathbb{E}\left[\sup_{t_0 \leqslant t \leqslant t_M}|\widetilde{\mathbf{x}}_t - \mathbf{x}_t|^2\right]} \leqslant Ch, \qquad \text{as } h \longrightarrow 0.$$

*Remark* C.9. From the above, it is easy to induce that the following order for the one step error

$$\mathbb{E}[|\mathbf{x}(t_1) - \mathbf{y}_1|^2] = \mathcal{O}(h^3).$$

Now, since $G_t g(t) = 0$ in this case (additive noise), it is easy to see from the truncated Itô-Taylor expansion of $\mathbf{x}$ that $|\mathbb{E}[\mathbf{x}(t_1) - \mathbf{y}_1]| = \mathcal{O}(h^2)$. As such, we apply Theorem C.4 to conclude that SEEDS-1 has strong convergence of global order 1.0.

### C.2.4 Discrete-time approximation

By Theorem 4.1, we know that SEEDS-1, being of strong order 1.0, it is immediately also of weak order 1. Nonetheless, let's give a discrete approach of this statement that we will use for the proofs of convergence for the remaining solvers as stated in Corollary 4.4. Define the discrete time process:

$$\mathbf{y}_{t_0} \leftarrow \mathbf{x}_T, \mathbf{y}_{t_i} \leftarrow \frac{\alpha_{t_i}}{\alpha_{t_{i-1}}}\mathbf{y}_{t_{i-1}} - 2\sigma_{t_i}(e^{h_i} - 1)\boldsymbol{\epsilon}(\mathbf{y}_{t_{i-1}}, t_{i-1}) - \sqrt{2}\alpha_{t_i}\int_{\lambda_{t_{i-1}}}^{\lambda_{t_i}}e^{-s}\mathrm{d}\bar{\boldsymbol{\omega}}(s).$$

We will prove that $\mathbb{E}[|\mathbf{y}_{t_M} - \mathbf{x}_{t_M}|]$ is of order $h$ as $h \longrightarrow 0$. Note that $(\mathbf{y}_{t_i})_i$ has the same distribution as $(\widetilde{\mathbf{x}}_{t_i})_i$ described in Algorithm 2 since the stochastic integrals $\left(\int_{\lambda_{t_{i-1}}}^{\lambda_{t_i}}e^{-s}\mathrm{d}\bar{\boldsymbol{\omega}}(s)\right)_i$ are independent and each $\int_{\lambda_{t_{i-1}}}^{\lambda_{t_i}}e^{-s}\mathrm{d}\bar{\boldsymbol{\omega}}(s)$ is distributed as $\frac{1}{\sqrt{2}}\frac{\sigma_{t_i}}{\alpha_{t_i}}\sqrt{e^{2h_i} - 1}\boldsymbol{\epsilon}$ with $h_i = \lambda_{t_i} - \lambda_{t_{i-1}}$, $\boldsymbol{\epsilon} \sim \mathcal{N}(\mathbf{0}, \mathbf{I}_d)$. We have:

$$\mathbf{y}_{t_i} - \mathbf{x}_{t_i} = \frac{\alpha_{t_i}}{\alpha_{t_{i-1}}}(\mathbf{y}_{t_{i-1}} - \mathbf{x}_{t_{i-1}}) + 2\alpha_{t_i}\int_{\lambda_{t_{i-1}}}^{\lambda_{t_i}}e^{-\lambda}(\widehat{\boldsymbol{\epsilon}}(\widehat{\mathbf{x}}_\lambda, \lambda) - \boldsymbol{\epsilon}(\mathbf{y}_{t_{i-1}}, t_{i-1}))\mathrm{d}\lambda.$$

For simplicity, in what follows $C$ will denote a constant not dependent on the subdivision of $[0, T]$ that may change from one line to the next by systematically denoting the maximum value of the

different constants appearing in the line before. Using the inequality $\alpha_t \leq 1$, the Lipschitz property of $\epsilon$, we deduce the bound:

$$|\mathbf{y}_{t_i} - \mathbf{x}_{t_i}| \leq \frac{\alpha_{t_i}}{\alpha_{t_{i-1}}} |\mathbf{y}_{t_{i-1}} - \mathbf{x}_{t_{i-1}}| + C\left[\int_{\lambda_{t_{i-1}}}^{\lambda_{t_i}} e^{-\lambda}|\widehat{\mathbf{x}}_\lambda - \mathbf{y}_{t_{i-1}}|\mathrm{d}\lambda + \int_{\lambda_{t_{i-1}}}^{\lambda_{t_i}} e^{-\lambda}|t_\lambda(\lambda) - t_{i-1}|\mathrm{d}\lambda\right].$$

Notice that

$$
\begin{aligned}
\int_{\lambda_{t_{i-1}}}^{\lambda_{t_i}} e^{-\lambda}|\widehat{\mathbf{x}}_\lambda - \mathbf{y}_{t_{i-1}}|\mathrm{d}\lambda &\leq \int_{\lambda_{t_{i-1}}}^{\lambda_{t_i}} e^{-\lambda}|\widehat{\mathbf{x}}_\lambda - \mathbf{x}_{t_{i-1}}|\mathrm{d}\lambda + \int_{\lambda_{t_{i-1}}}^{\lambda_{t_i}} e^{-\lambda}|\mathbf{x}_{t_{i-1}} - \mathbf{y}_{t_{i-1}}|\mathrm{d}\lambda \\
&\leq \int_{\lambda_{t_{i-1}}}^{\lambda_{t_i}} e^{-\lambda}|\widehat{\mathbf{x}}_\lambda - \mathbf{x}_{t_{i-1}}|\mathrm{d}\lambda + Ch|\mathbf{x}_{t_{i-1}} - \mathbf{y}_{t_{i-1}}|,
\end{aligned}
$$

and recall that $\widehat{\mathbf{x}}_u = \mathbf{x}_{t_\lambda(u)}$. Using the fact that $t_\lambda$ is increasing and Lemma C.11, we have:

$$\mathbb{E}[\int_{\lambda_{t_{i-1}}}^{\lambda_{t_i}} e^{-\lambda}|\widehat{\mathbf{x}}_\lambda - \mathbf{x}_{t_{i-1}}|\mathrm{d}\lambda] \leq C\int_{\lambda_{t_{i-1}}}^{\lambda_{t_i}} e^{-u}\sqrt{|t_\lambda - t_{i-1}|}\mathrm{d}\lambda \leq Ch^{3/2}.$$

Introduce $U_i = \mathbb{E}[|\mathbf{y}_{t_i} - \mathbf{x}_{t_i}|]$. Since $\int_{\lambda_{t_{i-1}}}^{\lambda_{t_i}} e^{-\lambda}|t_\lambda(\lambda) - t_{i-1}|\mathrm{d}\lambda \leq Ch^2$:

$$U_i \leq \left(\frac{\alpha_{t_i}}{\alpha_{t_{i-1}}} + Ch\right) U_{i-1} + Ch^2.$$

Let $a_i = \frac{\alpha_{t_i}}{\alpha_{t_{i-1}}} + Ch$ and $b_i = Ch^2$. By applying Lemma C.12, we have: $U_M \leq A_M U_0 + \sum_{k=1}^M A_{k,n} b_k$ with $A_M = \prod_{k=1}^M a_k$ and $A_{k,M} = A_M/A_k = \prod_{i=k+1}^M a_i$. Note that $U_0 = 0$ since $\mathbf{y}_{t_0} = \mathbf{x}_T$ and so:

$$U_M \leq Ch^2 \sum_{k=0}^{M-1} (\sup_{1\leq i\leq M} a_i)^k.$$

Using our hypothesis, we can bound:

$$\sum_{k=0}^{M-1} (\sup_{1\leq i\leq M} a_i)^k \leq \sum_{k=0}^{M-1} (\exp(Ch) + Ch)^k = \frac{(\exp(Ch) + Ch)^{1/h} - 1}{(\exp(Ch) + Ch) - 1}.$$

The quantity on the right is of order $C/h$. Indeed, as $h$ goes to 0, $\exp(Ch) + Ch - 1$ is equivalent to $2Ch$ and $(\exp(Ch) + Ch)^{1/h}$ converges to a constant. This gives the bound $U_M \leq Ch$, when $h \to 0$ and by using Proposition C.8 and Theorem C.6 we conclude that SEEDS-1 is convergent of weak order 1.

### C.2.5 Useful Lemmas

**Lemma C.10.** *(Continuous Grönwall Lemma) Let $I = [a, b]$ denote a compact interval of the real line with $a < b$. Let $\alpha, \beta, u$ be continuous real-valued functions defined on $I$. Assume $\beta$ is non-negative, $\alpha$ is non-decreasing and $u$ satisfies the integral inequality*

$$u(t) \leq \alpha(t) + \int_a^t \beta(s)u(s)\,\mathrm{d}s, \qquad \forall t \in I.$$

*Then*

$$u(t) \leq \alpha(t)\exp\left(\int_a^t \beta(s)\,\mathrm{d}s\right), \qquad \forall t \in I.$$

**Lemma C.11.** *Assume the following forward SDE is satisfied $dX_t = F(t, X_t)dt + G(t)dW_t, t \in [0, T]$, where $T > 0$, $F(t, x)$ is Lipschitz with respect to $(t, x)$, $G$ is continuous and $X_0$ is an integrable random variable. Then, there exists $C > 0$ such that for all $s < t \in [0, T]$ with $t - s \leq 1$,*

$$\mathbb{E}[|X_t - X_s|] \leq C\sqrt{t - s}.$$

*Proof.* We take $s = 0$ and apply the triangular inequality:

$$|X_t - X_0| \leq \int_0^t |F(u, X_u) - F(0, X_0))| \mathrm{d}u + |\int_0^t G(u)dW_u| + t|F(0, X_0)|.$$

Setting $u(t) = \mathbb{E}[|X_t - X_0|]$ and taking the expectation, we deduce:

$$u(t) \leq K\frac{t^2}{2} + t\mathbb{E}[|F(0, X_0)|] + \mathbb{E}[(\int_0^t G(u)dW_u)^2]^{\frac{1}{2}} + K\int_0^t u(s)ds,$$

where $K$ is a positive constant. Note that $\mathbb{E}[(\int_0^t G(u)dW_u)^2]^{\frac{1}{2}} = [\int_0^t G^2(s)ds]^{\frac{1}{2}}$ by the Itô isometry property which is less than $C\sqrt{t}$ since $G$ is continuous. Thus, we have proved that:

$$u(t) \leq \alpha(t) + K\int_0^t u(s)ds,$$

where $\alpha(t) = K\frac{t^2}{2} + t\mathbb{E}[|F(0, X_0)|] + C\sqrt{t}$ is non-decreasing. By Lemma C.10: $u(t) \leq \alpha(t)\exp(Kt) \leq \alpha(t)\exp(KT)$. Since $\alpha(t) \leq C\sqrt{t}$ for $t \in [0, 1]$, the lemma holds. $\qquad\square$

**Lemma C.12.** *(Discrete Grönwall Lemma) Consider a real number sequence $(u_n)_n$ such that*

$$u_{n+1} \leq a_{n+1}u_n + b_{n+1}, n \geq 0$$

*where $(a_n)$ and $(b_n)_n$ are two given sequences such that $b_n$ is positive. Then*

$$u_n \leq A_n u_0 + \sum_{k=1}^n A_{k,n} b_k,$$

*where $A_n = \prod_{k=1}^n a_k$ and $A_{k,n} = A_n/A_k = \prod_{i=k+1}^n a_i$.*

## C.3 Proof of Proposition 4.3

Let us prove this statement for SEEDS-2. On the one hand, we have

$$\mathbf{u}_1 = \frac{\alpha_{s_1}}{\alpha_s}\tilde{\mathbf{x}}_s - 2\bar{\sigma}_{s_1}\left(e^{\frac{h}{2}} - 1\right)F_\theta(\tilde{\mathbf{x}}_s, s) - B^1, \qquad B^1 := \sqrt{2}\alpha_{s_1}\int_{\lambda_s}^{\lambda_{s_1}} e^{-\lambda}\mathrm{d}\omega_\lambda.$$

Now $B^1$ depends on the Brownian movement $(\omega_{\lambda_{s_1}} - \omega_{\lambda_s})_{\lambda_{s_1} \geqslant \lambda_s}$. By the Markov property, this is independent of $(\omega_{\lambda_u})_{\lambda_u \leqslant \lambda_s}$. Since $\tilde{\mathbf{x}}_s$ is a function of $(\omega_{\lambda_u})_{\lambda_u \leqslant \lambda_s}$, we deduce that $B^1$ has to be independent of all $(\tilde{\mathbf{x}}_u)_{u \geqslant s}$. By Itô Isometry, we obtain

$$B^1 = \bar{\sigma}_{s_1}\sqrt{e^h - 1}z^1, \quad z^1 \sim \mathcal{N}(\mathbf{0}, \mathbf{I}_d).$$

On the other hand, the update $\tilde{\mathbf{x}}_t$ is

$$\tilde{\mathbf{x}}_t = \frac{\alpha_t}{\alpha_s}\tilde{\mathbf{x}}_s - 2\bar{\sigma}_t(e^h - 1)F_\theta(\mathbf{u}_1, s_1) - B^0, \qquad B^0 = \sqrt{2}\alpha_t\int_{\lambda_s}^{\lambda_t} e^{-\lambda}\mathrm{d}\omega_\lambda.$$

Hence, the Wiener process $W = \{\omega_\lambda : \lambda \in [\lambda_s, \lambda_t]\}$ is predetermined on the interval $[\lambda_s, \lambda_{s_1}]$. Then, by using independent increments property of Wiener process $W$, we can deduce, for $0 \leq \lambda_s < \lambda_{s_1} < \lambda_t$, that

$$\omega_{\lambda_{s_1}} - \omega_{\lambda_s} \quad \text{and} \quad \omega_{\lambda_t} - \omega_{\lambda_{s_1}} \text{ are independent.}$$

Then, by the above Brownian independence property, the random variables $B^0$ and $B^1$ are

- Independent on non-overlapping time intervals

- Dependent on overlapping intervals.

By the Chasles rule, we then decompose

$$B^0 = \sqrt{2}\alpha_t \int_{\lambda_s}^{\lambda_{s_1}} e^{-\lambda}\mathrm{d}\omega_\lambda + \sqrt{2}\alpha_t \int_{\lambda_{s_1}}^{\lambda_t} e^{-\lambda}\mathrm{d}\omega_\lambda$$

and, as

$$\int_{\lambda_s}^{\lambda_{s_1}} e^{-\lambda}\mathrm{d}\omega_\lambda = \frac{B^1}{\sqrt{2}\alpha_{s_1}},$$

we obtain

$$B^0 = \sqrt{2}\alpha_t \left[ \frac{B^1}{\sqrt{2}\alpha_{s_1}} + \int_{\lambda_{s_1}}^{\lambda_t} e^{-\lambda}\mathrm{d}\omega_\lambda \right].$$

Then we have

$$\begin{aligned}
B &= \sqrt{2}\alpha_t \int_{\lambda_s}^{\lambda_{s_1}} e^{-\lambda}\mathrm{d}\omega_\lambda + \sqrt{2}\alpha_t \int_{\lambda_s}^{\lambda_t} e^{-\lambda}\mathrm{d}\omega_\lambda \\
&= \sqrt{2}\alpha_t \frac{1}{\sqrt{2}}\sigma_{s_1}\sqrt{e^{2(\lambda_{s_1}-\lambda_s)}-1}z^1 + \sqrt{2}\alpha_t \frac{1}{\sqrt{2}}\sigma_t\sqrt{e^{2(\lambda_t-\lambda_s)}-1}z^2 \\
&= \alpha_t\sigma_{s_1}\sqrt{e^h-1}z^1 + \alpha_t\sigma_t\sqrt{e^{2h}-1}z^2 \\
&= \bar{\sigma}_t \left( \frac{\sigma_{s_1}}{\sigma_t}\sqrt{e^h-1}z^1 + \sqrt{e^{2h}-1}z^2 \right) \\
&= \bar{\sigma}_t \left( e^{\lambda_t-\lambda_{s_1}}\sqrt{e^h-1}z^1 + \sqrt{e^{2h}-1}z^2 \right) \\
&= \bar{\sigma}_t \left( e^{\frac{h}{2}}\sqrt{e^h-1}z^1 + \sqrt{e^{2h}-1}z^2 \right) \\
&= \bar{\sigma}_t \left( \sqrt{e^{2h}-e^h}z^1 + \sqrt{e^{2h}-1}z^2 \right),
\end{aligned}$$

which completes the proof.

The proof for SEEDS-3 is straightforward from the proof for SEEDS-2.

### C.4 Proof of Corollary 4.4

#### C.4.1 Convergence of SEEDS-2:

Let $\{\mathbf{y}_{t_i}\}_i$ be the discrete stochastic process defined as follows:

$$\mathbf{y}_{t_0} \leftarrow \mathbf{x}_T, \mathbf{y}_{t_i} \leftarrow \frac{\alpha_{t_i}}{\alpha_{t_{i-1}}}\mathbf{y}_{t_{i-1}} - 2\alpha_{t_i}\int_{\lambda_{t_{i-1}}}^{\lambda_{t_i}} e^{-u}\boldsymbol{\epsilon}(\boldsymbol{u}_i, s_i)\mathrm{d}u - \sqrt{2}\alpha_{t_i}\int_{\lambda_{t_{i-1}}}^{\lambda_{t_i}} e^{-s}\mathrm{d}\bar{\boldsymbol{\omega}}(s),$$

with $s_i \leftarrow t_\lambda \left( \lambda_{t_{i-1}} + \frac{h_i}{2} \right)$ and

$$\boldsymbol{u}_i \leftarrow \frac{\alpha_{s_i}}{\alpha_{t_{i-1}}}\mathbf{y}_{t_{i-1}} - 2\sigma_{s_i}\left( e^{\frac{h_i}{2}} - 1 \right)\boldsymbol{\epsilon}(\mathbf{y}_{t_{i-1}}, t_{i-1}) - \sqrt{2}\alpha_{s_i}\int_{\lambda_{t_{i-1}}}^{\lambda_{s_i}} e^{-s}\mathrm{d}\bar{\boldsymbol{\omega}}(s),$$

Then $\{\mathbf{y}_{t_i}\}_i$ has the same distribution as $\{\widetilde{\mathbf{x}}_{t_i}\}_i$ in Algorithm 3. We can compute the difference:

$$\mathbf{y}_{t_i} - \mathbf{x}_{t_i} = \frac{\alpha_{t_i}}{\alpha_{t_{i-1}}}(\mathbf{y}_{t_{i-1}} - \mathbf{x}_{t_{i-1}}) + \Gamma; \Gamma = \Gamma_1 + \Gamma_2$$

and:

$$\begin{aligned}
\Gamma_1 &= -2\alpha_{t_i}\int_{\lambda_{t_{i-1}}}^{\lambda_{t_i}} e^{-u}\left[ \boldsymbol{\epsilon}(\boldsymbol{u}_i, s_i) - \boldsymbol{\epsilon}(x_{s_i}, s_i) \right]\mathrm{d}u \\
\Gamma_2 &= -2\alpha_{t_i}\int_{\lambda_{t_{i-1}}}^{\lambda_{t_i}} e^{-u}\left[ \boldsymbol{\epsilon}(x_{s_i}, s_i) - \widehat{\boldsymbol{\epsilon}}(\widehat{\mathbf{x}}_u, u) \right]\mathrm{d}u
\end{aligned}$$

Similarly to the case $k = 1$, $\mathbb{E}[|\Gamma_2|] \leq Ch^2$. Note that $\mathbb{E}[|\Gamma_1|] \leq Ch\Delta_i$ with $\Delta_i = \mathbb{E}[|\boldsymbol{u}_i - \mathbf{x}_{s_i}|]$. Introduce: $U_i = \mathbb{E}[|\mathbf{y}_{t_i} - \mathbf{x}_{t_i}|]$. We will bound $\Delta_i$ by a function of $U_{i-1}$. For this, recall:

$$\boldsymbol{u}_i = \frac{\alpha_{s_i}}{\alpha_{t_{i-1}}}\mathbf{y}_{t_{i-1}} - 2\alpha_{s_i}\int_{\lambda_{t_{i-1}}}^{\lambda_{s_i}} e^{-\lambda}\boldsymbol{\epsilon}(\mathbf{y}_{t_{i-1}}, t_{i-1})\mathrm{d}\lambda - \sqrt{2}\alpha_{s_i}\int_{\lambda_{t_{i-1}}}^{\lambda_{s_i}} e^{-\lambda}\mathrm{d}\bar{\boldsymbol{\omega}}(\lambda),$$

and write the difference:

$$\boldsymbol{u}_i - \mathbf{x}_{s_i} = \frac{\alpha_{s_i}}{\alpha_{t_{i-1}}}(\mathbf{y}_{t_{i-1}} - \mathbf{x}_{t_{i-1}}) + 2\alpha_{t_i}\int_{\lambda_{t_{i-1}}}^{\lambda_{t_i}} e^{-u}(\widehat{\boldsymbol{\epsilon}}(\widehat{\mathbf{x}}_u, u) - \boldsymbol{\epsilon}_\theta(\mathbf{y}_{t_{i-1}}, \lambda_{t_{i-1}}))\mathrm{d}\lambda.$$

By the triangular inequality:

$$|\widehat{\boldsymbol{\epsilon}}(\widehat{\mathbf{x}}_u, u) - \boldsymbol{\epsilon}(\mathbf{y}_{t_{i-1}}, \lambda_{t_{i-1}})| \leq C(|\mathbf{x}_{t_\lambda(u)} - \mathbf{x}_{t_{i-1}}| + |\mathbf{x}_{t_{i-1}} - \mathbf{y}_{t_{i-1}}| + |t_\lambda(u) - t_{i-1}|),$$

and so: $\mathbb{E}[|\widehat{\boldsymbol{\epsilon}}(\widehat{\mathbf{x}}_u, u) - \boldsymbol{\epsilon}(\mathbf{y}_{t_{i-1}}, \lambda_{t_{i-1}})|] \leq C(\sqrt{h} + U_{i-1} + h)$. Finally:

$$\Delta_i \leq \frac{\alpha_{s_i}}{\alpha_{t_{i-1}}}U_{i-1} + Ch\left(\sqrt{h} + U_{i-1} + h\right) \leq U_{i-1} + Ch\left(\sqrt{h} + U_{i-1}\right),$$

and $\mathbb{E}[|\Gamma_1|] \leq Ch^{5/2} + C(h + h^2)U_{i-1}$. This gives the bound:

$$U_i \leq Ch^2 + \left(\frac{\alpha_{t_i}}{\alpha_{t_{i-1}}} + Ch + Ch^2\right)U_{i-1}.$$

Now using Theorem C.6, the proof can now be finished following as in proof in Section C.2.4.

### C.4.2  Convergence of SEEDS-3:

Continuing with the same notations as before, we will prove by analogy that:

$$U_i \leq Ch^2 + \left(\frac{\alpha_{t_i}}{\alpha_{t_{i-1}}} + Ch + Ch^2 + Ch^3\right)U_{i-1}, \tag{49}$$

so that, by Theorem C.6, we obtain the desired result.

Let $\mathbf{y}_{t_i} = \widetilde{\mathbf{x}}_{t_i}$. We have:

$$y_{t_i} = \frac{\alpha_{t_i}}{\alpha_{t_{i-1}}}y_{t_{i-1}} - 2\alpha_{t_i}\int_{\lambda_{t_{i-1}}}^{\lambda_{t_i}} e^{-\lambda}\boldsymbol{\epsilon}(y_{t_{i-1}}, t_{i-1})d\lambda$$

$$-\frac{2\alpha_{t_i}}{h_i r_2}\int_{\lambda_{t_{i-1}}}^{\lambda_{t_i}} e^{-\lambda}(\lambda - \lambda_{t_{i-1}})(\boldsymbol{\epsilon}(u_{2i}, s_{2i}) - \boldsymbol{\epsilon}(y_{t_{i-1}}, t_{i-1}))d\lambda + \text{Noise},$$

and $x_{t_i} = \frac{\alpha_{t_i}}{\alpha_{t_{i-1}}}x_{t_{i-1}} - 2\alpha_{t_i}\int_{\lambda_{t_{i-1}}}^{\lambda_{t_i}} e^{-\lambda}\widehat{\boldsymbol{\epsilon}}(\widehat{\mathbf{x}}_\lambda, \lambda)d\lambda + \text{Noise}$, where Noise is the same in both equations. From the proof for $k = 1$, it suffices to bound:

$$\Gamma = \frac{1}{h_i}\int_{\lambda_{t_{i-1}}}^{\lambda_{t_i}} e^{-\lambda}(\lambda - \lambda_{t_{i-1}})(\boldsymbol{\epsilon}(u_{2i}, s_{2i}) - \boldsymbol{\epsilon}(y_{t_{i-1}}, t_{i-1}))d\lambda,$$

in the $L^1$ norm. By the Lipschitz property of $\boldsymbol{\epsilon}$:

$$\begin{aligned}
\mathbb{E}[|\Gamma|] &\leq Ch(\mathbb{E}[|u_{2i} - x_{s_{2i}}|] + \mathbb{E}[|x_{t_{i-1}} - x_{s_{2i}}|] + \mathbb{E}[|y_{t_{i-1}} - x_{t_{i-1}}|]) + Ch^2 \\
&\leq Ch(\mathbb{E}[|u_{2i} - x_{s_{2i}}|] + \sqrt{h} + U_{i-1}) + Ch^2.
\end{aligned}$$

Now, let us bound $\mathbb{E}[|u_{2i} - x_{s_{2i}}|]$ and for this, write

$$u_{2i} = \frac{\alpha_{s_{2i}}}{\alpha_{t_{i-1}}}y_{t_{i-1}} - 2\alpha_{s_{2i}}\int_{\lambda_{t_{i-1}}}^{\lambda_{s_{2i}}} e^{-\lambda}\boldsymbol{\epsilon}(y_{t_{i-1}}, t_{i-1})\mathrm{d}\lambda$$

$$-2\alpha_{s_{2i}}\int_{\lambda_{t_{i-1}}}^{\lambda_{s_{2i}}} e^{-\lambda}(\lambda - \lambda_{t_{i-1}})\frac{\boldsymbol{\epsilon}(u_{2i-1}, s_{2i-1}) - \boldsymbol{\epsilon}(y_{t_{i-1}}, t_{i-1})}{r_1 h_i}\mathrm{d}\lambda + \text{Noise},$$

and

$$x_{s_{2i}} = \frac{\alpha_{s_{2i}}}{\alpha_{t_{i-1}}} x_{t_{i-1}} - 2\alpha_{s_{2i}} \int_{\lambda_{t_{i-1}}}^{\lambda_{s_{2i}}} e^{-\lambda} \widehat{\boldsymbol{\epsilon}}(\widehat{\mathbf{x}}_\lambda, \lambda) \mathrm{d}\lambda + \text{Noise},$$

where Noise is the same in both equations. So

$$
\begin{aligned}
u_{2i} - x_{s_{2i}} &= \frac{\alpha_{s_{2i}}}{\alpha_{t_{i-1}}}(y_{t_{i-1}} - x_{t_{i-1}}) - 2\alpha_{s_{2i}} \int_{\lambda_{t_{i-1}}}^{\lambda_{s_{2i}}} e^{-\lambda}(\boldsymbol{\epsilon}(\mathbf{y}_{t_{i-1}}, t_{i-1}) - \widehat{\boldsymbol{\epsilon}}(\widehat{\mathbf{x}}_\lambda, \lambda)) \mathrm{d}\lambda \\
&\quad - 2\alpha_{s_{2i}} \int_{\lambda_{t_{i-1}}}^{\lambda_{s_{2i}}} e^{-\lambda}(\lambda - \lambda_{t_{i-1}}) D \mathrm{d}\lambda,
\end{aligned}
$$

where $D = \frac{\epsilon(u_{2i-1}, s_{2i-1}) - \epsilon(y_{t_{i-1}}, t_{i-1})}{r_1 h_i}$. From the convergence proof of SEEDS-1, and the Lipschitz property of $\epsilon$, we obtain:

$$\mathbb{E}[|u_{2i} - x_{s_{2i}}|] \leq U_{i-1} + ChU_{i-1} + Ch^2 + Ch\mathbb{E}[|u_{2i-1} - y_{t_{i-1}}|].$$

Again, by the triangular inequality:

$$\mathbb{E}[|u_{2i-1} - y_{t_{i-1}}|] \leq \mathbb{E}[|u_{2i-1} - x_{s_{2i-1}}|] + C\sqrt{h} + U_{i-1},$$

and since

$$u_{2i-1} - x_{s_{2i-1}} = \frac{\alpha_{s_{2i-1}}}{\alpha_{t_{i-1}}}(y_{t_{i-1}} - x_{t_{i-1}}) - 2\alpha_{s_{2i-1}} \int_{\lambda_{t_{i-1}}}^{\lambda_{s_{2i-1}}} e^{-\lambda}(\widehat{\boldsymbol{\epsilon}}(\widehat{\mathbf{x}}_\lambda, \lambda) - \boldsymbol{\epsilon}(y_{t_{i-1}}, t_{i-1})) \mathrm{d}\lambda,$$

we have, as before,

$$\mathbb{E}[|u_{2i-1} - x_{s_{2i}}|] \leq U_{i-1} + ChU_{i-1} + Ch^2.$$

Combining the previous inequalities leads to (49). This finishes the proof of Corollary 4.4.

# D    Implementation Details

The SEEDS solvers used in our experiments are exactly the variants we described in Algorithms 2, 3, 4 in the main part of the paper. In particular, SEEDS-2 and SEEDS-3 solvers are completely determined by one-parameter families of deterministic exponential integrators from [14] of order 2 and 3 respectively and prescribed by the following Butcher tableaux:

$$
\begin{array}{c|c}
0 & \\
c_2 & c_2 \varphi_{1,2} \\
\hline
& \left(1 - \frac{1}{2c_2}\right)\varphi_1 \quad \frac{1}{2c_2}\varphi_1
\end{array}
\qquad
\begin{array}{c|ccc}
0 & 0 & 0 & 0 \\
c_2 & c_2\varphi_{1,2} & 0 & 0 \\
\frac{2}{3} & \frac{2}{3}\varphi_{1,3} - \frac{4}{9c_2}\varphi_{2,3} & \frac{4}{9c_2}\varphi_{2,3} & 0 \\
\hline
& \varphi_1 - \frac{3}{2}\varphi_2 & 0 & \frac{3}{2}\varphi_2
\end{array}
$$

In all experiments we fix the parameter $c_2 = 0.5$ for SEEDS-2 and $c_2 = \frac{1}{3}$ for SEEDS-3. We point out that the Butcher tableau here associated to SEEDS-3 is the result of a weakening of the *stiff order conditions* so it might suffer from order reduction in the deterministic case. We also point out that a full theory of stiff order conditions for stochastic exponential Runge-Kutta methods for semi-linear DEs with a time-varying linear coefficient have not been yet developed to the authors knowledge. As such, this is only an analogy whose purpose is to clarify how our solvers relate to well-known solvers from the literature, but such Butcher tableaux do not rigorously reflect their convergence order.

## D.1    Stabilization of the exponential terms

In the proposed algorithms, one subtle detail is to re-arrange all equations in order for them to make use only of the `expm1`$(h)$ function which computes $e^h - 1$ with great numerical stability, specially

for small values of $h$. We have for values $r_1 = 1/3$, $r_2 = 2/3$, the following identity

$$
\begin{aligned}
\sqrt{e^{2h_i} - e^{h_i}} z_i + \sqrt{e^{h_i} - 1} v_i &= \sqrt{e^{h_i} - 1} \left( \sqrt{e^{h_i}} z_i + v_i \right) \\
&= \sqrt{e^{h_i} - 1} \left( e^{\frac{h_i}{2}} z_i + v_i \right) \\
&= \sqrt{\mathrm{expm1}(h_i)} \left( \left( \mathrm{expm1}\left( \frac{h_i}{2} \right) + 1 \right) z_i + v_i \right) \\
&= \sqrt{\mathrm{expm1}(h_i)} \left( \mathrm{expm1}\left( \frac{h_i}{2} \right) + 1 \right) z_i + \sqrt{\mathrm{expm1}(h_i)} v_i.
\end{aligned}
$$

Now by using

$$
e^{2(r_2 - r_1)h_i} = e^{2\left( \frac{2}{3} - \frac{1}{3} \right) h_i} = e^{2\frac{1}{3} h_i} = e^{r_2 h_i},
$$

we now compute

$$
\begin{aligned}
\sqrt{e^{2r_2 h_i} - e^{r_2 h_i}} z_i^1 + \sqrt{e^{r_2 h_i} - 1} z_i^2 &= \sqrt{e^{r_2 h_i} - 1} \left( \sqrt{e^{r_2 h_i}} z_i^1 + z_i^2 \right) \\
&= \sqrt{e^{r_2 h_i} - 1} \left( e^{\frac{h_i}{3}} z_i^1 + z_i^2 \right) \\
&= \sqrt{\mathrm{expm1}(r_2 h_i)} ((\mathrm{expm1}(r_1 h_i) + 1) z_i^1 + z_i^2)
\end{aligned}
$$

$$
\begin{aligned}
\sqrt{e^{2h_i} - e^{2r_2 h_i}} z_i^1 &= \sqrt{e^{3r_2 h_i} - e^{2r_2 h_i}} z_i^1 \\
&= e^{r_2 h_i} \sqrt{e^{r_2 h_i} - 1} z_i^1 \\
&= \sqrt{\mathrm{expm1}(r_2 h_i)} (\mathrm{expm1}(r_2 h_i) + 1) z_i^1.
\end{aligned}
$$

Finally, we obtain

$$
\begin{aligned}
&\sqrt{e^{2h_i} - e^{2r_2 h_i}} z_i^1 + \sqrt{e^{2r_2 h_i} - e^{r_2 h_i}} z_i^2 + \sqrt{e^{r_2 h_i} - 1} z_i^3 \\
&= \sqrt{\mathrm{expm1}(r_2 h_i)} ((\mathrm{expm1}(r_2 h_i) + 1) z_i^1 + (\mathrm{expm1}(r_1 h_i) + 1) z_i^2 + z_i^3).
\end{aligned}
$$

## D.2 Noise schedules parameterizations

The inverses of $\lambda(t)$ are given in the (VP) linear and cosine schedules respectively by

$$
\begin{aligned}
t_\lambda(\lambda) &= \frac{2 \log(e^{-2\lambda} + 1)}{\sqrt{\beta_{\min}^2 + 2(\beta_{\max} - \beta_{\min}) \log(e^{-2\lambda} + 1)} + \beta_{\min}}, \\
t_\lambda(\lambda) &= \frac{2(1+s)}{\pi} \arccos \left( \exp \left( -\frac{1}{2} \log(e^{-2\lambda} + 1) + \log \cos \left( \frac{\pi s}{2(1+s)} \right) \right) \right) - s.
\end{aligned}
$$

In EDM Noise Prediction case, the inverses of $\lambda(t)$, as given in (37), with respect to the NPFO and NRSDE are respectively given by

$$
t_\lambda(\lambda) = \sigma_d \tan(e^{-\lambda}) \quad \text{and} \quad t_\lambda(\lambda) = \frac{\sigma_d}{\sqrt{\frac{1}{\sigma_d^2 e^{-2\lambda}} - 1}}.
$$

## D.3 EDM discretization

We follow Karras et al. [16] to implement the EDM discretization timesteps $\{t_i\}_{i=0}^M$ as $t_i = \sigma^{-1}(\sigma_i)$ such that, for $\rho > 0$,

$$
\sigma_{i < M} = \left( \sigma_{\max}^{\frac{1}{\rho}} + \frac{i}{N-1} \left( \sigma_{\min}^{\frac{1}{\rho}} - \sigma_{\max}^{\frac{1}{\rho}} \right) \right)^\rho \quad \text{and} \quad \sigma_N = 0.
$$

From the definition, we note that $\sigma_0 = \sigma_{max}$ and $\sigma_{M-1} = \sigma_{min}$, where $\sigma_{min}$ and $\sigma_{max}$ denote the minimum and maximum noise magnitude, respectively. We also keep default value $\rho = 7$ as in [16]. However, we figured out that when using EDM discretization with linear schedule, the noise schedule improvement in iDDPM pre-trained models [26] would result in two consecutive time steps of the same value, i.e. $t_j = t_{j+1}$ for some index $j = 1, \ldots, M-1$ and large steps ($M \geq 61$). Thus, for SEEDS-3 and DPM-Solver-3 [22], the usage of the function $\varphi_2(h) = \dfrac{e^h - h - 1}{h^2}$ (see Appendix E for more details) will cause zero division error where $h = \lambda_{t_{j+1}} - \lambda_{t_j} = 0$. Therefore, in our implementation, we ignore the noise schedule improvement of iDDPM [26] models when using solvers of order three.

### D.4 Final sampling step

The sampling phase in DPMs using SEEDS follows the RSDE, which requires gradual computing through discretization time steps $\{t_i\}_{i=0}^M$ and the latter goes from $t_0 = T$ to $t_M = 0$. In our implementation, to avoid the logarithm of zero error at the last step, i.e. $\log \sigma_{t_M} = \log(0)$, we stop the sampling phase at step $(M-1)$. Hence, the NFE used in a run will be given as

$$\text{NFE} = k \times (M-1),$$

where $k$ represents the order of the solver. We also do not use the "denoising" trick, i.e., ignoring the random noise at the last step and leave it to further research.

## E   Reminders on Stochastic Exponential Integrators

Let us consider a SDE of the form

$$\mathrm{d}\mathbf{x}(t) = [a(t)\mathbf{x}(t) + c(t)f(\mathbf{x}(t), t)]\mathrm{d}t + g(t)\mathrm{d}\boldsymbol{\omega}(t), \tag{50}$$

where $a, c : [0, T] \to \mathbb{R}$ and $g : [0, T] \to \mathbb{R}^{d \times d}$. In other words we concentrate to high-dimensional semi-linear non autonomous SDEs with additive noise. The objective of this section is to construct explicit stochastic exponential derivative-free methods for the above equation following the Runge-Kutta (RK) approach. These methods ideally should fulfill the following properties:

1. If $f \equiv 0$, then (50) can be solved *exactly*;
2. If $g \equiv 0$, then a SEEDS method for (50) identifies with an exponential RK (ERK) method;
3. If $a \equiv 0$, then a SEEDS method for (50) identifies with a stochastic RK (SRK) method and if moreover $g \equiv 0$ then it identifies with a classical RK ODE method.

Before tackling the aimed SEEDS problem let us rapidly recall elementary constructions of RK, ERK, weak and strong SRK methods. We will not deal with time-adaptive methods here.

### E.1   Derivative-free exponential ODE schemes

#### E.1.1   Runge-Kutta approach

Derivative-free schemes are obtained by comparing the Itô-Taylor expansion of the above paragraph with expressions of $\mathbf{x}(t)$ in terms of its intermediate evaluations between $s$ and $t = s + h$ and the Taylor expansions of such evaluations. As a simple example, in the ODE regime

$$\mathrm{d}\mathbf{x}(t) = f(\mathbf{x}(t), t)\mathrm{d}t. \tag{51}$$

As such, analytic solutions to the above equations are of the form

$$\mathbf{x}(t) \;\; = \;\; \mathbf{x}(s) + \int_s^t f(\mathbf{x}(\tau), \tau)\mathrm{d}\tau.$$

Now, Taylor expansion gives, up to order 2:

$$
\begin{aligned}
\mathbf{x}(t) &= \mathbf{x}(s) + h\mathbf{x}'(s) + \frac{h^2}{2}\mathbf{x}''(s) + \mathcal{O}(h^3) \\
&= \mathbf{x}(s) + hf(\mathbf{x}(s), s) + \frac{h^2}{2}(\mathbf{x}'(s)\partial_{\mathbf{x}}f(\mathbf{x}(s), s) + \partial_t f(\mathbf{x}(s), s)) + \mathcal{O}(h^3) \\
&= \mathbf{x}(s) + hf(\mathbf{x}(s), s) + \frac{h^2}{2}(f(\mathbf{x}(s), s)\partial_{\mathbf{x}}f(\mathbf{x}(s), s) + \partial_t f(\mathbf{x}(s), s)) + \mathcal{O}(h^3).
\end{aligned}
$$

A straightforward recursion yields a Taylor expansion

$$
\mathbf{x}(t) = \mathbf{x}(s) + \sum_{k=1}^{n} \frac{h^k}{k!}L_t^{k-1}f(\mathbf{x}(t), t) + \int_s^t \cdots \int_s^{\tau} L_t^n f(\mathbf{x}(\tau), \tau)\mathrm{d}\tau^{n+1},
$$

where we denote the generalized infinitesimal operator of the solution $\mathbf{x}$ of (51) by

$$
L_t(\bullet) = \partial_t(\bullet) + f(\mathbf{x}(t), t) \cdot \partial_{\mathbf{x}}(\bullet).
$$

Derivative-free methods seek to get rid of the derivatives in $L_t$ by computing Taylor expansions of $f$ at well-chosen points. In the explicit one-step case, this amounts of defining

$$
\widehat{\mathbf{x}}(t) = \widehat{\mathbf{x}}(s) + h\sum_{k=1}^{n} \Phi(\widehat{\mathbf{x}}(s), s, h),
$$

where the $\Phi$ function does not contain derivatives of $f$. The general high order case is given by well-tuned coefficients in the following scheme

$$
\begin{aligned}
\widehat{\mathbf{x}}(t) &:= \widehat{\mathbf{x}}(s) + h\sum_{i=1}^{n} \alpha_i f(\mathbf{x}_i, s + c_i h) \\
\mathbf{x}_i &= \widehat{\mathbf{x}}(s) + h\sum_{j=1}^{n} a_{i,j} f(\mathbf{x}_j, s + c_j h).
\end{aligned}
$$

By denoting $\alpha = [\alpha_1 \ldots \alpha_n]$, $c = [c_1 \ldots c_n]^T$ and $A = (a_{i,j})$, these can be represented by a Butcher tableau of the form

$$
\begin{array}{c|c}
c & A \\
\hline
& \alpha
\end{array}
$$

In the following sections we will present extended version of this tableau for representing more involved numerical schemes. As such, the Euler, midpoint, Heun and general second order explicit methods are respectively:

$$
\begin{array}{c|c}
0 & 0 \\
\hline
& 1
\end{array}
\qquad
\begin{array}{c|cc}
0 & 0 & 0 \\
1/2 & 1/2 & 0 \\
\hline
& 0 & 1
\end{array}
\qquad
\begin{array}{c|cc}
0 & 0 & 0 \\
1 & 1 & 0 \\
\hline
& 1/2 & 1/2
\end{array}
\qquad
\begin{array}{c|cc}
0 & 0 & 0 \\
c_2 & c_2 & 0 \\
\hline
& 1 - \frac{1}{2c_2} & \frac{1}{2c_2}
\end{array}
$$

### E.1.2 Exponential Runge-Kutta approach

We now concentrate on a non-autonomous semi-linear ODE of the form

$$
\mathrm{d}\mathbf{x}(t) = [f(t)\mathbf{x}(t) + g(\mathbf{x}(t), t)]\mathrm{d}t, \tag{52}
$$

where $f(t) = \bar{f}(t) \cdot \mathrm{Id}_d$ and $g(\mathbf{x}(t), t) = \bar{g}(\mathbf{x}(t), t) \cdot \mathrm{Id}_d$ are diagonal time-dependent matrices with identical diagonal coefficients. In particular, we have $[f^{(k)}(t), f^{(l)}(s)] = 0$, a property we will need to facilitate exponential matrix multiplications. As $f(s)$ is a $d$-dimensional diagonal matrix, the fundamental matrix for (52), usually given by the Peano-Baker series, simplifies in this case to the form

$$
\Phi(t; s) := e^{\int_s^t f(r)\mathrm{d}r} = \sum_{n=0}^{\infty} \frac{1}{n!}\left(\int_s^t f(r)\mathrm{d}r\right)^n,
$$

which satisfies $\Phi'(t; s) = f(t)\Phi(t; s)$ and $\Phi(s; s) = 1$. Then the exact solution for (52) is given, via the variation of constants formula, by

$$\mathbf{x}(t) = \Phi(t; s)\left(\mathbf{x}(s) + \int_s^t \Phi^{-1}(\tau; s)g(\mathbf{x}(\tau), \tau)\mathrm{d}\tau\right) = \Phi(t; s)\mathbf{x}(s) + \int_s^t \Phi(t; \tau)g(\mathbf{x}(\tau), \tau)\mathrm{d}\tau.$$

In light of this integral form, one can formalize a general class of

exponential $n$-stage RK methods

$$\mathbf{x}_i = \gamma_i(h, f_s)\mathbf{x}(s) + \sum_{j=1}^n a_{ij}(h, f_s)g(\mathbf{x}_j, s + c_j h)$$

$$\mathbf{x}(t) = \gamma_0(h, f_s)\mathbf{x}(s) + \sum_{i=1}^n b_i(h, f_s)g(\mathbf{x}_i, s + c_i h),$$

where $\gamma_0, \gamma_i, a_{ij}, b_j$ are functions of the step-size $h$, $f$ and $f_s(r) := \int_0^r f(s + \tau)\mathrm{d}\tau$. There are two possible approaches to create exponential integrators, namely the exponential time-differencing (ETD) approach that uses the variation of constants formula and makes use of the $\varphi$ functions, and the Lawson approach, also know as integrating factor (IF), which makes a change of variables on the above SDE thus avoiding the use of the $\varphi$ functions but computing exponential factors step-wise.

**The Lawson and the ETD approaches for Exponential Euler schemes**

There are two choices one can make when computing first order approximations of

$$\mathbf{x}(t) = \Phi(t; s)\mathbf{x}(s) + \int_s^t \Phi(t; \tau)g(\mathbf{x}(\tau), \tau)\mathrm{d}\tau.$$

First, by interpolating $g(\mathbf{x}(\tau), \tau)$ as $g(\mathbf{x}(s), s)$ we obtain

$$\mathbf{x}(t) = \Phi(t; s)\mathbf{x}(s) + g(\mathbf{x}(s), s)\int_s^t \Phi(t; \tau)\mathrm{d}\tau.$$

Now two choices remain. Either $\int_s^t \Phi(t; \tau)\mathrm{d}\tau$ is computed exactly or we again interpolate $\Phi(t; \tau)$ as $\Phi(t; s)$. Taking for simplicity $f \equiv A$ to be constant and by denoting $h = t - s$, the first case yields the ETD Euler method

$$\begin{aligned}
\widehat{\mathbf{x}}(t) &= \Phi(t; s)\widehat{\mathbf{x}}(s) + h\gamma_1(f, t, s)g(\widehat{\mathbf{x}}(s), s) \\
&= e^{Ah}\widehat{\mathbf{x}}(s) + h\varphi_1(Ah)g(\widehat{\mathbf{x}}(s), s) \\
&= \widehat{\mathbf{x}}(s) + h\varphi_1(Ah)[g(\widehat{\mathbf{x}}(s), s) - \widehat{\mathbf{x}}(s)],
\end{aligned}$$

and the second yields the IF Euler (also called Lawson-Euler) method

$$\begin{aligned}
\widehat{\mathbf{x}}(t) &= \Phi(t; s)\widehat{\mathbf{x}}(s) + h\Phi(t; s)g(\widehat{\mathbf{x}}(s), s) \\
&= e^{Ah}(\widehat{\mathbf{x}}(s) + hg(\widehat{\mathbf{x}}(s), s)).
\end{aligned}$$

Now, consider a solution

$$\mathbf{x}(t) = \Phi(t; s)\mathbf{x}(s) + \int_0^h \Phi(h; \tau)g(\mathbf{x}(s + \tau), s + \tau)\mathrm{d}\tau.$$

Exponential methods then aim to approximate the term $g(\mathbf{x}(s + \tau), s + \tau)$ by its interpolation polynomial in certain non-confluent quadrature nodes $c_1, \ldots, c_n$.

**The ETD approach**

In this case, the variation of constants formula yields

$$\begin{aligned}
\mathbf{x}(s + ch) &= e^{\int_s^{s+ch} a(\tau)\mathrm{d}\tau}\mathbf{x}(s) + \int_s^{s+ch} e^{\int_\tau^{s+ch} a(r)\mathrm{d}r} f(\mathbf{x}(\tau), \tau)\mathrm{d}\tau. \\
&= e^{\int_0^{ch} a(s+\tau)\mathrm{d}\tau}\mathbf{x}(t) + \int_0^{ch} e^{\int_{\tau-s}^{ch} a(s+r)\mathrm{d}r} f(\mathbf{x}(s + \tau), s + \tau)\mathrm{d}\tau.
\end{aligned}$$

Now, as before, the Taylor expansion of $f$ yields

$$f(\mathbf{x}(s+\tau), s+\tau) = \sum_{j=1}^{q} \frac{\tau^{j-1}}{(j-1)!} f^{(j-1)}(\mathbf{x}(s), s)$$

$$+ \int_0^\tau \frac{(\tau-\tau_1)^{q-1}}{(q-1)!} f^{(q)}(\mathbf{x}(s+\tau_1), s+\tau_1) d\tau_1.$$

Recall that the $\varphi$ functions are given in an integral form as follows

$$\varphi_{k+1}(t) = \int_0^1 e^{(1-\delta)t} \frac{\delta^k}{k!} d\delta,$$

which satisfy $\varphi_k(0) = \frac{1}{k!}$. Now denote

$$\varphi_j(t, a) := \frac{1}{t^j} \int_0^t e^{\int_\tau^t a(r)dr} \frac{\tau^{j-1}}{(j-1)!} d\tau, \qquad j \geq 1.$$

$$\varphi_1(h, a) = \frac{1}{h} \int_0^h e^{\int_\tau^h a(r)dr} d\tau = \int_0^1 e^{h \int_\theta^1 a(r)dr} d\theta.$$

The exact solution at $s + ch$ now reads

$$\mathbf{x}(s+ch) = e^{\int_0^{ch} a(s+\tau)d\tau} \mathbf{x}(s) + \sum_{j=1}^{q} (ch)^j \varphi_j(ch, a) f^{(j-1)}(\mathbf{x}(s), s)$$

$$+ \int_0^{ch} e^{\int_\tau^{ch} a(s+r)dr} \left( \int_0^\tau \frac{(\tau-\tau_1)^{q-1}}{(q-1)!} f^{(q)}(\mathbf{x}(s+\tau_1), s+\tau_1) ds \right) d\tau_1.$$

Using the left endpoint rule yields

$$\mathbf{x}(t) = e^{\int_0^h a(s+\tau)d\tau} \mathbf{x}(s) + f(\mathbf{x}(s), s) \int_0^h e^{\int_\tau^h a(s+r)dr} d\tau + \mathcal{O}(h^2)$$

$$= e^{\int_0^h a(s+\tau)d\tau} \mathbf{x}(s) + h\varphi_1(h, a) f(\mathbf{x}(s), s) + \mathcal{O}(h^2).$$

**Second order examples:**

The condition $b_2(z)c_2 = \varphi_2(z)$ implies $b_2 = \varphi_2(z)/c_2$ and we obtain

$$\widehat{\mathbf{x}} = e^{chA}\mathbf{x}(s) + ch\varphi_1(chA)f(\mathbf{x}(s), s)$$

$$\mathbf{x}(t) = e^{hA}\mathbf{x}(s) + h\left(\varphi_1(hA) - \frac{1}{c}\varphi_2(hA)\right) f(\mathbf{x}(s), s) + \frac{h}{c}\varphi_2(hA)f(\widehat{\mathbf{x}}, s+ch). \quad (53)$$

A second method is obtained by weakening the above condition to $b_2(0)c_2 = \varphi_2(0) = 1/2$ giving

$$\widehat{\mathbf{x}} = e^{chA}\mathbf{x}(s) + ch\varphi_1(chA)f(\mathbf{x}(s), s)$$

$$\mathbf{x}(t) = e^{hA}\mathbf{x}(s) + h\varphi_1(hA)\left(1 - \frac{1}{2c}\right) f(\mathbf{x}(s), s) + \frac{h}{2c}\varphi_1(hA)f(\widehat{\mathbf{x}}, s+ch). \quad (54)$$

In some cases this method can suffer from order reduction and not reach order 2 of convergence. Moreover, setting $c = 1/2$ gives the exponential midpoint method:

$$\widehat{\mathbf{x}} = e^{chA}\mathbf{x}(s) + ch\varphi_1(chA)f(\mathbf{x}(s), s)$$

$$\mathbf{x}(t) = e^{hA}\mathbf{x}(s) + h\varphi_1(hA)f(\widehat{\mathbf{x}}, s+ch). \quad (55)$$

The above order 1 and 2 exponential methods are represented by the following *exponential* Butcher tableaux

| $0$ | $0$ |
|---|---|
| | $\varphi_1$ |

| $0$ | | |
|---|---|---|
| $c_2$ | $c_2\varphi_{1,2}$ | |
| | $\varphi_1 - \frac{1}{c_2}\varphi_2$ | $\frac{1}{c_2}\varphi_2$ |

| $0$ | | |
|---|---|---|
| $c_2$ | $c_2\varphi_{1,2}$ | |
| | $\left(1 - \frac{1}{2c_2}\right)\varphi_1$ | $\frac{1}{2c_2}\varphi_1$ |

| $0$ | | |
|---|---|---|
| $c_2$ | $c_2\varphi_{1,2}$ | |
| | $0$ | $\varphi_1$ |

We check that when formally setting $A = 0$ then the first two methods are identical and give the generic 2nd order RK method, the choice $c = 1$ gives the Heun method and the choice $c = 1/2$ gives the midpoint method.

**Fourth order methods**

It can be shown that ERK methods need at least 5 stages to achieve order 4. By setting formally $A = 0$ these methods do not have a non-exponential counterpart to our knowledge.

**5-stage sequential** We have a fourth-order ERK scheme given by

$$
\begin{array}{c|ccccc}
0 \\
\frac{1}{2} & \frac{1}{2}\varphi_{1,2} \\
\frac{1}{2} & \frac{1}{2}\varphi_{1,3} - \varphi_{2,3} & \varphi_{2,3} \\
1 & \varphi_{1,4} - 2\varphi_{2,4} & \varphi_{2,4} & \varphi_{2,4} \\
\frac{1}{2} & \frac{1}{2}\varphi_{1,5} - 2a_{5,2} - a_{5,4} & a_{5,2} & a_{5,2} & \frac{1}{4}\varphi_{2,5} - a_{5,2} \\
\hline
& \varphi_1 - 3\varphi_2 + 4\varphi_3 & 0 & 0 & -\varphi_2 + 4\varphi_3 & 4\varphi_2 - 8\varphi_3
\end{array}
$$

with

$$
a_{5,2} = c_5\varphi_{2,5} - \varphi_{3,4} + c_5^2\varphi_{2,4} - c_5\varphi_{3,5}.
$$

This can also be represented by Rosenbrock-like Butcher tableau

$$
\begin{array}{c|ccccc}
0 \\
\frac{1}{2} & \frac{1}{2}\varphi_{1,2} \\
\frac{1}{2} & \frac{1}{2}\varphi_{1,3} & \varphi_{2,3} \\
1 & \varphi_1 & \varphi_2 & \varphi_2 \\
\frac{1}{2} & \frac{1}{2}\varphi_{1,5} & a_{5,2} & a_{5,2} & \frac{1}{4}\varphi_{2,5} - a_{5,2} \\
\hline
& \varphi_1 & 0 & 0 & 4\varphi_3 - \varphi_2 & 4\varphi_2 - 8\varphi_3
\end{array}
$$

with

$$
a_{5,2} = c_5\varphi_{2,5} - c_3\varphi_{3,3} + c_5^2\varphi_2 - \varphi_3.
$$

This traduces into (recall that $D_i = f(\mathbf{x}_i, s + c_i h) - f(\mathbf{x}(s), s)$)

$$
\begin{aligned}
\mathbf{x}(t) &= \mathbf{x}(s) + h(\varphi_1(hA))F(\mathbf{x}(s), s) + h(4\varphi_3(hA) - \varphi_2(hA))D_4 \\
&\quad + h(4\varphi_2(hA) - 8\varphi_3(hA))D_5 \\
\mathbf{x}_2 &= \mathbf{x}(s) + c_2 h\varphi_1(c_2 hA)F(\mathbf{x}(s), s) \\
\mathbf{x}_3 &= \mathbf{x}(s) + hc_3\varphi_1(c_3 hA)F(\mathbf{x}(s), s) + h\varphi_2(c_3 hA)D_2 \\
\mathbf{x}_4 &= \mathbf{x}(s) + h\varphi_1(hA)F(\mathbf{x}(s), s) + h\varphi_2(hA)(D_2 + D_3) \\
\mathbf{x}_5 &= \mathbf{x}(s) + h\varphi_1(hA)F(\mathbf{x}(s), s) + ha_{5,2}(hA)(D_2 + D_3) + h(c_5^2\varphi_2(c_5 hA) - a_{5,2}(hA))D_4
\end{aligned}
$$

Inspired by this, we deduce the following fourth order Algorithm 7 specialized for DPMs in the VP noise prediction regime.

## E.2   Derivative-free exponential SDE schemes

Let $\mathbf{x}_t$ be the path at the continuous limit $h \to 0$, and $\{\widehat{\mathbf{x}}_t\}_{t_0}^{t_M}$ be the discretized numerical path, computed by a numerical scheme $\mathcal{S}$ with $M = 1/h$ steps of length $h > 0$. Then, $\mathcal{S}$ has

1. *strong order of convergence* $\gamma$ if there is $K > 0$ such that

$$
\mathbb{E}[|\mathbf{x}_{t_M} - \widehat{\mathbf{x}}_{t_M}|] \leq Kh^\gamma, \tag{56}
$$

2. *weak order of convergence* $\beta$ if there is $K > 0$ and a function class $\mathcal{K}$ such that

$$
|\mathbb{E}[\phi(\mathbf{x}_{t_M})] - \mathbb{E}[\phi(\widehat{\mathbf{x}}_{t_M})]| \leq Kh^\beta, \qquad \forall\phi(\cdot) \in \mathcal{K}. \tag{57}
$$

Strong convergence is concerned with the precision of the path, while the weak convergence is with the precision of the moments. As, for diffusion models, the center of attention is the evolution of the probability densities rather than that of the noising process of single data samples, weak convergence is enough to guarantee the well-conditioning of our numerical schemes. Moreover, when the diffusion coefficient vanishes, then both strong and weak convergence (with the choice $\phi = \mathrm{id}$) reduce to the usual deterministic convergence criterion for ODEs.

---

**Algorithm 7** DPM-Solver-4

1: **def** DPM-Solver-4$(\epsilon_\theta, \tilde{\mathbf{x}}_{t_{i-1}}, t_{i-1}, t_i, r = 0.5)$:

2: $\quad h_i \leftarrow \lambda_{t_i} - \lambda_{t_{i-1}}$

3: $\quad s_2 \leftarrow t_\lambda\left(\lambda_{t_{i-1}} + rh_i\right), \quad s_3 \leftarrow t_\lambda\left(\lambda_{t_{i-1}} + rh_i\right)$

4: $\quad s_4 \leftarrow t_\lambda\left(\lambda_{t_{i-1}} + h_i\right), \quad s_5 \leftarrow t_\lambda\left(\lambda_{t_{i-1}} + rh_i\right)$

5: $\quad \boldsymbol{k_1} \leftarrow \epsilon_\theta(\tilde{\mathbf{x}}_{t_{i-1}}, t_{i-1})$

6: $\quad \boldsymbol{k_2} \leftarrow \frac{\alpha_{s_2}}{\alpha_s}\tilde{\mathbf{x}}_{t_{i-1}} - \sigma_{s_2}(e^{rh} - 1)\boldsymbol{k_1}$

7: $\quad \boldsymbol{k_3} \leftarrow \frac{\alpha_{s_3}}{\alpha_s}\tilde{\mathbf{x}}_{t_{i-1}} - \sigma_{s_3}(e^{rh} - 1)\boldsymbol{k_1} - \sigma_{s_3}\left(4\frac{e^{rh}-1}{h} - 2\right)\left[\epsilon_\theta(\boldsymbol{k_2}, s_2) - \boldsymbol{k_1}\right]$

8: $\quad \boldsymbol{k_4} \leftarrow \frac{\alpha_{s_4}}{\alpha_s}\tilde{\mathbf{x}}_{t_{i-1}} - \sigma_{s_4}(e^h - 1)\boldsymbol{k_1} - \sigma_{s_4}\left(\frac{e^h-1}{h} - 1\right)\left[\epsilon_\theta(\boldsymbol{k_3}, s_3) + \epsilon_\theta(\boldsymbol{k_2}, s_2) - 2\boldsymbol{k_1}\right]$

9: $\quad A = \sigma_{s_5}(e^{rh} - 1)\boldsymbol{k_1} - \frac{1}{4}\sigma_{s_5}\left(\frac{e^h-1}{h} - 1\right)\left[\boldsymbol{k_1} + \epsilon_\theta(\boldsymbol{k_2}, s_2) + \epsilon_\theta(\boldsymbol{k_3}, s_3)\right]$

10: $\quad B = \sigma_{s_5}\left(\frac{e^{rh}-1}{h} - \frac{1}{2}\right)\left[\boldsymbol{k_1} + 4\epsilon_\theta(\boldsymbol{k_2}, s_2) + 4\epsilon_\theta(\boldsymbol{k_3}, s_3) - \epsilon_\theta(\boldsymbol{k_4}, s_4)\right]$

11: $\quad C = \sigma_{s_5}\left(\frac{e^h-1+4(e^{rh}-1)-3h}{h^2} - 1\right)\left[-\boldsymbol{k_1} - \epsilon_\theta(\boldsymbol{k_2}, s_2) - \epsilon_\theta(\boldsymbol{k_3}, s_3) + \epsilon_\theta(\boldsymbol{k_4}, s_4)\right]$

12: $\quad \boldsymbol{k_5} \leftarrow \frac{\alpha_{s_5}}{\alpha_{t_{i-1}}}\tilde{\mathbf{x}}_{t_{i-1}} - A - B - C$

13: $\quad D = \sigma_t(e^h - 1)\boldsymbol{k_1} - \sigma_t\left(\frac{e^h-1}{h} - 1\right)\left[4\epsilon_\theta(\boldsymbol{k_5}, s_5) - \epsilon_\theta(\boldsymbol{k_4}, s_4) - 3\boldsymbol{k_1}\right]$

14: $\quad E = \sigma_t\left(4\frac{e^h-1-h}{h^2} - 2\right)\left[\boldsymbol{k_1} + \epsilon_\theta(\boldsymbol{k_4}, s_4) - 2\epsilon_\theta(\boldsymbol{k_5}, s_5)\right]$

15: $\quad \tilde{\mathbf{x}}_{t_i} \leftarrow \frac{\alpha_t}{\alpha_{t_{i-1}}}\tilde{\mathbf{x}}_{t_{i-1}} - D - E$

16: $\quad$ Return $\mathbf{x}_{t_i}$

---

### E.2.1 Strong and Weak Stochastic Runge-Kutta approach

In all what follows $\omega$ is considered a $d$-dimensional Wiener process (with identity diffusion matrix).

Consider the following SDE

$$d\mathbf{x}(t) = f(\mathbf{x}(t), t)dt + g(t)d\omega(t),$$

where $g(t) = \hat{g}(t) \cdot \mathrm{Id}_d$ is considered here as a diagonal matrix with identical diagonal entries $\hat{g}(t)$. Given an initial value independent of $\omega$, the integral form of $\mathbf{x}(t)$ is given by

$$\mathbf{x}(t) = \mathbf{x}(s) + \int_s^t f(\mathbf{x}(\tau), \tau)d\tau + \int_s^t g(\tau)d\omega(\tau).$$

The idea underlying stochastic numerical schemes is very similar to the one used in the deterministic approach, that is to take expansions of the terms inside the integrals based at the integral's initial value and replace the obtained derivatives that appear by interpolated approximations. A key difference here is that as we have to consider Itô-Taylor expansions, the infinitesimal operators are different but most importantly most of the stochastic iterated integrals will need to be approximated in an appropriate sense, whenever that is possible. We will develop this expansion up to triple integrals.

### E.2.2 Truncated Itô-Taylor expansions

Applying Itô formula to $h = f$ or $g$ yields

$$h(\mathbf{x}(t), t) = h(\mathbf{x}(s), s) + \int_s^t g(\tau) \cdot \partial_\mathbf{x} h(\mathbf{x}(\tau), \tau)d\omega(\tau)$$

$$+ \int_s^t \left(\partial_t h(\mathbf{x}(\tau), \tau) + f(\mathbf{x}(\tau), \tau) \cdot \partial_\mathbf{x} h(\mathbf{x}(\tau), \tau) + \frac{g^2(\tau)}{2}\partial_{\mathbf{x}^2}^2 h(\mathbf{x}(\tau), \tau)\right)d\tau.$$

This allows us to define two differential operators $L, G$ as

$$L_t = \partial_t + f(\mathbf{x}(t), t) \cdot \partial_\mathbf{x} + \frac{g^2(t)}{2} \cdot \partial_\mathbf{x}^2 \tag{58}$$

$$G_t = g(t) \cdot \partial_\mathbf{x}. \tag{59}$$

In particular $L_t g(t) = \partial_t g(t)$ and $G_t g(t) = 0$. With this notation, we have

$$h(\mathbf{x}(t), t) = h(\mathbf{x}(s), s) + \int_s^t L_t h(\mathbf{x}(\tau), \tau)\mathrm{d}\tau + \int_s^t G_t h(\mathbf{x}(\tau), \tau)\mathrm{d}\omega(\tau),$$

so our solution reads

$$
\begin{aligned}
\mathbf{x}(t) &= \mathbf{x}(s) + \int_s^t f(\mathbf{x}(\tau_1), \tau_1)\mathrm{d}\tau_1 + \int_s^t g(\tau_1)\mathrm{d}\omega(\tau_1) \\
&= \mathbf{x}(s) + \int_s^t \left( f(\mathbf{x}(t), t) + \int_s^{\tau_1} L_t f(\mathbf{x}(\tau_2), \tau_2)\mathrm{d}\tau_2 + \int_s^{\tau_1} G_t f(\mathbf{x}(\tau_2), \tau_2)\mathrm{d}\omega(\tau_2) \right) \mathrm{d}\tau_1 \\
&\quad + \int_s^t \left( g(t) + \int_s^{\tau_1} L_t g(\tau_2)\mathrm{d}\tau_2 + \int_s^{\tau} G_t g(\tau_2)\mathrm{d}\omega(\tau_2) \right) \mathrm{d}\omega(\tau_1) \\
&= \mathbf{x}(s) + f(\mathbf{x}(s), s)h + g(t)(\omega(t) - \omega(s)) + R_1.
\end{aligned}
$$

Now $G_t g(\tau_2) = 0$ and

$$g(s) + \int_s^{\tau_1} L_t g(\tau_2)\mathrm{d}\tau_2 = g(s) + \int_s^{\tau_1} \partial_t g(\tau_2)\mathrm{d}\tau_2 = g(s) + g(\tau_1) - g(s) = g(\tau_1).$$

So we have

$$\int_s^t \left( g(s) + \int_s^{\tau_1} L_t g(\tau_2)\mathrm{d}\tau_2 + \int_s^{\tau} G_t g(\tau_2)\mathrm{d}\omega(\tau_2) \right) \mathrm{d}\omega(\tau_1) = g(s)(\omega(t) - \omega(s)),$$

and now our solution reads

$$\mathbf{x}(t) = \mathbf{x}(s) + f(\mathbf{x}(s), s)h + g(s)(\omega(t) - \omega(s)) + R_1,$$

where

$$R_1 = \int_s^t \int_s^{\tau_1} L_t f(\mathbf{x}(\tau_2), \tau_2)\mathrm{d}\tau_2 \mathrm{d}\tau_1 + \int_s^t \int_s^{\tau_1} G_t f(\mathbf{x}(\tau_2), \tau_2)\mathrm{d}\omega(\tau_2)\mathrm{d}\tau_1.$$

Now we have

$$
\begin{aligned}
& L_t f(\mathbf{x}(t), t) \\
=\ & L_t f(\mathbf{x}(s), s) + \int_s^t g(\mathbf{x}(\tau), \tau) \cdot \partial_{\mathbf{x}} L_t f(\mathbf{x}(\tau), \tau)\mathrm{d}\omega(\tau) \\
& + \int_s^t \left( \partial_t L_t f(\mathbf{x}(\tau), \tau) + f(\mathbf{x}(\tau), \tau) \cdot \partial_{\mathbf{x}} L_t f(\mathbf{x}(\tau), \tau) + \frac{g^2(\mathbf{x}(\tau), \tau)}{2} \partial_{\mathbf{x}^2}^2 L_t f(\mathbf{x}(\tau), \tau) \right) \mathrm{d}\tau \\
=\ & L_t f(\mathbf{x}(s), s) + \int_s^t L_t^2 f(\mathbf{x}(\tau), \tau)\mathrm{d}\tau + \int_s^t G_t L_t f(\mathbf{x}(\tau), \tau)\mathrm{d}\omega(\tau),
\end{aligned}
$$

and

$$
\begin{aligned}
& G_t f(\mathbf{x}(t), t) \\
=\ & G_t f(\mathbf{x}(s), s) + \int_s^t g(\mathbf{x}(\tau), \tau) \cdot \partial_{\mathbf{x}} G_t f(\mathbf{x}(\tau), \tau)\mathrm{d}\omega(\tau) + \\
& \int_s^t \left( \partial_t G_t f(\mathbf{x}(\tau), \tau) + f(\mathbf{x}(\tau), \tau) \cdot \partial_{\mathbf{x}} G_t f(\mathbf{x}(\tau), \tau) + \frac{g^2(\mathbf{x}(\tau), \tau)}{2} \partial_{\mathbf{x}^2}^2 G_t f(\mathbf{x}(\tau), \tau) \right) \mathrm{d}\tau \\
=\ & G_t f(\mathbf{x}(s), s) + \int_s^t L_t G_t f(\mathbf{x}(\tau), \tau)\mathrm{d}\tau + \int_s^t G_t^2 f(\mathbf{x}(\tau), \tau)\mathrm{d}\omega(\tau).
\end{aligned}
$$

Now, if we denote $\mathrm{d}\omega^0(\tau) = \mathrm{d}\tau$, $\mathrm{d}\omega^1(\tau) = \mathrm{d}\omega(\tau)$ and

$$I_{(i)} = \int_s^t \mathrm{d}\omega^i(\tau_1) \qquad I_{(i,j)} = \int_s^t \int_s^{\tau_1} \mathrm{d}\omega^j(\tau_2)\mathrm{d}\omega^i(\tau_1) \qquad i, j \in \{0, 1\},$$

applying the same procedure to $R_1$ leads to

$$R_1$$

$$= \int_s^t \int_s^{\tau_1} L_t f(\mathbf{x}(\tau_2), \tau_2) \mathrm{d}\tau_2 \mathrm{d}\tau_1 + \int_s^t \int_s^{\tau_1} G_t f(\mathbf{x}(\tau_2), \tau_2) \mathrm{d}\omega(\tau_2) \mathrm{d}\tau_1$$

$$= \int_s^t \int_s^{\tau_1} \left[ L_t f(\mathbf{x}(t), t) + \int_s^{\tau_2} L_t^2 f(\mathbf{x}(\tau_3), \tau_3) \mathrm{d}\tau_3 + \int_s^{\tau_2} G_t L_t f(\mathbf{x}(\tau_3), \tau_3) \mathrm{d}\omega(\tau_3) \right] \mathrm{d}\tau_2 \tau_1$$

$$+ \int_s^t \int_s^{\tau_1} \left[ G_t f(\mathbf{x}(t), t) + \int_s^{\tau_2} L_t G_t f(\mathbf{x}(\tau_3), \tau_3) \mathrm{d}\tau + \int_s^{\tau_2} G_t^2 f(\mathbf{x}(\tau_3), \tau_3) \mathrm{d}\omega(\tau_3) \right] \mathrm{d}\omega(\tau_2) \mathrm{d}\tau_1$$

$$= L_t f(\mathbf{x}(t), t) \int_s^t \int_s^{\tau_1} \mathrm{d}\tau_2 \mathrm{d}\tau_1 + G_t f(\mathbf{x}(t), t) \int_s^t \int_s^{\tau_1} \mathrm{d}\omega(\tau_2) \mathrm{d}\tau_1$$

$$+ \int_s^t \int_s^{\tau_1} \int_s^{\tau_2} L_t^2 f(\mathbf{x}(\tau_3), \tau_3) \mathrm{d}\tau_3 \mathrm{d}\tau_2 \mathrm{d}\tau_1 + \int_s^t \int_s^{\tau_1} \int_s^{\tau_2} G_t L_t f(\mathbf{x}(\tau_3), \tau_3) \mathrm{d}\omega(\tau_3) \mathrm{d}\tau_2 \mathrm{d}\tau_1$$

$$+ \int_s^t \int_s^{\tau_1} \int_s^{\tau_2} L_t G_t f(\mathbf{x}(\tau_3), \tau_3) \mathrm{d}\tau_3 \mathrm{d}\omega(\tau_2) \mathrm{d}\tau_1$$

$$+ \int_s^t \int_s^{\tau_1} \int_s^{\tau_2} G_t^2 f(\mathbf{x}(\tau_3), \tau_3) \mathrm{d}\omega(\tau_3) \mathrm{d}\omega(\tau_2) \mathrm{d}\tau_1$$

$$= L_t f(\mathbf{x}(s), s) I_{(0,0)} + G_t f(\mathbf{x}(s), s) I_{(0,1)} + L_t^2 f(\mathbf{x}(s), s) I_{(0,0,0)} + G_t L_t f(\mathbf{x}(s), s) I_{(0,0,1)}$$

$$+ L_t G_t f(\mathbf{x}(s), s) I_{(0,1,0)} + G_t^2 f(\mathbf{x}(s), s) I_{(1,1,0)} + R_2,$$

and with $R_2$ consisting on quadruple integrals. As such, our solution now reads

$$\mathbf{x}(t) = \mathbf{x}(s) + f(\mathbf{x}(s), s) h + g(s)(\omega(t) - \omega(s)) + L_t f(\mathbf{x}(s), s) I_{(0,0)} + G_t f(\mathbf{x}(s), s) I_{(0,1)}$$

$$+ L_t^2 f(\mathbf{x}(s), s) I_{(0,0,0)} + G_t L_t f(\mathbf{x}(s), s) I_{(0,0,1)}$$

$$+ L_t G_t f(\mathbf{x}(s), s) I_{(0,1,0)} + G_t^2 f(\mathbf{x}(s), s) I_{(1,1,0)} + R_2.$$

Now, in the SDE regime, one cannot get rid of the iterated Itô integral and so stochastic RK methods cannot be derived as simple extensions of their deterministic counterparts. In order to continue we now take into account the fact that the diffusion SDE has additive and diagonal noise. In this case, both the Itô and the Stratonovich SDE coincide.

**Iterated integrals**

Now, for simplicity, set $t = 0$. We then have $I_{(0)} = h$, $I_{(0,0)} = \int_0^h \int_0^{\tau_1} \mathrm{d}\tau_2 \mathrm{d}\tau_1 = \frac{h^2}{2}$, $I_{(0,0,0)} = \frac{h^3}{6}$. Now notice that

$$I_{(1)} := \hat{w}_h \sim \mathcal{N}(0, h)$$

$$I_{(0,1)} := \hat{z}_h := \int_0^h \int_0^{\tau_1} \mathrm{d}\omega(\tau_2) \mathrm{d}\tau_1 = \lim_{n \to \infty} \frac{h}{n} \sum_{i=0}^{n-1} \sum_{j=1}^{i} \epsilon_j, \qquad \epsilon_j \sim \mathcal{N}\left(0, \frac{h}{n}\right).$$

Additionally, $\hat{w}_h$ and $\hat{z}_h$ satisfy $\mathbb{E}[\hat{w}_h^2] = h$,

$$\mathbb{E}[(\hat{w}_h h - \hat{z}_h)^2] = \mathbb{E}\left[ \left( \int_0^h \tau \mathrm{d}\omega(\tau) \right)^2 \right] = \frac{1}{3} h^3$$

$$\mathbb{E}[\hat{w}_h \hat{z}_h] = \mathbb{E}\left[ \hat{w}_h \int_0^h \tau \mathrm{d}\omega(\tau) \right] = \mathbb{E}\left[ \int_0^h \tau \mathrm{d}\tau \right] = \frac{1}{2} h^2$$

$$\mathbb{E}[\hat{z}_h^2] = \mathbb{E}[(\hat{w}_h h - \hat{z}_h)^2 - h^2 \hat{w}_h^2 + 2h \hat{w}_h \hat{z}_h] = \frac{1}{3} h^3.$$

### E.2.3 Integral approximations

**Weak Approximations**

When crafting weak stochastic approximations to SDEs one may replace multiple Itô integrals by other random variables satisfying the corresponding moment conditions. We will denote $\hat{I}_\alpha$ the approximation of $I_\alpha$ for $\alpha$ a multi-index following [17, Corollary 5.12.1]. Of course, the deterministic integrals $I_{(0,\ldots,0)}$ need not to be approximated.

**First order approximations**

The random variable $\hat{I}_{(1)}$ must satisfy for some constant $K$:

$$|\mathbb{E}[\hat{I}_{(1)}]| + |\mathbb{E}[(\hat{I}_{(1)})^3]| + |\mathbb{E}[(\hat{I}_{(1)})^2 - h]| \leqslant Kh^2$$

Two possible choices for $\hat{I}_{(1)}$ are either $\hat{I}_{(1)} \sim \mathcal{N}(0, h)$ or $\hat{I}_{(1)}$ is a two-pointed distributed discrete random variable with

$$\mathbb{P}\left[\hat{I}_{(1)} = \pm\sqrt{h}\right] = \frac{1}{2}.$$

**Second order approximations**

The random variable $\hat{I}_{(1)}$ must satisfy for some constant $K$:

$$|\mathbb{E}[\hat{I}_{(1)}]| + |\mathbb{E}[(\hat{I}_{(1)})^3]| + |\mathbb{E}[(\hat{I}_{(1)})^5]| + |\mathbb{E}[(\hat{I}_{(1)})^2 - h]| + |\mathbb{E}[(\hat{I}_{(1)})^4 - 3h^2]| \leqslant Kh^2.$$

Two possible choices for $\hat{I}_{(1)}$ are either $\hat{I}_{(1)} \sim \mathcal{N}(0, h)$ or $\hat{I}_{(1)}$ is a three-pointed distributed random variable with

$$\mathbb{P}\left[\hat{I}_{(1)} = \pm\sqrt{3h}\right] = \frac{1}{6}, \qquad \mathbb{P}[\hat{I}_{(1)} = 0] = \frac{2}{3}.$$

The rest follows from the above calculations:

$$\hat{I}_{(0,1)} = \frac{1}{2}h\hat{I}_{(1)}, \qquad i = 0, 1.$$

**Third order approximations**

One can choose $\hat{I}_{(1)} \sim \mathcal{N}(0, h)$, $\hat{I}_{(0,1)} \sim \mathcal{N}\left(0, \frac{1}{3}h^3\right)$ satisfying $\mathbb{E}[\hat{I}_{(1)}\hat{I}_{(0,1)}] = \frac{1}{2}h^2$.

Then, one can deduce the following:

$$\hat{I}_{(1,0,0)} = \hat{I}_{(0,1,0)} = \hat{I}_{(0,0,1)} = \frac{1}{6}h^2\hat{I}_{(1)}$$

$$\hat{I}_{(0,1,1)} = \hat{I}_{(1,0,1)} = \hat{I}_{(1,1,0)} = \frac{1}{6}h(\hat{I}_{(1)}^2 - h).$$

Thus, we can write the solution weak approximation as

$$
\begin{aligned}
\mathbf{x}(t) = {} & \mathbf{x}(s) + f(\mathbf{x}(s), s)h + g(s)\hat{I}_{(1)} + L_t f(\mathbf{x}(s), s)\frac{h^2}{2} + R_2 \\
& + G_t f(\mathbf{x}(s), s)\hat{I}_{(0,1)} + L_t^2 f(\mathbf{x}(s), s)\frac{h^3}{6} + G_t L_t f(\mathbf{x}(s), s)\frac{1}{6}h^2\hat{I}_{(1)} \\
& + L_t G_t f(\mathbf{x}(s), s)\frac{1}{6}h^2\hat{I}_{(1)} + G_t^2 f(\mathbf{x}(s), s)\frac{1}{6}h(\hat{I}_{(1)}^2 - h).
\end{aligned}
\tag{60}
$$

An example of such a pair $(\hat{I}_{(1)}, \hat{I}_{(0,1)}) = (\hat{w}_h, \hat{z}_h)$ can be easily obtained as follows

$$
\begin{bmatrix} \hat{w}_h \\ \hat{z}_h \end{bmatrix} = \begin{bmatrix} \sqrt{h} & 0 \\ \frac{h\sqrt{h}}{2} & \frac{h\sqrt{h}}{2\sqrt{3}} \end{bmatrix} \begin{bmatrix} u_1 \\ u_2 \end{bmatrix}, \qquad u_1, u_2 \overset{\text{i.i.d.}}{\sim} \mathcal{N}(0, 1).
$$

Indeed, for such a pair we have

$$
\mathbb{E}\left[\begin{bmatrix} \hat{w}_h \\ \hat{z}_h \end{bmatrix} \begin{bmatrix} \hat{w}_h & \hat{z}_h \end{bmatrix}\right] = \begin{bmatrix} h & \frac{h^2}{2} \\ \frac{h^2}{2} & \frac{h^3}{3} \end{bmatrix}.
$$

In light of the above expression, the truncated Taylor expansions we refer in the main part of the paper consists on the consideration of only the coefficients in $L_t^k$. The only noise noise contribution we will consider corresponds to $g(s)\hat{I}_{(1)}$.

# F   Experiment Details

We evaluate the Fréchet inception distance (FID) after generating 50K samples with each solver, and compare with the statistics of real-data. In our experiments we make use of the code from [16] for continuously trained models as well as their reference FID stats[2] and that of [22] for discretely trained models.

All the experiments of SEEDS for continuous-time models are parameterized within the EDM framework with the discretization of type EDM, linear schedule, and scaling none as described on [16] in noise prediction mode unless explicitly stated. We use the SEEDS-3 method that has 3 NFEs per step and fixed step-size and report FID scores at NFEs divisible by 3.

We leverage the explicit Langevin-like "churn" trick using in [16] to add or remove noise in the sampling phase. Specifically, [16] uses 4 hyper-parameters $S_{\mathrm{churn}}, S_{\mathrm{min}}, S_{\mathrm{max}}$ and $S_{\mathrm{noise}}$ in which $S_{\mathrm{churn}}$ controls the overall amount of stochasticity added before giving the input to the SEEDS-3 method when the noise level (or time step in EDM configuration) $t_i$ is contained in the noise interval $[S_{\mathrm{min}}, S_{\mathrm{max}}]$. It means that the EDM proposed sampler is stochastic under some conditions of those hyper-parameters and deterministic otherwise, while our method is completely stochastic. In our experiments, we set $S_{\mathrm{churn}} = 0$ except for ImageNet-64 EDM optimized model. We noticed that using the additional stochasticity indeed helps to improve the image quality as in Fig. 1 (c). Moreover, setting $S_{\mathrm{noise}}$ slightly above 1 might correct the errors in earlier steps more effectively as indicated in [16].

## F.1   Pre-trained model specifications

For producing the CIFAR-10 time-continuous results in Table 1, we use the VP DDPM++ continuous architecture. These models are publicly available in conditional[3] and unconditional[4] versions and were directly derived from [34] under the Apache 2.0 license. On the unconditional mode (Figure 1 (a-b)), we first generate the FID curves of 3 types of DPM-Solver (with orders 1, 2 and 3) using the updates from their official implementation[5] in noise prediction mode. Taking profit of the tuning advancements proposed by [16], we used a linear noise schedule with $\beta_{\mathrm{d}} = 19.1$ and $\beta_{\mathrm{min}} = 0.1$ that slightly differs from the original parameters from [34] but were proven beneficial. We set the end time of sampling $\varepsilon$ to 1e-4 as recommended by [22, Appendix D.2]. The values of all benchmark models for Figure 1 (b) were taken directly from tables provided by [22].

In our FFHQ-64 experiments, we employ the unconditional VP pretrained[6] model provided by [16].

For the CelebA-64 experiments, we use the pre-trained VP unconditional model whose checkpoint[7] is provided by [33]. We use the Type-1 discretization proposed in [22] to ensure compatibility of our method with the prescribed trained steps of such model.

For ImageNet-64, we both use the baseline and the optimized pre-trained models given in [16]. We note that the baseline is trained on the iDDPM class of model [26], which actually uses different preconditioning and thus the change of variables compared to the optimized model. The Figure 1 (c) was obtained using the EDM-preconditioned checkpoint[8]. The added noise settings of SEEDS-3 solver were not subject to a grid-search optimization procedure. The chosen hyper-parameters were $S_{\mathrm{churn}} = 11$, $S_{\mathrm{noise}} = 1.003$, $S_{\mathrm{min}} = 0.05$, and $S_{\mathrm{max}} = 15$ but we are confident that this configuration can be optimized to further improve our results.

---

[2]https://nvlabs-fi-cdn.nvidia.com/edm/fid-refs/

[3]https://nvlabs-fi-cdn.nvidia.com/edm/pretrained/baseline/baseline-cifar10-32x32-cond-vp.pkl

[4]https://nvlabs-fi-cdn.nvidia.com/edm/pretrained/baseline/baseline-cifar10-32x32-uncond-vp.pkl

[5]https://github.com/LuChengTHU/dpm-solver

[6]https://nvlabs-fi-cdn.nvidia.com/edm/pretrained/baseline/baseline-ffhq-64x64-uncond-vp.pkl

[7]https://drive.google.com/file/d/1R_H-fJYXSH79wfSKs9D-fuKQVan5L-GR/view?usp=sharing

[8]https://nvlabs-fi-cdn.nvidia.com/edm/pretrained/edm-imagenet-64x64-cond-adm.pkl

## F.2 Noise vs. Data Prediction approaches

In Appendix B of [23], the authors compare DPM-Solver2 and DPM-Solver++(2S), which amounts on comparing in our framework the difference between the obtained exponential integrators for the PFO in the noise and data prediction regimes to detect exactly a coefficient on the non-linear term that is absent in the noise prediction regime. The term they find corresponds exactly to the difference between applying the variation of constants formula before (instead of after) replacing the score function with the desired neural network. In Tab. 3 we report both data and noise prediction SEEDS-3. At low NFEs the DP approach gives better results but stabilizes in high NFEs at a FID score that is worse than the one the NP approach reaches.

Table 3: Comparison between noise prediction $F_{\theta,t}$ and data prediction $D_{\theta,t}$ modes of SEEDS-3 on CIFAR-10 (VP uncond. cont.).

| METHOD \ NFE | 9 | 30 | 60 | 90 | 150 | 165 | 180 |
|---|---|---|---|---|---|---|---|
| SEEDS-3 DATA PREDICTION | 60.75 | 22.42 | 12.47 | 2.95 | 2.51 | 2.54 | 2.55 |
| SEEDS-3 NOISE PREDICTION | 471.29 | 288.20 | 33.92 | 3.76 | 2.40 | **2.39** | 2.47 |

## F.3 Low vs. High stage Solvers

Similar to DPM-Solver [22], the FID scores in Tab. 4 and 5 show that at low NFEs, higher stage methods performs more poorly while at higher NFEs, DPM-Solver-3 and SEEDS-3 are better than their counterparts with 1 and 2 stages.

Table 4: FID comparison between SEEDS (Ours) and DPM-Solver for low NFEs on CIFAR-10 VP uncond. discrete. We recomputed the DPM-Solver score using the "non-deep" model while [22] reports results for the "deep" architecture. The symbol $\star$ is used when using 1 NFE more and $\dagger$ when using 1 NFE less because the given NFE cannot be divided by 2 or 3. This corresponds to Figure 1 (a).

| METHOD \ NFE | 10 | 12 | 15 | 20 | 30 | 40 | 50 | 100 |
|---|---|---|---|---|---|---|---|---|
| DPM-SOLVER-1 | 22.90 | 17.73 | 13.36 | 9.78 | 6.87 | 5.77 | 5.17 | 4.22 |
| DPM-SOLVER-2 | **12.22** | **6.52** | $\star$**4.55** | – | 3.75 | 3.68 | 3.64 | 3.60 |
| DPM-SOLVER-3 | $\dagger$66.92 | 9.72 | 5.32 | $\star$3.83 | **3.66** | – | $\star$**3.61** | $\dagger$3.58 |
| SEEDS-1 | 303.48 | 239.79 | 279.84 | 192.68 | 84.78 | 45.26 | 28.18 | 8.24 |
| SEEDS-2 | 481.09 | 473.48 | $\star$430.98 | 305.88 | 223.01 | 51.41 | 11.10 | **3.19** |
| SEEDS-3 | $\dagger$483.04 | 482.19 | 479.63 | $\star$462.61 | 280.48 | $\dagger$247.44 | $\star$62.62 | $\dagger$3.53 |

Table 5: FID comparison between SEEDS (Ours) and DPM-Solver for high NFEs on CIFAR-10 VP uncond. discrete. We recomputed the DPM-Solver score using the "non-deep" model while [22] reports results for the "deep" architecture. The symbol $\star$ is used when using 1 NFE more because the given NFE cannot be divided by 2 or 3. This corresponds to Figure 1 (a).

| METHOD \ NFE | 150 | 200 | 300 | 510 |
|---|---|---|---|---|
| DPM-SOLVER-3 | 3.59 | $\star$3.58 | – | 3.58 |
| SEEDS-1 | – | 4.07 | 3.40 | – |
| SEEDS-3 | 3.12 | $\star$3.08 | 3.14 | 3.24 |

Table 6: FID comparison between SEEDS-3 (Ours) and other solvers on CIFAR-10 VP unconditional discrete. The symbol $\star$ is used when using 1 NFE more and $\dagger$ when using 1 NFE less because the given NFE cannot be divided by 2 or 3. This corresponds to Figure 1 (b).

| METHOD \ NFE | 10 | 12 | 15 | 20 | 50 | 200 | 1000 |
|---|---|---|---|---|---|---|---|
| EULER-MARUYAMA | 278.67 | 246.29 | 197.63 | 137.34 | 32.63 | 4.03 | 3.16 |
| ANALYTIC DDPM | 35.03 | 27.69 | 20.82 | 15.35 | 7.34 | 4.11 | 3.84 |
| ANALYTIC DDIM | 14.74 | 11.68 | 9.16 | 7.20 | 4.28 | 3.60 | 3.86 |
| DDIM | 13.58 | 11.02 | 8.92 | 6.94 | 4.73 | 4.07 | 3.95 |
| DPM-SOLVER-3 | $\dagger$**6.92** | **9.72** | **5.32** | $\star$**3.83** | $\star$**3.61** | $\star$3.58 | – |
| SEEDS-3 | $\dagger$483.04 | 482.19 | 479.63 | $\star$462.61 | $\star$62.62 | $\star$**3.08** | – |

## F.4 Deterministic vs. Stochastic Solvers

Deterministic solvers as DPM-Solver [22] are fast and well-adapted to applications in which speed is the most concern. As shown in [22], Table 4 and 5, DPM-Solver converges to a local minimum at early steps and cannot be improved in large NFEs. Moreover, preconditioned deterministic solver in EDM gives optimal quality on unconditional CIFAR-10 [16, Fig. 5 (b)]. However, for more complicated datasets as ImageNet64, the stochasticity indeed helps improve the samples quality [16, Fig. 5 (c)]. We can consider the random noise as a corrector that approaches a better local or even global minimum. At $S_{\text{churn}} = 0$, our SEEDS-3 with stochasticity gives lower FID score than EDM deterministic Heun (see Fig. 1 (c)). SEEDS-3 also reaches the best quality prior to the number steps needed in Euler Maruyama and other solvers as in Table 6. Table 7 completes the analysis in Table 1 for CIFAR-10 showing FID scores for varying NFEs and multiple versions of the EDM solver, depending on the chosen optimization hyper-parameters. Tables 8 to 10 complete the findings in Table 1 for the remaining used pretrained models. In particular, we report the exact FID values of SEEDS in the low NFE regime.

Table 7: FID comparison of different solvers on CIFAR-10-uncond-vp-continuous. The symbol $\star$ is used when using 1 NFE more and $\dagger$ when using 1 NFE less because the given NFE cannot be divided by 2 or 3. This corresponds to Figure 1 (b). All values for EDM are retrieved from the Latex source file SdePlotNfe.tex in the arXiv version of [16].

| METHOD \ NFE | 30 | 48 | 63 | 126 | 165 | 180 | 511 |
|---|---|---|---|---|---|---|---|
| GOTTAGOFAST | - | 82.42 | - | - | 2.75 | 2.44 | - |
| EDM ($S_{\text{CHURN}} = 0$) | $\star$3.10 | $\dagger$2.99 | 2.94 | $\star$2.98 | - | - | 2.93 |
| EDM ($S_{\text{TMIN,TMAX}} + S_{\text{NOISE}} = 1$) | $\star$3.44 | $\dagger$2.89 | 2.77 | 2.72 | - | - | 2.69 |
| EDM ($S_{\text{NOISE}} = 1$) | $\star$3.99 | $\dagger$3.13 | 2.90 | 2.60 | - | - | 2.55 |
| EDM ($S_{\text{TMIN,TMAX}} = [0, \infty]$) | $\star$3.43 | $\dagger$2.87 | 2.71 | 2.52 | - | - | 2.54 |
| EDM (OPTIMIZED) | $\star$3.77 | $\dagger$3.08 | 2.84 | 2.47 | - | - | **2.27** |
| DPM-SOLVER (NON-DEEP) | 2.95 | 2.90 | 2.89 | - | - | - | - |
| SEEDS ($S_{\text{CHURN}} = 0$) | $\star$288.20 | $\dagger$90.25 | 33.91 | 2.45 | **2.39** | 2.47 | - |

## F.5 Run-time Comparison

As a sanity check, we provide a run-time comparison table for our experiments on different datasets (see Tab. 11). One can see that the run-time is linear with respect to the NFE also for SEEDS, as the main advantage of the SETD method is to analytically compute the stochastic components in our solver, making their computation cost negligible. Some mild improvements about repetitive terms allowed our implementation of SEEDS to be slightly more effective than EDM and even DPM-Solver at same NFE.

Table 8: FID comparison of different solvers on CelebA 64x64 discrete. The symbol $\star$ is used when using 1 NFE more and $\dagger$ when using 1 NFE less because the given NFE cannot be divided by 2 or 3.

| METHOD \ NFE | 9 | 12 | 15 | 21 | 51 | 60 | 90 | 102 | 200 |
|---|---|---|---|---|---|---|---|---|---|
| E-M | $\star$310.22 | 227.16 | 207.97 | $\dagger$120.44 | $\dagger$29.25 | - | - | - | 3.90 |
| AN.-DDPM | $\star$28.99 | 25.27 | 21.80 | $\dagger$18.14 | $\dagger$11.23 | - | - | - | 6.51 |
| AN.-DDIM | $\star$15.62 | 13.90 | 12.29 | $\dagger$10.45 | $\dagger$6.13 | - | - | - | 3.46 |
| DDIM | $\star$10.85 | 9.99 | 7.78 | $\dagger$6.64 | $\dagger$5.23 | - | - | - | 4.78 |
| DPM-SOLVER | 6.92 | 4.20 | 3.05 | 2.82 | 2.72 | - | - | - | - |
| SEEDS | 460.87 | 374.48 | 301.66 | 261.87 | 3.84 | 6.58 | **1.88** | 1.97 | - |

Table 9: SEEDS-3 on CIFAR-10-cond-vp-continuous (using the baseline model in [16]).

| METHOD \ NFE | 15 | 21 | 30 | 60 | 90 | 120 | 129 | 150 | 165 | 180 |
|---|---|---|---|---|---|---|---|---|---|---|
| SEEDS | 239.2 | 167.5 | 131.1 | 25.06 | 3.19 | 2.17 | **2.08** | 2.15 | 2.16 | 2.19 |

## F.6 Hardware configuration

During the experiments, we used three Linux-based servers with 60GB memory each and 4 GPUs NVIDIA V100 32GB, 4 GPUs NVIDIA V100 16GB, and 2 GPUs NVIDIA V100 32GB, respectively. Table 12 shows the detail of the configuration utilized for each experiment. We noted that when using the 4 GPUs configuration, the FID results were slightly lower (around 2%), even after using a stacked fixed random seed. We run the experiments multiple times and reported the minimum FID value each time.

## F.7 Licences

Pre-trained models:

- CIFAR-10 models by [34]: Apache V2.0 license
- FFHQ-64 model by [16]: Creative Commons Attribution-NonCommercial-ShareAlike 4.0 International License.
- CelebA-64 model by [33]: Apache V2.0 license
- ImageNet-64 model by [16]: Apache V2.0 license
- Inception-v3 model by [35]: Apache V2.0 license

## F.8 Supporting samples

In this subsection, we report the image grids supporting our claims in section 5.

Table 10: Tailored SEEDS-3 for the EDM-preconditioned pretrained model [16] on ImageNet 64x64.

| Method \ NFE | 12 | 15 | 21 | 51 | 102 | 201 | 270 |
|---|---|---|---|---|---|---|---|
| SEEDS | 209.12 | 197.79 | 153.72 | 63.75 | 16.35 | 1.56 | **1.38** |

Table 11: Run-time comparison (second/batch + std) on a single NVIDIA V100 of EDM, DPM-Solver and SEEDS. Discretely-trained models are run with the implementation based on the code from [22]. Continuously-trained models are run using the code from [16].

| Method \ NFE | 9 | 21 | 51 | 90 |
|---|---|---|---|---|
| **CIFAR-10 32×32 continuous (batch size = 128)** | | | | |
| EDM | 2.096(±0.003) | 4.891(±0.004) | 11.592(±0.019) | 21.213(±0.009) |
| DPM-Solver | 2.099(±0.002) | 4.871(±0.004) | 11.841(±0.014) | 20.888(±0.020) |
| SEEDS | 2.086(±0.002) | 4.867(±0.003) | 11.817(±0.006) | 20.896(±0.028) |
| **FFHQ 64×64 continuous (batch size = 128)** | | | | |
| EDM | 4.361(±0.005) | 10.179(±0.005) | 24.738(±0.018) | 44.145(±0.016) |
| DPM-Solver | 4.344(±0.005) | 10.148(±0.009) | 24.637(±0.007) | 43.526(±0.029) |
| SEEDS | 4.353(±0.003) | 10.157(±0.004) | 24.661(±0.011) | 43.537(±0.013) |
| **ImageNet 64×64 continuous (batch size = 128)** | | | | |
| EDM | 7.525(±0.007) | 17.579(±0.004) | 42.696(±0.010) | 76.175(±0.024) |
| DPM-Solver | 7.535(±0.048) | 17.556(±0.009) | 42.629(±0.018) | 75.222(±0.026) |
| SEEDS | 7.429(±0.006) | 17.572(±0.013) | 42.654(±0.014) | 75.245(±0.034) |
| **CIFAR-10 32×32 discrete (batch size = 128)** | | | | |
| DPM-Solver | 0.272(±0.004) | 0.529(±0.007) | 1.324(±0.007) | 2.632(±0.004) |
| SEEDS | 0.261(±0.002) | 0.523(±0.002) | 1.299(±0.003) | 2.598(±0.003) |
| **CelebA 64×64 discrete (batch size = 128)** | | | | |
| DPM-Solver | 0.936(±0.004) | 1.812(±0.003) | 4.558(±0.008) | 9.108(±0.008) |
| SEEDS | 0.912(±0.002) | 1.808(±0.004) | 4.526(±0.002) | 9.033(±0.004) |
| **LSUN-Bedroom 256×256 discrete (batch size = 32)** | | | | |
| DPM-Solver | 5.566(±0.019) | 11.124(±0.018) | 27.815(±0.031) | 55.648(±0.021) |
| SEEDS | 5.498(±0.008) | 11.001(±0.022) | 27.503(±0.029) | 54.842(±0.020) |
| **LDM-CelebAHQ 256×256 (batch size = 64)** | | | | |
| DPM-Solver | 8.648(±0.013) | 17.492(±0.013) | 39.569(±0.015) | 68.205(±0.017) |
| SEEDS | 8.652(±0.005) | 17.469(±0.010) | 39.524(±0.010) | 68.240(±0.026) |
| **Stable Diffusion 512×512 (batch size = 16)** | | | | |
| DPM-Solver | 19.598(±0.070) | 40.679(±0.107) | 92.914(±0.079) | 163.902(±0.116) |
| SEEDS | 19.656(±0.106) | 40.781(±0.136) | 93.409(±0.123) | 161.451(±0.518) |

Table 12: Details of GPUs utilized during the experiments.

| Experiment | Model | Number | GPU Size |
|---|---|---|---|
| CIFAR-10 continuous | Nvidia V100 | 4 | 16 GB |
| CIFAR-10 discrete | Nvidia V100 | 2 | 32 GB |
| FFHQ64 | Nvidia V100 | 4 | 16 GB |
| CelebA64 | Nvidia A100 | 2 | 16 GB |
| ImageNet64 | Nvidia V100 | 4 | 32 GB |

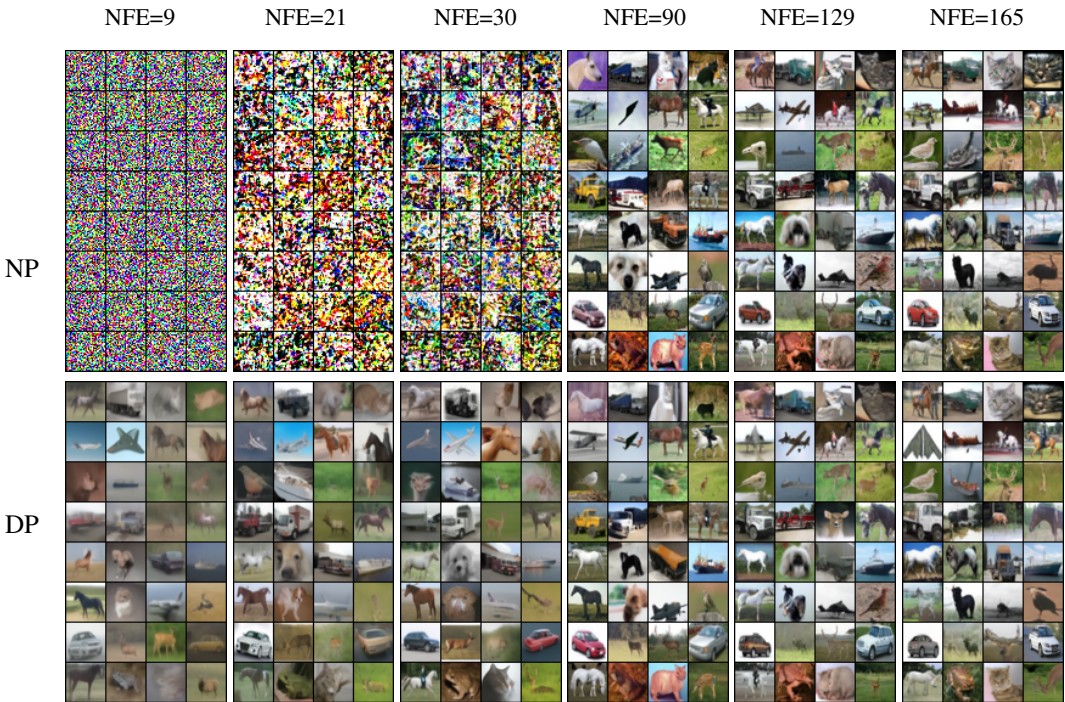

Figure 4: Samples on CIFAR-10 from low to high NFEs by SEEDS-3 in Noise Prediction (NP) and Data Prediction (DP) modes, using conditional VP continuous baseline model [16].

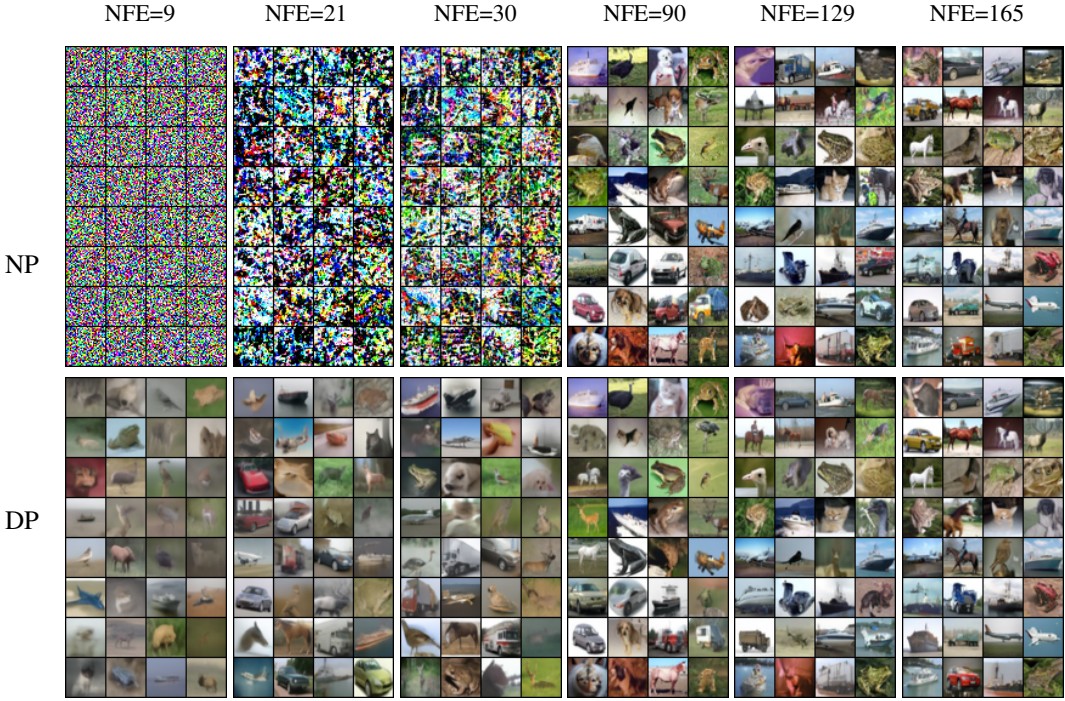

Figure 5: Samples on CIFAR-10 from low to high NFEs by SEEDS-3 in Noise Prediction (NP) and Data Prediction (DP) modes, using unconditional VP continuous baseline model [16].

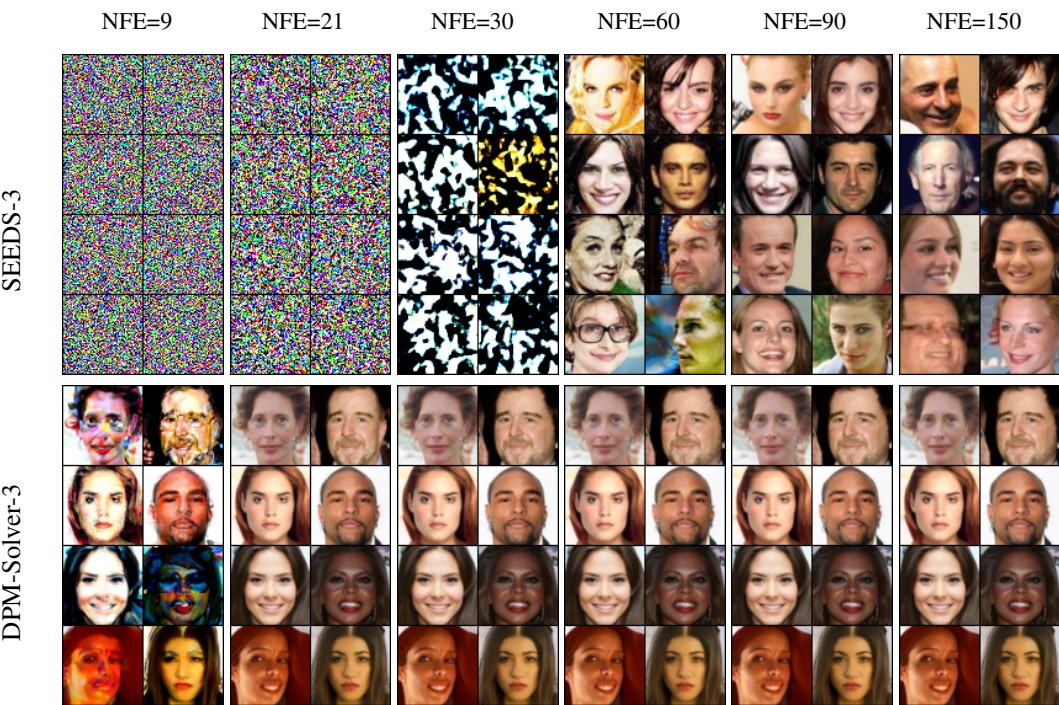

Figure 6: Samples on CelebA-64 from low to high NFEs by SEEDS-3 and DPM-Solver-3, using pre-trained model from [33].

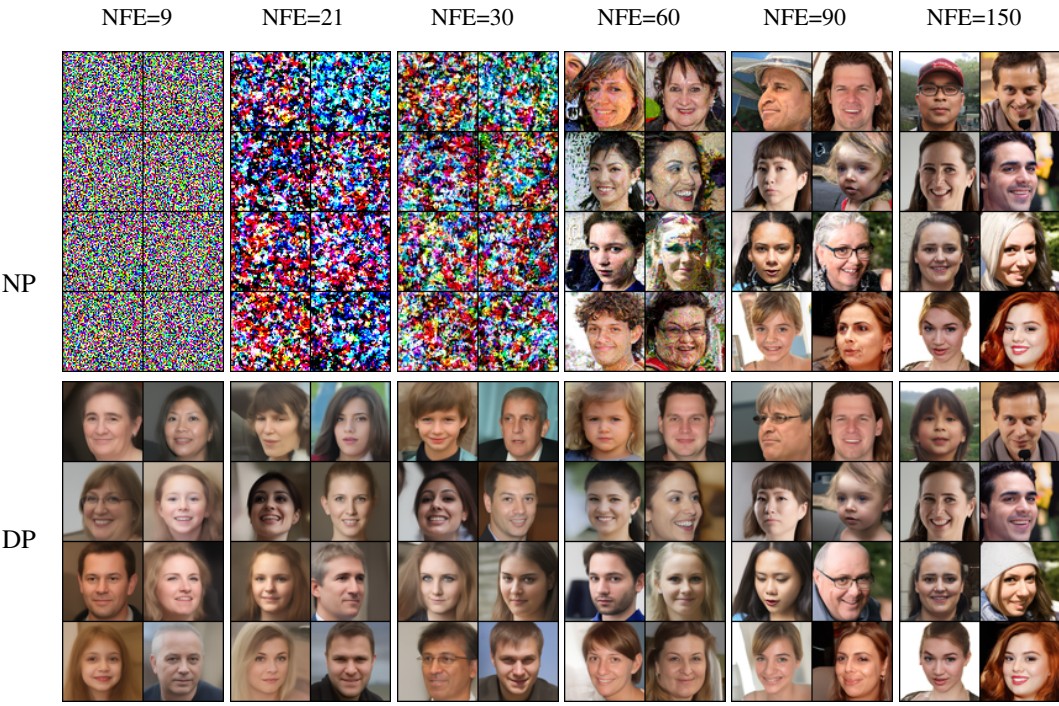

Figure 7: Samples on FFHQ-64 from low to high NFEs by SEEDS-3 in Noise Prediction (NP) and Data Prediction (DP) modes, using unconditional VP continuous baseline model [16].

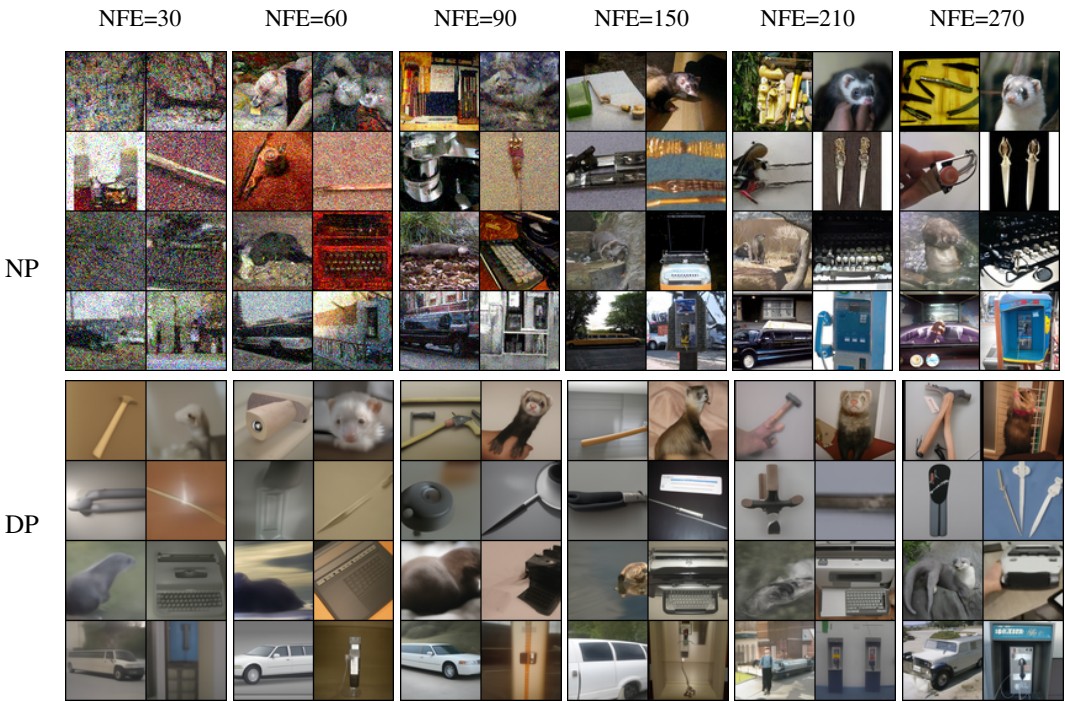

Figure 8: Samples on ImageNet-64 from low to high NFEs in Noise Prediction (NP) and Data Prediction (DP) modes, using conditional EDM optimized model [16].

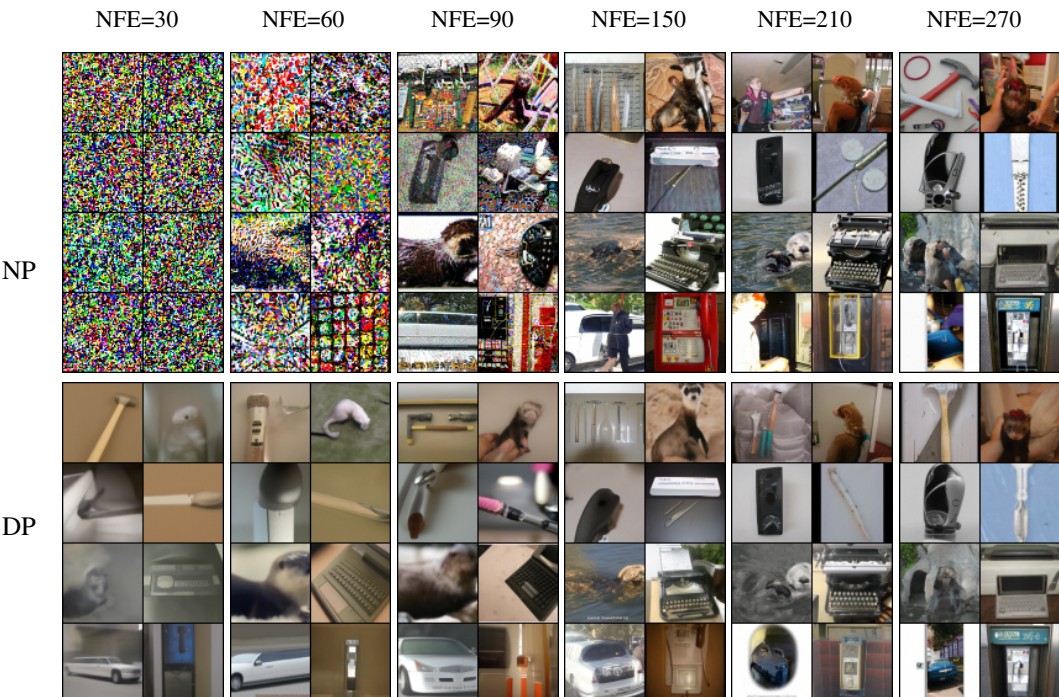

Figure 9: Samples on ImageNet-64 from low to high NFEs in Noise Prediction (NP) and Data Prediction (DP) modes, using conditional EDM baseline model [16].

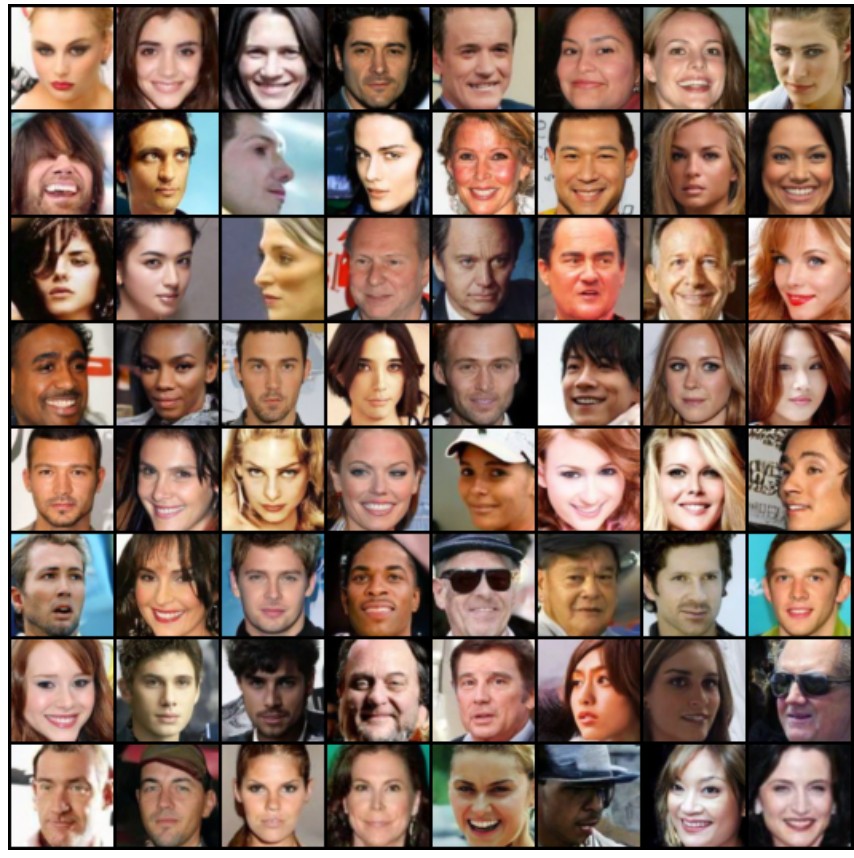

Figure 10: Example of samples on CelebA-64 generated by SEEDS-3 in 90 NFEs, using pre-trained model from [33].

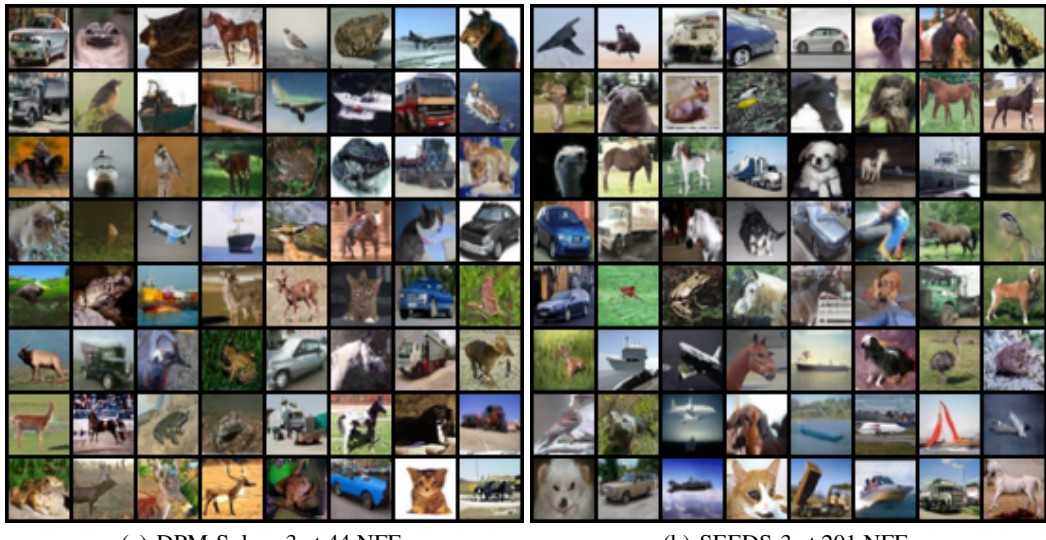

(a) DPM-Solver-3 at 44 NFE.
(b) SEEDS-3 at 201 NFE.

Figure 11: Visual sample quality comparison between DPM-Solver-3 and SEEDS-3 using their optimal settings and unconditional CIFAR-10 discrete model [34].

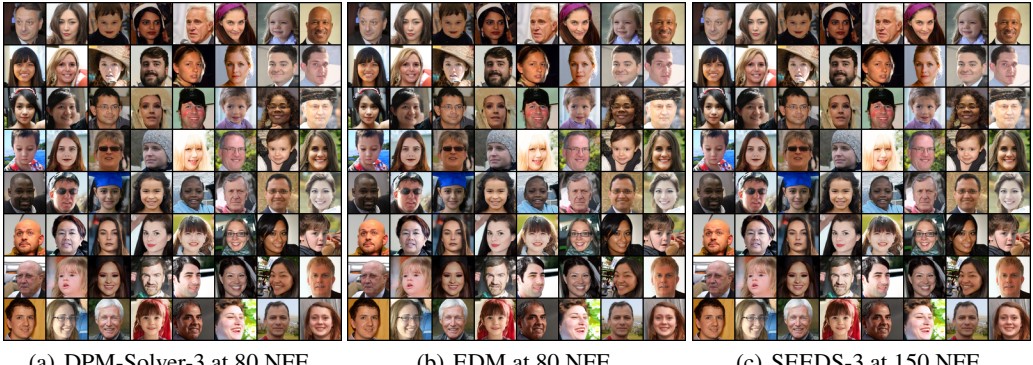

| (a) DPM-Solver-3 at 80 NFE. | (b) EDM at 80 NFE. | (c) SEEDS-3 at 150 NFE. |

Figure 12: Visual sample quality comparison between DPM-Solver-3, EDM and SEEDS-3 using their optimal settings and unconditional VP FFHQ-64 continuous model [16].

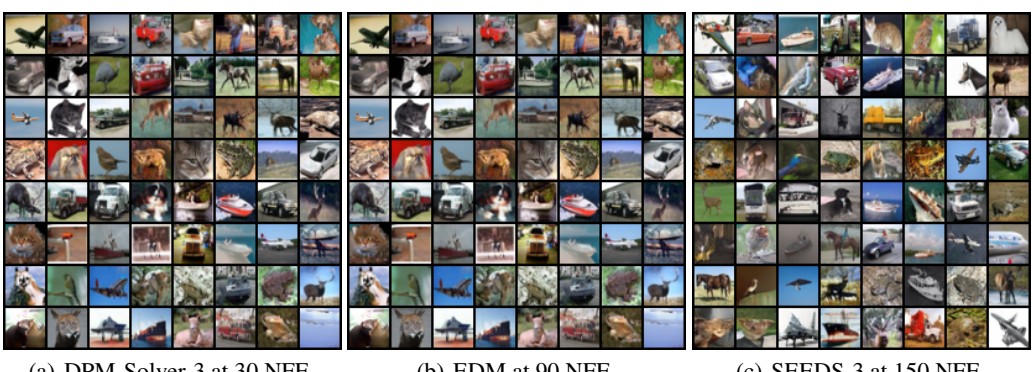

| (a) DPM-Solver-3 at 30 NFE. | (b) EDM at 90 NFE. | (c) SEEDS-3 at 150 NFE. |

Figure 13: Visual sample quality comparison between DPM-Solver-3, EDM and SEEDS-3 using their optimal settings and conditional VP CIFAR-10 baseline continuous model [16].

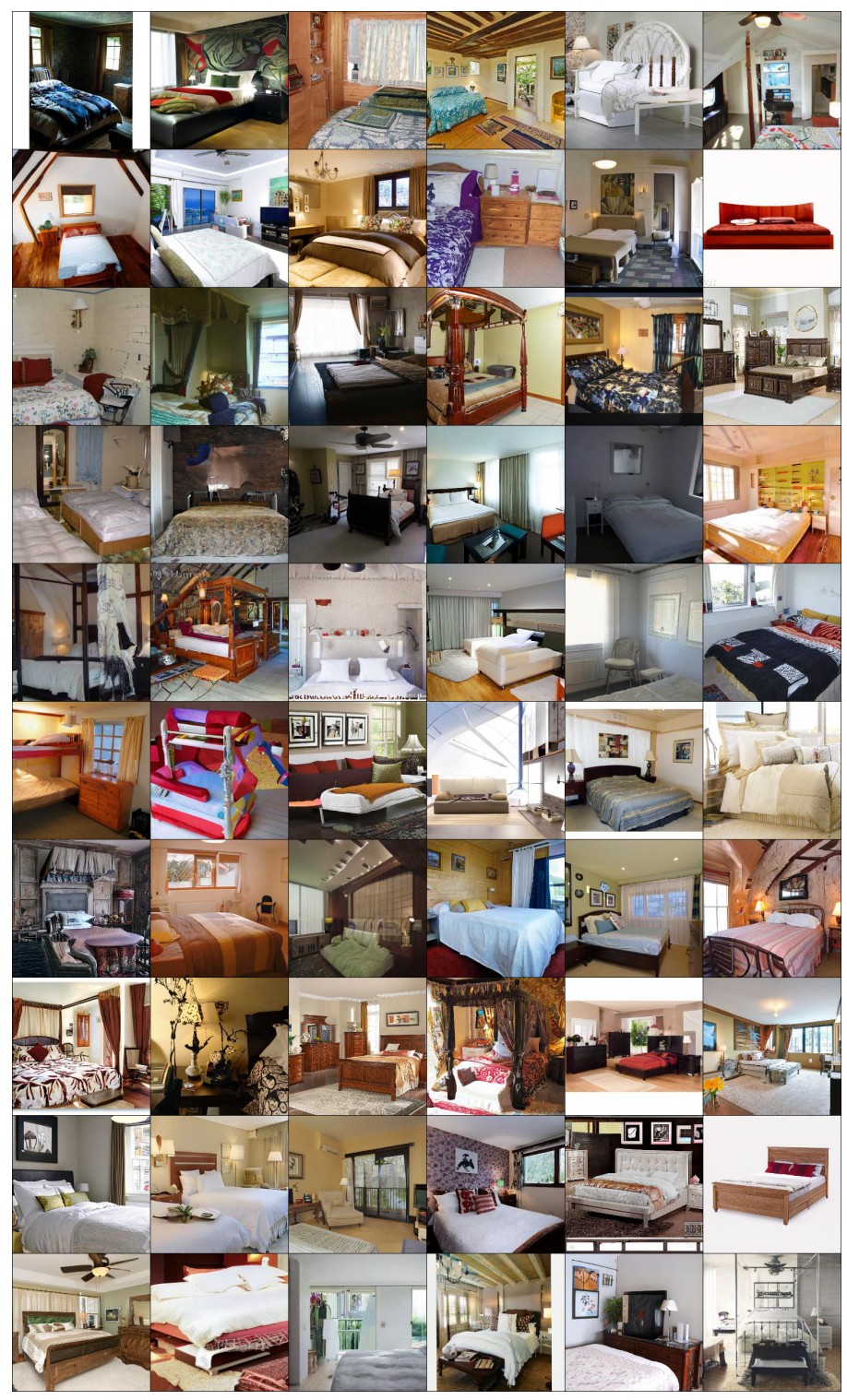

Figure 14: Example of samples on LSUN-Bedroom-256 by SEEDS-3 in 201 NFEs, using pre-trained model from [8].

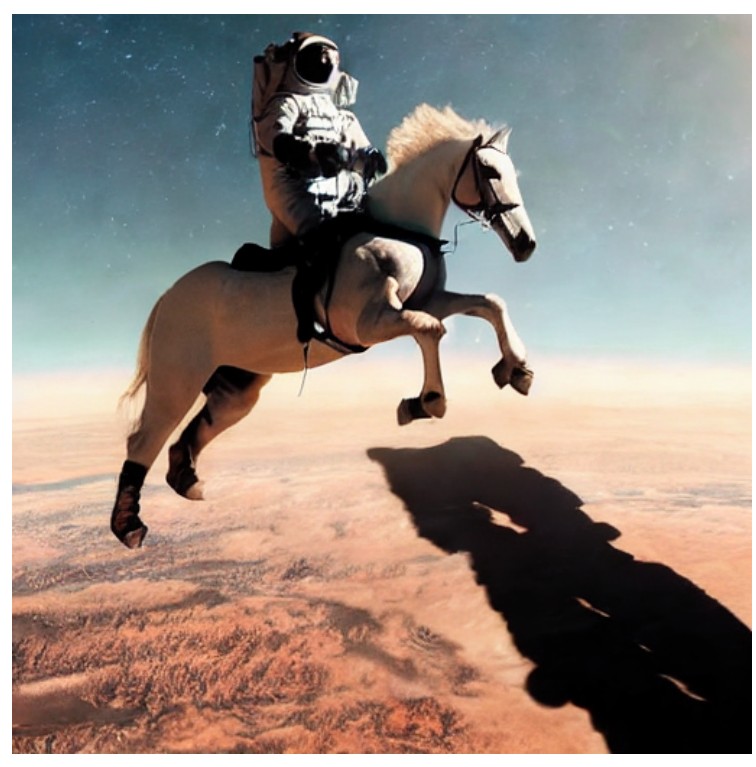

Figure 15: StableDiffusion-$512 \times 512$ by SEEDS-1 with prompt "*High quality photo of an astronaut riding a horse in space*" and NFE $= 90$.

