# OpenReview forum: "SEEDS: Exponential SDE Solvers for Fast High-Quality Sampling from Diffusion Models"
_NeurIPS.cc/2023/Conference — NeurIPS 2023 poster_

### Official Review · Reviewer_YvZd · 2023-07-07

**Soundness:** 3 good
**Presentation:** 3 good
**Contribution:** 3 good
**Rating:** 6
**Confidence:** 4

**Summary:**

There are powerful ODE solvers to speed up the sampling process of diffusion models. Despite being quick, ODE solvers do not usually reach the optimal quality achieved by SDE solvers which are however slow. To tackle this problem, the paper proposes SEEDS, which is based on Exponential Integrator in the stochastic case. SEEDS are derivative and training-free solvers with proven convergence order.

**Strengths:**

1. The authors provide exact solutions for diffusion SDEs and analytical computation of their variance.

2. The paper contains rigorous theoretical derivation and convergence order guarantee for the proposed methods. It also points out its connection with gDDIM.

3. The paper is overall well-written and self-contained.

4. The method achieves outstanding sampling performance.

**Weaknesses:**

The proposed SEEDS method still needs around a hundred NFEs to achieve good sampling qualities, the quality will degrade fast when NFEs decrease, which limits its applications.


**Questions:**

Would you please include the connection between the proposed method SEEDS-1/2/3 and the stochastic Runge-Kutta method properly combined with Exponential Integrator?

Also, the convergence order included is order 1 for SEEDS-1 in the mean-square sense and order 1 for SEEDS-2/3 in the weak sense, since the stochastic Runge-Kutta method can achieve higher convergence order even in the strong sense, the reviewer is concerned the convergence order provided is not optimal.

**Limitations:**

The author addressed the limitations of their work.

---

> ### Author Rebuttal · Authors · 2023-08-09
>
> We thank the reviewer for their careful lecture of our work. Below a discussion to the raised questions, which will be included in the paper.
>
> > Would you please include the connection between the proposed method SEEDS-1/2/3 and the stochastic Runge-Kutta method properly combined with Exponential Integrator?
>
> Contrary to the ODE case, there are many stochastic Runge-Kutta approaches, usually tailored for SDEs of a specific form. Nonetheless, a common way of distinguishing solvers with same strong order is to assign them couples $(p_d,p_s)$, where $p_s$ is the stochastic order and $p_d$ is the resulting order determined if setting $g=0$ in the considered SDE i.e. when they are deterministic. Example: [R, Tab. 6.2] determines solvers with orders (1,1.0) and (2,1.0) respectively and [R, Tab. 6.3] determines solvers SRA1 and SRA3 with orders (2,1.5) and (3,1.5) respectively. Many of these solvers' speed was already tested in [GGF, Table 3] on CIFAR-10 (VP) which we reproduce below with our SEEDS method added to it.
>
> | Method      | Strong-Order | Speed     |
> | :---        |    :----:   |          :---: |
> | Euler-Maruyama (EM)      | 0.5       | Baseline speed  |
> |SEEDS | 1 | 6.83x faster |
> |Lamba EM (atol=1e-3)  |0.5 | 2x faster |
> |Lamba Euler-Heun  |0.5 | 1.75x faster |
> |Lamba EM (atol=1e-3, rtol=1e-3)  |0.5 | 1.27x faster |
> |Euler-Heun |0.5  | 1.86x slower |
> |SOSRA |1.5 | 5.92x slower |
> |SRA3  |1.5 | 6.93x slower |
> |SOSRI  |1.5 | 8.57x slower |
> |Lamba EM (default)  |0.5 | Diverged |
> |RKMil |1.0 | Diverged |
> |ImplicitRKMil  |1.0 | Diverged |
> |ISSEM | 0.5 | Diverged |
>
> To our knowledge, the only available strong order SERKs method for SDEs with inhomogeneous diffusion coefficients are the exponential Euler-Maruyama (EEM) method [K] and the stochastic RK Lawson (SRKL) schemes [D]. In short, the SRKL schemes only compute analytically the linear coefficient and use the Integrating Factor (IF) method to approximate the integrals in the representation of the exact solution given by the variation-of-parameters formula. This way, by a special change of variables (see [D, Alg. 1]), one can create exponential integrator versions of many SDE methods. We implemented our own version of the SRKL schemes to take into account that $\sigma,\alpha$ are not constant and used the Integrating Factor method to approximate the integrals in the representation of the exact solution given by the variation-of-parameters formula (after properly changing variables). Interestingly, the SRKL schemes seem to stabilize at increasing NFEs but at much higher FID values than their SETD counterparts.
>
> **Comparison of SEEDS with current Stochastic RKL methods on CIFAR-10-vp-uncond (discrete)**
>
> | Method\NFEs      | 10 | 20 | 50 | 100 | Best known
> | :---        |    :----:   |          :---: |   :---: |   :---: |   :---: |
> | SRKL1($\lambda$)      |    332.52   |   282.96 |  33.42   |   8.62 | \ |
> | SEEDS-1   |     303.48    |   153.21    |  22,70   |   7.97 | 3.13(500NFE) |
> | SRKL2($\lambda$)      |    475.20   |   469.64 |  134.82   |   7.74 | \ |
> | SEEDS-2   |     476.9    |   226.7    |  7.17   |   3.23  | 3.21(90NFE) |
> | SRKL3($\lambda$)       |    462.24   |   376.15 |  8.36   |   7.46 | \ |
> | SEEDS-3   |     483.0    |   428.6    |  43.3   |   3.41 | 3.08(201NFE) |
>
> > Also, the convergence order included is order 1 for SEEDS-1 in the mean-square sense and order 1 for SEEDS-2/3 in the weak sense, since the stochastic Runge-Kutta method can achieve higher convergence order even in the strong sense, the reviewer is concerned the convergence order provided is not optimal.
>
> The proposed convergence order for each SEEDS-1/2/3 is optimal: this is a consequence of the general result from [CC] about maximum convergence rates for SDE schemes with uncorrelated Gaussian increments. The underlying idea is that any solver with strong order $\geq 1.5$ has to account for double stochastic integrals in the non-truncated Itô-Taylor expansion, ultimately forcing any SRK-like solver to use correlated random variables (see [R, Tab. 6.3] and [KP] more generally). SEEDS avoids this additional complexity but an interesting future avenue would be to extend SEEDS to the higher strong order case (and not in the IF approach but the SETD approach). Another interesting path would be to craft weak second order SERK methods for DPMs (the work [KCB] addressed this only for homogeneous semi-linear SDEs).
>
> We hope to have addressed all necessary concerns from the reviewer to promote the acceptance of our paper.
>
> References:
>
> [R] Rossler. Runge-Kutta Methods for the Strong Approximation of Solutions of Stochastic Differential Equations
>
> [K] Komori. Exponential Runge-Kutta Methods for Stiff Stochastic Differential Equations
>
> [D] Debrabant et al. Runge-Kutta Lawson Schemes for Stochastic Differential Equations
>
> [GGF] Jolicoeur-Martineau et al. Gotta Go Fast when Generating Data with Score-based Models
>
> [CC] Clark & Cameron. The Maximum Rate of Convergence of Discrete Approximations for Stochastic Differential Equations
>
> [KP] Kloeden & Platen. Numerical Solution of Stochastic Differential Equations
>
> [KCB] Komori et al. Weak Second Order Explicit Exponential Runge-Kutta methods for Stochastic Differential Equations

---

### Official Review · Reviewer_daVP · 2023-07-09

**Soundness:** 3 good
**Presentation:** 3 good
**Contribution:** 2 fair
**Rating:** 6
**Confidence:** 4

**Summary:**

This paper proposes an off-the-shelf (i.e., no further training) few-step sampler for diffusion probabilistic models (DPMs). By isolating linear terms in the exact solution of diffusion SDEs and using a change-of-variable method, the proposed sampler, SEEDS, simplifies the integrals and can approximate the solution of stochastic equations. SEEDS can accelerate the sampling of DPMs without compromising the quality of the samples.

**Strengths:**

- (Quality & Clarity) The authors have successfully provided a theoretical derivation and error analysis of their proposed method, as well as details of the experiments. The paper is well-written and the reviewer had no difficulty comprehending its contents.

- (Significance) Numerical experiments show that SEEDS produces competitive results compared to previous sampling methods in terms of quality, with fewer function evaluations, in several benchmarks, namely CIFAR-10, CelebA-64, and ImageNet-64.

**Weaknesses:**

- (Originality) The proposed method has shown promising performance. However, its theoretical background is based on a classical idea of the *variation-of-parameters* formula in the literature of differential equations. Although the authors claim that equation (5) is a novel representation of diffusion SDEs, this kind of solution form is very well-known in the SDE community even for general Hilbert spaces (e.g. see Curtain & Falb and Da Prato & Zabczyk). Therefore, it seems that justification is needed to claim that the contribution of this paper regarding the solution representation suggested by the author is novel. Furthermore, the derivation of exponentially weighted integral is quite similar to the DPM-solver (Lu et al.), except for utilizing Itō-Taylor expansion due to the diffusion part in the solution representation. The convergence result of the proposed algorithm may not be an incremental result. However, this paper's argument on the contribution has an excessive aspect, and it is quite difficult for reviewers to agree with the author's claim.

[Curtain & Falb] Stochastic Differential Equations in Hilbert Space, *Journal of Differential Equations*, (1971).

[Da Prato & Zabczyk] Stochastic Equations in Infinite Dimensions, (2014).

[Lu et al.] DPM-Solver: A Fast ODE Solver for Diffusion Probabilistic Model Sampling in Around 10 Steps, *NeurIPS*, (2022).

**Questions:**

To discuss the generation speed of the SEEDS-3 algorithm in Table 1, it is necessary to compare it quantitatively with the actual runtime, as well as with NFE. For example, in the case of the DPM-Solver, a comparison of DDIM with respect to runtime was made in Table 7 of Appendix E.7.

**Limitations:**

The authors mentioned the limations of their work.

---

> ### Author Rebuttal · Authors · 2023-08-09
>
> We thank and agree with reviewer daVP on the poor choice of words for presenting the well-known the variation-of-parameters formula and we remove all mentions of novelty in this point giving the impression of overselling our work. Instead, we highlighted in the general rebuttal response what we find to be novel in our work.
>
> As requested, we report below the average runtime of a batch on a single NVIDIA V100 of EDM, DPM-Solver and SEEDS for sampling (at increasing NFEs) by all diffusion models mentioned in Table 1, and additional higher dimension models: Latent Diffusion Model (LDM) and Stable Diffusion (SD). For reference, we added in Fig.2 of the appended PDF file a 512x512 image generated by Stable Diffusion using SEEDS-1 at only 90 NFEs to ensure that the runtime in this case follows an improvement in generation quality. We set batch sizes of 16 for SD, 32 for LSUN-Bedroom, 64 for CelebaHQ256 and 128 for all other datasets. One can see that the runtime is linear with respect to the NFE also for SEEDS, as the main advantage of the SETD method is to analytically compute the stochastic components in our solver, making their computation cost negligible. Some mild improvements about repetitive terms allowed our implementation of SEEDS to be slightly more effective than EDM and even DPM-Solver at same NFE.
>
> **Runtime comparison (second /batch + std) on a single NVIDIA V100 of EDM, DPM-Solver and SEEDS**
>
> (*Discretely-trained models, implementation based on the [DDIM] code.†computed using the [EDM] unmodified code. ‡our implementation in the [EDM] code.)
> | Sampling method\NFE           | 9 | 21 | 51 | 90 | 99 |
> | :---              |    :---:   |    :---: |  :---: |  :---: |  :---: |
> | **CIFAR-10 32x32** |
> | EDM† | 2.096(±0.003)|4.891(±0.004)|11.952(±0.019)|21.213(±0.009)|23.086(±0.010)|
> | DPM-Solver‡ | 2.099(±0.002) |	4.871(±0.004) |	11.841(±0.014) |	20.888(±0.020) |	23.006(±0.025)|
> | SEEDS‡ | 2.086(±0.002) |	4.867(±0.003) |	11.817(±0.006) |	20.896(±0.028) |	22.957(±0.009)|
> | **FFHQ 64x64** |
> | EDM† | 4.361(±0.005)|	10.179(±0.005)|	24.738(±0.018)|	44.145(±0.016)|	48.007(±0.020)|
> | DPM-Solver‡ | 4.344(±0.005)|	10.148(±0.009)|	24.637(±0.007)|	43.526(±0.029)|	47.857(±0.025)|
> | SEEDS‡ | 4.353(±0.003)|	10.157(±0.004)|	24.661(±0.011)|	43.537(±0.013)|	47.885(±0.018)|
> | **ImageNet 64x64** |
> | EDM† | 7.525(±0.007)|	17.579(±0.004)|	42.696(±0.010)|	76.175(±0.024)|	82.870(±0.032)|
> | DPM-Solver‡ | 7.535(±0.048)|	17.556(±0.009)|	42.629(±0.018)|	75.222(±0.026)|	82.725(±0.016)|
> | SEEDS‡ | 7.429(±0.006)|	17.572(±0.013)|	42.654(±0.014)|	75.245(±0.034)|	82.793(±0.036)|
> | **CIFAR-10 32x32*** |
> | DPM-Solver | 0.272(±0.004)|	0.529(±0.007)|	1.324(±0.007)|	2.632(±0.004)|	2.895(±0.004)|
> | SEEDS | 0.261(±0.002)|	0.523(±0.002)|	1.299(±0.003)|	2.598(±0.003)|	2.863(±0.008)|
> | **Celeba 64x64***|
> | DPM-Solver | 0.936(±0.004)|	1.821(±0.003)|	4.558(±0.008)|	9.108(±0.008)|	10.030(±0.008)|
> | SEEDS | 0.912(±0.009)|	1.808(±0.004)|	4.526(±0.002)|	9.033(±0.004)|	9.922(±0.012)|
> | **LSUN-Bedroom 256x256*** |
> | DPM-Solver | 5.566(±0.019)|	11.124(±0.018)|	27.815(±0.031)|	55.648(±0.021)|	61.168(±0.026)|
> | SEEDS | 5.498(±0.008)|	11.001(±0.022)|	27.503(±0.029)|	54.842(±0.020)|	60.262(±0.027)|
> | **LDM-CelebAHQ 256x256**|
> | DPM-Solver | 8.648(±0.013)|	17.492(±0.013)|	39.569(±0.015)|	68.205(±0.017)| - |
> | SEEDS | 8.652(±0.005)|	17.469(±0.010)|	39.524(±0.010)|	68.240(±0.026)| - |
> | **Stable Diffusion 512x512** |
> | DPM-Solver | 19.598(±0.070)|40.679(±0.107)|92.914(±0.079)|163.902(±0.116)|-|
> | SEEDS | 19.656(±0.106)|	40.781(±0.136)|	93.409(±0.123)|	161.451(±0.518)| - |
>
> We hope to have addressed all concerns from the reviewer necessary to promote the acceptance of our work.
>
> References:
>
> [DDIM] Song et al. Denoising Diffusion Implicit Models
>
> [EDM] Karras et al. Elucidating the Design Space of Diffusion-based Generative Models
>
> [DPM-Solver] Lu et al. DPM-solver: A Fast ODE Solver for Diffusion Probabilistic Model Sampling in Around 10 steps

---

> > ### Comment · Reviewer_daVP · 2023-08-20
> >
> > Thank you for the detailed response and clarifications. The response sufficiently addresses my questions. After carefully considering the responses, I will be keeping my current rating. I think the authors need to address the remaining concerns of other reviewers sufficiently.

---

> > > ### Author Response · Authors · 2023-08-20
> > >
> > > We warmly thank the reviewer for their response. Following your advice, we will proceed to address the remaining concerns of others reviewers to the best of our effort, hoping it will eventually be considered to be sufficient.

---

### Official Review · Reviewer_MHs8 · 2023-07-13

**Soundness:** 2 fair
**Presentation:** 2 fair
**Contribution:** 1 poor
**Rating:** 3
**Confidence:** 3

**Summary:**

The paper proposed an efficient method for solving diffusion SDEs with some convergence guarantee. Experiments are done to verify their claims.

**Strengths:**

The experiments show SEEDS-3 achieves optimal results with minimal NFE on many data sets.

**Weaknesses:**

1. Authors claim they have found a novel representation of the exact solution of the SDE, page 3, line 106. However, it is just a simple technique that's commonly used when solving SDEs. For example, when solving OU process, you multiply some exponential multiplier like this. I don't find anything novel about this.
2. Essentially, in the solution, the first and last terms are treated using a multiplier and change of variables. The second is rewritten using Taylor expansion. The convergence theorem is sort of too straightforward, as the Taylor expansion of course approximates the original function.
3. The experiments only report the NFE taken when achieving optimal performance, which I am not sure is a commonly accepted metric of evaluation in this community. Please comment on this. I am especially open to modifying the score if this question is addressed sufficiently.

**Questions:**

See weaknesses.

**Limitations:**

yes

---

> ### Author Rebuttal · Authors · 2023-08-09
>
> We thank reviewer MHs8 for their comments. We have clarified in a general rebuttal response what building principles in SEEDS are our own contributions. In particular:
> - Building principle of eq.5 will not be presented as novel in the revised version
> - The truncated Itô-Taylor expansion, the SETD method and specially the specific combination of stochastic terms that we used are novel and not incremental over [DPM-Solver,DEIS].
>     - We point out that this already suggests that our convergence theorem is not straightforward. We refer the reviewer to Appedix B.2 in our manuscript for its proof. In particular, the use of Itô's Lemma, Itô isometry, Doob's martingale inequality, Hölder's, Lyapounov's and Grönwall's inequality are necessary to prove our theorem.
>
> > The experiments only report the NFE taken when achieving optimal performance, which I am not sure is a commonly accepted metric of evaluation in this community.
>
> Actually, besides optimal FID, we also report FID/NFE curves (in Fig. 1) as evaluation metrics (with supporting FID\NFE Table 3--5 in App. E.4). For sake of completeness, and in hope to fully address the reviewers question, we report below FID\NFE tables for experiments that lead to the results in Table 1 and we will add them in our manuscript.
>
> In what follows: †=discrete models. ‡=continuous models. (+) resp. (-) means value at n+1 resp. n-1 NFE.
>
> ### CIFAR-10-uncond-vp‡
>
> (*value obtained with the non-deep vp pretrained model. The reported 2.59 FID at 51 NFE in Table 1 of our manuscript is obtained with the "VP deep" architecture)
> | Method\NFE|11|15|24|30|48|63|126|165|180|511
> |:---|:----:|:----:|:----:|:----:|:----:|:----:|:----:|:----:|:----:|:----:|
> |Euler-Maruyama|304.73(-)|248.13|\ |\ |66.32(NFE=50)|\ |\ |\ |12.27(NFE=200)|2.44(NFE=1000)
> |GottaGoFast|\ |\ |\ |\ |82.42|\ |\ |2.73(NFE=151)|2.44|\
> |EDM $(\text{S}_{\text{churn}}=0)$|18.41|6.52|3.52(-)|3.10(+)|2.99(-)|2.94|2.98(+)|\ |\ |2.93
> |EDM $(\text{S}\_{\text{tmin,tmax}}+\text{S}\_{\text{noise}}=1)$|34.31|11.26|4.59(-)|3.44(+)|2.89(-)|2.77|2.72|\ |\ |2.69
> |EDM $(\text{S}_{\text{noise}}=1)$|35.21|12.44|5.38(-)|3.99(+)|3.13(-)|2.90|2.60|\ |\ |2.55
> |EDM $(\text{S}_{\text{tmin,tmax}}=[0,\infty])$|29.63|10.01|4.51(-)|3.43(+)|2.87(-)|2.71|2.52|\ |\ |2.54
> |EDM $(\text{optimal})$|29.89|10.21|4.85(-)|3.77(+)|3.08(-)|2.84|2.47|\ |\ |**2.27**
> |DPM-Solver-3|9.04|3.76|3.00|2.95|2.90|2.89*|\ |\ |\ |\
> |SEEDS-3 $(\text{S}_{\text{churn}}=0)$|\ |\ |\ |288.22(+)|90.25(-)|33.91|2.45|**2.39**|2.45|\
>
> ### CelebA 64x64†
>
> |Method\NFE|9|12|15|21|51|60|90|102|201
> |:---|:---:|:---:|:---:|:---:|:---:|:---:|:---:| :---:|:---:
> |Euler-Maruyama|310.22(+)|227.16|207.97|120.44(-)|29.25(-)|\ |\ |\ |3.90(-)
> |Analytic-DDPM|28.99(+)|25.27|21.80|18.14(-)|11.23(-)|\ | \ | \ |6.51(-)
> |Analytic-DDIM|15.62(+)|13.90|12.29|10.45(-)|6.13(-)|\ |\  |\ |3.46(-)
> |DDIM|10.85(+)|9.99|7.78|6.64(-)|5.23(-)|\ |\ |\ |4.78(-)
> |DPM-Solver-3|6.92|4.20|3.05|2.82|**2.71**(NFE=36)
> |SEEDS-3|460.87|374.48|301.66|261.87|3.84|6.58|**1.88**|1.97|\
>
> In particular, the reported values for DPM-Solver-3 on CIFAR-10-vp-uncond (on the non-deep model) were computed by us, with sole purpose to propose a FID\NFE baseline. For baseline CIFAR-10-vp-conditional and optimized ImageNet64-EDM-preconditioned pretrained DPM, we didn't find FID/NFE curves that we could use as a baseline but we still report various FID values at increasing NFEs so that future works can use them as baselines.
>
> ### SEEDS-3 on other datasets
>
> |Dataset\NFE|15|21|30|60|90|120|129|150|165|180
> |:---|:---:|:---:|:---:|:---:|:---:|:---:|:---:|:---:|:---:|:---:
> |CIFAR-10-cond-vp‡|239.23|167.58|131.09| 25.06|3.19|2.17|2.08|2.15|2.16|2.19
>
> |Dataset\NFE|12|15|21|51|102|201|270
> |:---|:---:|:---:|:---:|:---:| :---:|:---:|:---:
> | ImageNet 64x64‡|209.12|197.79|153.72|63.75|16.35|1.56|1.38
>
> ### Discussion on evaluation metrics for DPM sampling methods
>
> All benchmarks on image generation tasks in paperswithcode.com classify papers by FID score and, when applicable, the corresponding NFE.
> While comparing solver's performance should ideally be done by comparing their FID/NFE curves at fixed datasets & pretrained DPMs, computing such curve is computationally expensive. As such, we have many FID/NFE curves on CIFAR-10, which become strong comparison baselines (one baseline per pretrained DPM on CIFAR). Usually, authors establish FID/NFE curves on small datasets to illustrate the solver's convergence behaviour and then proceed to report FID scores at specific NFE values. Some limitations:
> - [EDM] does not report FID/NFE curves for FFHQ, AFHQ and EDM-preconditioned ImageNet64, only showing the best obtained FID score for those. As such, using FID curves to compare SEEDS to EDM on those datasets is not possible unless we produce ourselves EDM curves. Additionally, [EDM] computes FID curves for CIFAR-10-VP uncond. cont. but not in the "deep" version (which has 8 instead of 4 residual blocks) and the FID points are given on different NFEs than [DPM-Solver] (which uses the "deep" architecture). Evaluating solvers is something that needs to be done with fixed pretrained DPMs and making FID/NFE tables needs to have FID values at same NFEs.
>
> We hope to have convinced the reviewer that we did our best to use the ideal FID/NFE curve+table evaluation metric whenever it was feasible, and reported the best found FID+corresponding NFE for all experiments. We also hope we addressed a discussion on evaluation metrics that will satisfy the reviewer and help promoting the acceptance of our work.
>
> References:
>
> [DEIS] Zhang & Chen. Fast Sampling of Diffusion Models with Exponential Integrator
>
> [EDM] Karras et al. Elucidating the Design Space of Diffusion-based Generative Models
>
> [DPM-Solver] Lu et al. DPM-solver: A Fast ODE Solver for Diffusion Probabilistic Model Sampling in Around 10 steps

---

### Official Review · Reviewer_CfgK · 2023-07-31

**Soundness:** 2 fair
**Presentation:** 3 good
**Contribution:** 1 poor
**Rating:** 3
**Confidence:** 3

**Summary:**

This paper proposes a sampling method that achieves 3 to 5 times faster computation by exploiting the semi-linearity in the time-reversal stochastic differential equation (SDE). It analytically computes the linear part and reaches optimal quality sampling. This sampling method demonstrates comparable results to existing SDE solvers on datasets like CIFAR10 or Celeba without requiring any additional training methods.

**Strengths:**

The paper is well-written in a clear and readable manner. It is easy to understand the paper without much difficulty, and the derivation of formulas is well-guided, focusing on the essential parts. One of the strengths of the paper is that it proposes a method that achieves approximately 3 to 5 times faster computation while still reaching optimal performance, which sets it apart from other ODE-based methods. Additionally, the paper provides a thorough and clear analysis of convergence, which is commendable.

**Weaknesses:**

In (5), the claim of a "Novel representation of exact solutions of diffusion SDEs" seems to be overclaimed as the equation is well-known in the form of the weak solution of the SDE. Therefore, it is difficult to consider this as a newly discovered novel representation. Moreover, while this integral representation leads to a practical sampling method, much of it is based on formulas used in the DPM-solver [1]. This suggests that the contribution of this work may be relatively weak. The main difference from the DPM-solver appears to be the inclusion of a stochastic term, but beyond that, significant distinctions are not apparent.

The claimed advantage of reducing the number of function evaluations (NFE) by 3 to 5 times seems less compelling when compared to ODE-based methods that can achieve sampling speeds more than 50 times faster and still attain optimal performance. For more persuasive results, it would be better if this method demonstrated similar NFE to ODE-based methods while reaching optimal performance.

Additionally, it is unclear whether this method addresses any meaningful issues beyond the problem of ODE-based methods not achieving optimal performance.

[1] DPM-Solver: A Fast ODE Solver for Diffusion Probabilistic Model Sampling in Around 10 Steps, *NeurIPS*, (2022).

**Questions:**

- Is it possible to maintain fast sampling speeds for datasets with resolutions higher than 256, such as CelebA-HQ or ImageNet?
- Apart from the ability to reach optimal performance, are there any other distinguishing features compared to the DPM-solver? For instance, does it generate a more diverse set of samples or exhibit stronger capabilities in tasks like imputation?

**Limitations:**

The paper has effectively summarized its limitations.

---

> ### Author Rebuttal · Authors · 2023-08-09
>
> We thank reviewer CfgK for their effort reading our work. We have clarified in a general rebuttal response what building principles in SEEDS are our own contributions. In particular,
> - eq.5 will not be presented as novel in the revised version
> - The change-of-variables used in the EDM case (Prop. 3.2) is new and leads to the version of SEEDS achieving same performance as the optimized EDM solver but twice faster than the latter.
> - The truncated Itô-Taylor expansion, the SETD method and specially the specific combination of stochastic terms that we used are novel and not incremental over prior works.
> - Although for small datasets, ODE methods achieve optimal performance, already for ImageNet64 it has been shown in [EDM (arXiv version v2), Fig. 5(c)] the necessity of stochasticity to obtain optimal performance. It is precisely in that scenario that SEEDS show competitive results with twice less NFEs than EDM solver.
>
> > Is it possible to maintain fast sampling speeds for datasets with resolutions higher than 256, such as CelebA-HQ or ImageNet?
>
> Indeed, SEEDS maintains fast sampling speeds on higher-resolution images. For unconditional generation using Latent Diffusion Model, SEEDS is able to generate good quality images already at 100 NFEs. In the attached PDF file to this rebuttal, we added a 512x512 image generated with SEEDS-1 at 90 NFEs on Stable Diffusion. This is consistent with the suggestion in [EDM, end of Section 5], stating that more diverse datasets continue to benefit from stochastic sampling rather than deterministic sampling. We hope this convinces the reviewer on the scalability of SEEDS.
>
> > Apart from the ability to reach optimal performance, are there any other distinguishing features compared to the DPM-solver? For instance, does it generate a more diverse set of samples or exhibit stronger capabilities in tasks like imputation?
>
> We computed the Inception Score for SEEDS and saw only marginal improvements compared to DPM-Solver so both exhibit comparable sample diversity. Nevertheless, SEEDS have at least two distinctive capabilities in the realm of adversarial robustness showing substantial capabilities compared to DPM-Solver and EDM, which we hope will be relevant enough to the reviewer for promoting the acceptance of our work.
>
> - For more than 2 years, the leaderboard of RobustBench has been dominated by Diffusion-based data augmentation techniques + Adversarial Training. The current SOTA is [DMAT, Table 4], which uses EDM-preconditioned DPMs and need to generate as many as 50M images to achieve SOTA robustness results. For ImageNet-64, [DMAT] use the EDM pretrained model+stochastic sampler for data augmentation, leading to a 5% robust accuracy improvement compared to doing so for the baseline ADM pretrained model. Since SEEDS reach same FID quality as EDM, but twice faster, we believe it will have a positive impact in this domain, making diffusion-based data augmentation schemes more affordable with limited computational capacity.
> - The work [DiffPure] uses DPM-based adversarial purification as a test-time adversarial defense. The idea is to use off-the-shelf DPMs to annihilate the adversarial content in inputs in test-time before feeding them to a pretrained classifier. Below we reproduce [DiffPure, Table 6], showing standard and robust accuracy of a WideResNet-28-10 pretrained classifier with AutoAttack $l_\infty$ ($\epsilon$ = 8/255) for CIFAR-10 whose inputs are purified by a pretrained CIFAR-10-VP-unconditional DPM using ODE versus SDE sampling schemes. One can see that using SDE solvers show substantial robustness capabilities compared to ODE solvers.
>
> | Sampling Method      | Standard Acc | Robust Acc     |
> | :---        |    :----:   |        :----: |
> | VP-ODE      | 90.79       | 39.86   |
> | VP-SDE   | 89.02        | 70.64     |
>
> References:
>
> [EDM] Karras et al. Elucidating the Design Space of Diffusion-based Generative Models
>
> [DM-AT] Wang et al. Better Diffusion Models further Improve Adversarial Training
>
> [DiffPure] Nie et al. Diffusion Models for Adversarial Purification

---

> > ### Comment · Reviewer_CfgK · 2023-08-18
> >
> > Looking at the new change of variable, in the end, it seems the model calculates the score model for one NFE twice. It's uncertain whether the NFE has decreased, but the total running time might not differ significantly. I'm not sure if this can be considered a significant contribution...

---

> > > ### Author Response · Authors · 2023-08-18
> > >
> > > We thank the reviewer for their response, to which we happily bring answers and clarifications.
> > >
> > > Both in [DPM-Solver] and in our method, the score model is calculated exactly once per NFE (this is just the definition of "Number of Function Evaluations"). Since theoretically both DPM-Solver and SEEDS compute analytically all components in the sampling method, then the total runtime is expected to increase linearly with respect to increasing NFEs in both cases, as was experimentally checked. You may also verify that there is no significant improvement done, in terms of runtime vs NFEs, between DDIM and DPM-Solver in Table 7 of Appendix E.7 in [DPM-Solver]. But DPM-Solver achieves good FID score significantly faster than DDIM, a fact that can't be seen just by looking at runtime vs NFEs. As such, one cannot evaluate the significance of a sampling method solely on analysing the runtime vs NFE behaviour. In our case, since we compute every term (both with the VP or our new EDM-optimized change of variables) analytically, then the computing cost of anything other than the score model's pass is negligible.
> > >
> > > That being said, the NFEs that are necessary to obtain state-of-the-art FID score in SEEDS shows a significant improvement compared to other SDE sampling methods. It shows to be twice faster than optimized EDM sampling method on ImageNet-64 as was reported in our experiments.
> > >
> > > We hope that this, and the involved theoretical proofs we made in this work that have no equivalent in the current literature on the subject, convinces you of our contribution.

---

> > > > ### Comment · Reviewer_CfgK · 2023-08-18
> > > >
> > > > Thank you for your quick response.
> > > > However, I find the author's reply somewhat disappointing. Are you claiming that your paper is acceptable based solely on improvements in NFE, since the DPM-solver evaluation is also done in NFE? During the rebuttal process of DPM-solver, the authors were asked to provide information about runtime, as they initially only mentioned NFE and omitted runtime. While it's true that there are many factors influencing runtime, such as architecture and various perspectives, isn't it true that the linear increase in time during the sampling process is a fact?
> > > > What is the original purpose of reducing NFE? It seems people wanted to decrease the total sampling time, and one of the indicators they used is NFE. However, isn't it somewhat illogical to argue that even though total time is similar, the reduction in NFE is acceptable?
> > > > Of course, it's reasonable to assume that performing analytic calculations based on the order in Taylor expansion would increase accuracy and thus improve performance. However, if performing multiple model calculations per NFE leads to a linear slowdown in speed per calculation, wouldn't this diminish the practical merits? Also, there are various methods to apply the change of variables, so it's possible to derive non-equivalent equations. Is this really significant and unique enough to be the main contribution? I still haven't received a satisfactory answer regarding this matter.

---

> > > > > ### Author Response · Authors · 2023-08-18
> > > > >
> > > > > Thank you for your reply. In the hope of making our answer satisfactory, let us go step by step. First, we'd like to stress that the NFEs and the number of steps in a solver are not the same. For instance, DPM-Solver-3 uses 3 NFEs per step and achieves an acceptable FID score with as few as 15 NFEs (meaning 5 steps).
> > > > >
> > > > > > Are you claiming that your paper is acceptable based solely on improvements in NFE, since the DPM-solver evaluation is also done in NFE?
> > > > >
> > > > > We claim, in our general rebuttal, 4 points that we find are worthy novelties in our work. At an experimental level: improvements are done on both FID and NFE, by prioritising FID. The question we try to answer is: "Can we obtain current SOTA FID with fewer NFEs than those that were needed by the sampling method that achieved that SOTA FID ?". Notice that this is not the same as DPM-Solver, who prioritise NFEs. Their question is, "At fixed low NFEs, can we get a SOTA FID, even if this FID will not become the absolute SOTA FID found for that model (regardless of how many NFEs were used to achieve it)?
> > > > >
> > > > > > During the rebuttal process of DPM-solver, the authors were asked to provide information about runtime, as they initially only mentioned NFE and omitted runtime. While it's true that there are many factors influencing runtime, such as architecture and various perspectives, isn't it true that the linear increase in time during the sampling process is a fact?
> > > > >
> > > > > We included the runtime vs. NFE behaviour in our rebuttal to reviewer daVP. It might seem like a fact, but derivative-based methods will need to do score model's computation and also compute their derivatives. This might break such linearity. A "sanity check" we experimentally ensured that nothing unexpected might be happening. As SEEDS is fully derivative-free, we checked such expected linearity.
> > > > >
> > > > > >  What is the original purpose of reducing NFE?
> > > > >
> > > > > Talking about NFEs arose when solvers using multiple NFEs per step started to appear in the DPM literature. If one only thinks about the number of sampling steps, then one would find that DPM-Solver-3 only needs 5 steps to achieve average sampling quality. As a first thought, one would think that this means that they need to make 5 score model's inference to do so, but it actually uses 15 score model's inferences. As such, reducing the NFEs to reach a targeted FID (either optimal FID like SEEDS, or average FID like DPM-Solver) is a more pertinent evaluation metric for speeding up sampling than the number of steps.
> > > > >
> > > > > > However, isn't it somewhat illogical to argue that even though total time is similar, the reduction in NFE is acceptable?
> > > > >
> > > > > Notice that when comparing runtime and NFEs, what really matters is the behaviour of the increase of runtime when increasing NFEs. Experimenting on it somehow is a sanity check that your sampling algorithm is doing what we expect from it.
> > > > >
> > > > > > However, if performing multiple model calculations per NFE leads to a linear slowdown in speed per calculation, wouldn't this diminish the practical merits?
> > > > >
> > > > > Thank you for this question that allows us to make a good example to clarify the roles of steps and NFEs. On the contrary, multiple model calculations (which are called NFEs) per sampling step (and not per NFE) actually have shown to increase the practical merits. A method that uses 1 NFE per step (like DDIM) but needs, say, 50 steps (so 50 NFEs) to get average FID, will be much slower than a method (like DPM-Solver) that uses 3 NFEs per step but only needs 5 steps (so only 15 NFEs) to get that same average FID.
> > > > >
> > > > > > Also, there are various methods to apply the change of variables, so it's possible to derive non-equivalent equations. Is this really significant and unique enough to be the main contribution?
> > > > >
> > > > > As we responded in the general rebuttal response that you may find at the beginning of this webpage, we detailed what we claim are our principal contributions. Besides our EDM change of variables, we cited as main novelties
> > > > > - the use of the Stochastic Exponential Time-Differencing method (not incremental over DPM-Solver),
> > > > > - the use of a specially chosen Itô-Taylor expansion (that is very different from its deterministic Taylor counterpart)
> > > > > - and mainly the Markov-chain preserving choice of multiple analytic stochastic noise contributions for each higher-order correction in each step. It is, in particular, this last point that we believe is the most novel as it has no stochastic counterpart to our knowledge. Notice that DPM-Solver is heavily inspired by tables 5.3, 5.4, and 5.8 in [HO] to obtain their solvers. Our choice of stochastic components has, to our knowledge, no precedent in the Numerical Analysis literature.
> > > > >
> > > > > Taken as a whole and backed by experiments, we consider this set to be unique and significant, and we are committed to keep the dialog open, hoping it would be deemed satisfactory enough for your endorsement.
> > > > >
> > > > > [HO] Hochbruck, Ostermann. Explicit Exponential Runge-Kutta Methods for Semilinear Parabolic Problems

---

> > > > > > ### Comment · Reviewer_CfgK · 2023-08-18
> > > > > >
> > > > > > Thank you for your response. However, ultimately, I decided to lower the rating.
> > > > > >
> > > > > > Firstly, the equation (4) introduced for the stochastic Exponential Time-differencing method is not new, but a well-known formula. Secondly, the Itô-Taylor expansion introduced to derive the sampling is fundamentally not different from the deterministic Taylor expansion. This is because the noise term for equation (4) can be directly calculated using the Itô isometry, and practically, "the deterministic Taylor expansion" is used for the drift term. The technique used to expand (6) is already employed in the DPM-solver. It's hard to consider obtaining a different formula from the DPM-solver by setting a different change of variables as a significant distinction. Moreover, it's also challenging to perceive significant advantages from an order-of-magnitude perspective.
> > > > > >
> > > > > > The claimed advantage, apart from reducing NFE, is the improved robustness, but the specific benefits in terms of generation aspects and why approximating with the addition of stochastic terms are better are not explained. The contribution aspect appears to be lacking, and I still feel that I haven't received a satisfying answer to my question.

---

> > > > > > > ### Author Response · Authors · 2023-08-19
> > > > > > >
> > > > > > > We would like to point out to the reviewer that:
> > > > > > > - SETD is not the same as the exponential representation of the exact solution (eq.4). We already clarified eq. 4, and the SETD method means expressing a stochastic integral in terms of $\varphi$-functions (and you need exponentially weighted stochastic integrals to be able to do so here).
> > > > > > > - The explicit form of the Itô-Taylor expansion (see App. D.2) to degree 2 is needed to show that SEEDS-1 is not of strong order 0.5 but 1.0: SEEDS cannot be theoretically grounded by a purely deterministic Taylor-like argument.
> > > > > > > - One can put a highly precise sampling method to a badly trained DPM that the generation process will still be bad. This is because the sampling error and the learning approximation error are independent one to the other.
> > > > > > > - Itô Isometry is not sufficient to successfully determine the stochastic contributions in SEEDS. In addition to the ablation experiment done in Fig. 1 of the appended PDF, the theoretical explanation of our choice of the stochastic components is contained in Appendix B.2.4 and B.3 in an implicit for. We have singled it out below to make a clear explanation of it and will be added to our manuscript to the request of the reviewer. This also clarifies how changing the choice in the stochastic terms leads inevitably to failing to satisfy independence of brownian increments on non overlapping intervals and generalises to SEEDS-3. Hopefully this will convince you that our approach is theoretically grounded.
> > > > > > >
> > > > > > > **The sequence $(\tilde{\mathbf{𝐱}}_t )_t$ induced by the choice of stochastic noise contributions in SEEDS-2 presented in Alg. 3 satisfies the Markov property.**
> > > > > > >
> > > > > > > **Proof**:
> > > > > > >
> > > > > > > On the one hand, we have
> > > > > > > $$\mathbf{u}_1=\frac{\alpha\_{s_1}}{\alpha_s} \tilde{\mathbf{x}}_s-2\sigma\_{s_1} \left( e^{\frac{h}{2}} - 1 \right) \mathbf{\epsilon}\_{\theta} (\tilde{\mathbf{x}}_s, s) - B^1, \qquad B^1:=\sqrt{2}\alpha\_{s_1} \int\_{\lambda\_s}\^{\lambda\_{s_1}} e^{- \lambda} d  \omega\_{\lambda}.$$
> > > > > > >
> > > > > > > Now $B^1$ depends on the Brownian movement $\left( \omega\_{\lambda\_{s_1}}-\omega\_{\lambda\_s} \right)\_{\lambda\_{s_1} \geq \lambda_s}$. By the Markov property, this is independent of $(\omega\_{\lambda_u})\_{\lambda_u \leq \lambda_s}$.
> > > > > > >
> > > > > > > Since $\tilde{\mathbf{x}}\_s$ is a function of $(\omega\_{u} )\_{u \geq s}$, we deduce that $B^1$
> > > > > > > has to be independent of all $( \tilde{\mathbf{x}}\_u )\_{u \geq s}$. By Itô Isometry, we obtain (noticing that $h\varphi(h)=e^h - 1$ to make clear the use of the SETD method in our approach)
> > > > > > > $$B^1 = \sigma\_{s_1} \sqrt{e^h - 1} z^1, \quad z^1 \sim \mathcal{N} (\mathbf{0}, \mathbf{I}\_d).$$
> > > > > > >
> > > > > > > On the other hand, the update $\tilde{\mathbf{x}}\_t$ is
> > > > > > > $$\tilde{\mathbf{x}}\_t = \frac{\alpha_t}{\alpha_s} \tilde{\mathbf{x}}\_s - 2 \sigma_t (e^h - 1) \mathbf{\epsilon}\_{\theta} (\tilde{\mathbf{x}}\_{s_1}, s_1) - B^0, \qquad B^0 = \sqrt{2} \alpha_t \int\_{\lambda\_s}\^{\lambda\_t} e^{- \lambda} d \omega\_{\lambda}. $$
> > > > > > > Then, by using independent increments property of Wiener process $ \{ \omega_{\lambda} : \lambda \in [\lambda_s,
> > > > > > > \lambda_t] \}$, we can
> > > > > > > deduce, for $0 \leq \lambda_s < \lambda_{s_1} < \lambda_t$, that
> > > > > > > $$ \omega\_{\lambda\_{s_1}} - \omega\_{\lambda_s} \quad \text{and} \quad
> > > > > > >    \omega\_{\lambda_t} - \omega\_{\lambda\_{s_1}}$$
> > > > > > >  are independent. Then, by the above Brownian independence property, the random variables $B^0$
> > > > > > > and $B^1$ are
> > > > > > > - Independent on non-overlapping time intervals
> > > > > > > - Dependent on overlapping intervals.
> > > > > > >
> > > > > > > By Chasles rule, we then decompose
> > > > > > >
> > > > > > > $$ B^0 = \sqrt{2} \alpha_t \int\_{\lambda\_s}\^{\lambda\_{s_1}} e^{- \lambda}
> > > > > > >    d \omega\_{\lambda} + \sqrt{2} \alpha_t
> > > > > > >    \int\_{\lambda\_{s_1}}\^{\lambda\_t} e^{- \lambda} d \omega\_{\lambda} $$
> > > > > > > and, as
> > > > > > > $$\int\_{\lambda\_s}^{\lambda\_{s_1}} e^{- \lambda} d \omega\_{\lambda} =
> > > > > > >    \frac{B^1}{\sqrt{2} \alpha\_{s_1}}, $$
> > > > > > > we obtain
> > > > > > > $$B^0 = \sqrt{2} \alpha_t \left[ \frac{B^1}{\sqrt{2} \alpha\_{s_1}} +
> > > > > > >    \int\_{\lambda\_{s_1}}\^{\lambda\_t} e^{- \lambda} d \omega\_{\lambda}
> > > > > > >    \right] . $$
> > > > > > >
> > > > > > > Then we have
> > > > > > > $$B=\sqrt{2} \alpha_t \int\_{\lambda\_s}\^{\lambda\_{s_1}} e^{- \lambda}
> > > > > > >   d \omega\_{\lambda} + \sqrt{2} \alpha_t \int\_{\lambda\_s}\^{\lambda\_t}
> > > > > > >   e^{- \lambda} d \omega\_{\lambda}=  \sqrt{2} \alpha_t \frac{1}{\sqrt{2}} \frac{\sigma\_{s_1}}{\alpha\_{s_1}} \sqrt{e\^{2
> > > > > > >   (\lambda\_{s_1} - \lambda\_s)} - 1} z^1 + \sqrt{2} \alpha_t \frac{1}{\sqrt{2}}
> > > > > > >   \frac{\sigma_t}{\alpha_t} \sqrt{e^{2 (\lambda_t - \lambda_s)} - 1} z^2 = \frac{ \sigma\_t}{\alpha_t} \left( \frac{\sigma\_{s_1} \alpha_t}{\sigma_t \alpha\_{s_1}} \sqrt{e^h - 1} z^1 + \sqrt{e^{2 h} - 1} z^2 \right) =  \sigma_t \left( e\^{\lambda_t - \lambda\_{s_1}} \sqrt{e^h - 1}
> > > > > > >   z^1 + \sqrt{e^{2 h} - 1} z^2 \right) = \sigma\_t \left( e^{\frac{h}{2}} \sqrt{e^h - 1} z^1 + \sqrt{e^{2
> > > > > > >   h} - 1} z^2 \right) = \sigma\_t \left( e^{\frac{h}{2}} \sqrt{h \varphi_1(h) } z^1 + \sqrt{2h  \varphi_1(2h)} z^2 \right)
> > > > > > > =  \sigma\_t \left( \sqrt{e^{2 h} - e^h} z^1 + \sqrt{e^{2 h} - 1}z^2 \right)
> > > > > > > $$
> > > > > > > which completes the proof.

---

> > > > > > > > ### Author Response · Authors · 2023-08-21
> > > > > > > >
> > > > > > > > Hoping to fully address the reviewer's concerns satisfyingly, we would like to make some further points that might have been missed in our initial responses.
> > > > > > > >
> > > > > > > > > The reviewer asked about the extent to which our work potentially addresses distinguishing features compared to DPM-Solver beyond optimal performance, like sample diversity or exhibiting stronger capabilities in tasks like imputation.
> > > > > > > >
> > > > > > > > We proposed the distinguishing features of SDE solvers for robustness tasks in comparison to ODE solvers. As a great number of DPM-based papers have been widely adopted in the field of robustness, which is very significant in ML, we hope this to be a first acceptable answer.
> > > > > > > >
> > > > > > > > > As we understand, the reviewer's concern is centred on the significance of our contribution and, in particular, the extend to which it differs from [DPM-Solver]. The reviewer pointed out the similarities between these two works: the use of the variation-of-parameters formula and the use of a change of variables to obtain exponentially weighted integrals, the similarity between the Taylor expansion of [DPM-Solver] and our truncated Itô-Taylor expansion.
> > > > > > > >
> > > > > > > > In l.42 of our introduction we stated that one of the objectives of SEEDS is to enhance and generalise [DPM-Solver] to the stochastic case in various frameworks (like the VE, iDDPM and EDM frameworks in Tab.1 in [EDM]) so these similarities are to be expected. The contribution we would like the reviewer to focus on is Theorem 4.1. The involved proof of this result shows the extent to which generalising DPM-Solver to the stochastic case is not trivial precisely from the point of view of **showing strong order convergence guarantees**.
> > > > > > > > - Heuristically **constructing SEEDS** has, as distinctive novelty, the principled choice of noise components whose theoretical justification was clarified during this rebuttal.
> > > > > > > > - The EDM change of variables allowed our main **experimental feature**: to match the EDM solver's performance on ImageNet64 twice faster, and without needing any optimisation procedure to do so.
> > > > > > > > - Formally, we believe that **proving that SEEDS are SDE solvers with convergence guarantees** should be seen as a distinctive and significant contribution. Up to our knowledge, **none of the proposed SDE solvers for DPMs have provided such guarantees**. Actually [EDM, Sec. 4 & Tab. 5] states that, in their sampling method with injected stochasticity, their optimal choices for $S_{churn},... , S_{noise}$ all break convergence guarantees (due to the choice $S_{noise} > 1$). So it cannot be seen as a proper (well-conditioned) SDE solver. Convergence guarantees are what rigorously justify that a sampling scheme is reliable. For instance, it is shown in [R] that the convergence order for DPM-Solver (in Data Prediction mode) has been over-estimated (i.e. **suffer from order reduction** [H, Sec. 5]) and propose a new exponential integrator based on [H, Tab. 5.9] to enforce correct convergence order degree, consequently showing generation quality improvements. In contrast, our proven convergence orders are optimal, not overestimated, and need the full Itô-Taylor expansion expression to be proven.
> > > > > > > >
> > > > > > > > We hope this may satisfy the reviewer's concern on our contributions.
> > > > > > > >
> > > > > > > > > The specific benefits in terms of generation aspects and why approximating with the addition of stochastic terms are better are not explained
> > > > > > > >
> > > > > > > > We provided explanation of the stochastic terms. Benefits of SEEDS, as a SDE solver, in terms of generation aspects are:
> > > > > > > > - As proven in [L, App. B], even if the score model has been trained to the optimal score function, the distributions between SDEs and ODEs are **still different**. This is because the distribution at time $T$ in the forward process is always not exactly a standard normal distribution as in the reverse SDE/ODE process, and thus the distributions at time 0 are different: **SDE solvers and ODE solvers may have different sample quality, even if the score models are optimal**.
> > > > > > > > - [X, Th.1] shows how, in the large NFE regime, SDE methods (with proven convergence orders) **will outperform ODE methods** in terms of sampling quality.
> > > > > > > > - Both [X, C] show that the sampling error divides into **discretisation** and **approximation** errors, indicating that one can hardly expect from any sampling method to enable DPMs to exhibit significant improvements on generative tasks that the pretrained score model didn't already possess.
> > > > > > > >
> > > > > > > > We kindly ask all reviewers to take this answers into account.
> > > > > > > >
> > > > > > > > [R] Zhang et al. Improved Order Analysis and Design of Exponential Integrator for Diffusion Models Sampling
> > > > > > > >
> > > > > > > > [H] Hochbruck, Ostermann. Explicit Exponential Runge-Kutta Methods for Semilinear Parabolic Problems
> > > > > > > >
> > > > > > > > [L] Lu et al. Maximum likelihood training for score-based diffusion ODEs by high order denoising score matching
> > > > > > > >
> > > > > > > > [X] Xu et al. Restart Sampling for Improving Generative Processes
> > > > > > > >
> > > > > > > > [C] Chen et al. Sampling is as easy as learning the score: theory for diffusion models with minimal data assumptions

---

### Official Review · Reviewer_nfDR · 2023-08-01

**Soundness:** 3 good
**Presentation:** 3 good
**Contribution:** 3 good
**Rating:** 6
**Confidence:** 4

**Summary:**

This paper aims to accelerate the SDE solvers through exponential integrators. The authors compute the linear part of the SDE analytically, employ a more accurate interaction of the stochastic part. Experimentally, the proposed method improves over the previous ODE solvers.

**Strengths:**

- The paper proposes to utilize the exponential integrator to solve the diffusion SDEs (Eq. (5)). In addition, it approximates the integral components through Taylor expansion.

- The paper utilizes the stochastic exponential time differencing method to compute the variance of the stochastic term analytically.

- Experimentally, the third-order solver yields better FID than previous ODE solvers while requiring significantly smaller NFEs than SDE solvers.

**Weaknesses:**

- It seems that the same formula of the exponential integrated version of SDE (Eq.5) has been proposed in prior works [1] (see their eq.17, arXiv version 1). Since the DPM-solver [2] is essentially the same as DEIS, and they also use a Talyer expansion of the score function, I wonder how this work differs from these prior works.

- The authors show in Prop 4.4 that setting $g=0$ in Eq.15, the resulting SEEDs solvers do not equal to DPM-solver. I wonder why this is the case based on the point above. In addition, setting $g=0$ in Eq.15 would not lead to meaningful backward ODE/SDEs.

- The SEEDS-3 solver indeed showcases great improvement over ODE solvers. However, when using an increased NFE, the EDM solvers perform better than SEEDS-3 (SEEDS-3 even gets worse when increasing NFE as shown in Fig.1). In addition, I notice the SEEDS solver does not even apply to a low NFE regime (NFE<100) on CIFAR-10. It might be helpful to utilize the techniques in a concurrent work [3] to further improve the sample quality when increasing the NFE, or to achieve better results in the low NFE regime. It also is helpful to discuss [3] as they also focus on improving the existing SDEs.

### Minors

- Line 74 : $\mu_t \to \alpha_t$



[1] Zhang, Qinsheng, and Yongxin Chen. "Fast sampling of diffusion models with exponential integrator." arXiv preprint arXiv:2204.13902 (2022).

[2] Lu, Cheng, et al. "Dpm-solver: A fast ode solver for diffusion probabilistic model sampling in around 10 steps." Advances in Neural Information Processing Systems 35 (2022): 5775-5787

[3] Xu, Yilun, et al. "Restart Sampling for Improving Generative Processes." arXiv preprint arXiv:2306.14878 (2023).

**Questions:**

- Is this work a straightforward application of prior works ([1], [2]) to SDEs?

- In Figure 1, why does the SEEDS solver have a such high FID score (worse sample quality) in the low NFE regime? Is it due to the error when simulating the noise term?

**Limitations:**

Discussed in Sec 6.

---

> ### Author Rebuttal · Authors · 2023-08-09
>
> We thank reviewer nfDR for their effort reading our work.
>
> > Is this work a straightforward application of prior works ([1], [2]) to SDEs?
>
> We have clarified in a general rebuttal response what building principles in SEEDS are our own contributions. In particular:
> - Building principle of eq.5 will not be presented as novel in the revised version
> - The change-of-variables used in the EDM case (Prop. 3.2) is new and leads to the version of SEEDS achieving same performance as optimized EDM solver but twice faster than the latter.
>     - We point out that Fig.1(c) is not a FID/NFE curve.
>     - We refer to Fig.1 of the attached PDF file to this rebuttal to illustrate that SEEDS stabilizes at increasing NFEs.
> - The truncated Itô-Taylor expansion, the SETD method and specially the specific combination of stochastic terms that we used are novel and not incremental over [DPM-Solver,DEIS].
> - Although for small datasets, ODE methods have demonstrated to achieve optimal performance, already for ImageNet64 it has been shown in [Karras, Fig. 5(c)] the necessity of stochasticity to obtain optimal performance. It is precisely in that scenario that SEEDS show competitive results with twice less NFEs that EDM solver.
> - Setting $g=0$ in Eq. 15 yields a both a valid ODE and valid ODE solvers. But such ODE is the PFO of a forward SDE process which is not equivalent to the one we used (the induced marginal probability trajectories via Fokker-Planck do not follow the same PDE).
>
> We hope this will convince you that our work is not a straightforward application of the mentioned prior works.
>
> > In Figure 1, why does the SEEDS solver have a such high FID score (worse sample quality) in the low NFE regime? Is it due to the error when simulating the noise term?
>
> We thank the reviewer for pointing out the work [XY] that appeared after our submission (so cannot be considered as a baseline for this submission). Interestingly, in [XY, Th.1] there is a formal explanation on why ODE samplers outperform SDE samplers in the small NFE regime and fall short in the large NFE regime. In particular, it is theoretically shown that, at large step sizes, it is the discretization error that dominates sampling errors while, at small step sizes, it is the approximation error that dominates it. We will include this discussion in our manuscript, hoping that this will encourage the reviewer into promoting the acceptance of our work, and we definitely look forward to combine SEEDS with the new ideas in [XY].
>
> References:
>
> [DPM-Solver] Lu et al. DPM-solver: A Fast ODE Solver for Diffusion Probabilistic Model Sampling in Around 10 steps
>
> [DEIS] Zhang & Chen. Fast Sampling of Diffusion Models with Exponential Integrator
>
> [EDM] Karras et al. Elucidating the Design Space of Diffusion-based Generative Models
>
> [XY] Xu et al. Restart Sampling for Improving Generative Processes

---

### Author Rebuttal · Authors · 2023-08-09

## General Rebuttal Response
As reviewers 1-4 pointed out, and with whom we fully agree, the widely known variation-of-parameters formula is not a contribution of ours and we will modify this in our manuscript. In our effort to clarify the building principles of SEEDS, we unintentionally stressed incorrectly which of those principles are novelties in our work.

Let's clarify the 4 contributions :
- Using the stochastic exponential time differencing (SETD) method (l.130) to express the noise variance in terms of the $\varphi$ functions for the exponentially weighted stochastic integrals is new and not incremental over [DPM-solver]
- The truncated Itô-Taylor expansion is a new tool. It is different from - and not incremental over - Taylor expansions (as highlighted by reviewer daVP and fully detailed in App. D.2.2)
- Our principled use of the Chasles rule (l.182) to enforce dependence on overlapping paths for SEEDS-2/3 is new and is the central key of success of SEEDS. It also ensures that the set of resulting iterations of our solvers satisfy the Markov property. Furthermore, we claim that the specially chosen combination of stochastic terms in SEEDS is not incremental over any prior work. To highlight this, we added an ablation study on how choosing different noise decompositions (e.g. the values of A and B in Algorithm 4) has an impact on the performance of SEEDS. We consider 4 different combinations of noise components in SEEDS-2/3. For simplicity let's explain it at the SEEDS-2 (Alg. 3) level.
    - Set $(z^1,z^2)$ two independent standard Gaussian random variables. Denote $A=\sigma_{s_1} \sqrt{e^h - 1} z^1$ for the noise contribution in the $\mathbf{u}$ term. We have the following choices for the noise contribution for $\tilde{\mathbf{x}}_t$:
        - SEEDS-2: our noise combination $B=\sigma_t \left( \sqrt{e^{2 h} - e^h} z^1 + \sqrt{e^h - 1} z^2 \right)$
        - Naive v1: one noise per stage $B=\sigma_t \left( \sqrt{e^{2 h} - e^h} + \sqrt{e^h - 1} \right) z^2$
        - Naive v2: one noise per step $B=\sigma_t \left( \sqrt{e^{2 h} - e^h} + \sqrt{e^h - 1} \right) z^1$
        - Naive v3: one noise per integral evaluation $B=\sigma_t \left( \sqrt{e^{2 h} - e^h} z^3 + \sqrt{e^h - 1} z^2 \right)$ (where $z^3$ is also standard Gaussian)
        - Naive v4: noises in inverse position $B=\sigma_t \left( \sqrt{e^{2 h} - e^h} z^2 + \sqrt{e^h - 1} z^1 \right)$

     For each of these combinations, we generated FID/NFE curves (see Fig. 1 in the attached PDF file), showing that using all naive combinations of noises in SEEDS-2/3 lead to a sharp drop of performance both in quality and speed. Finally, our choice is enforced with a theoretical justification of it (using the independence property of stochastic integrals on disjoint intervals) that has no counterpart in the deterministic setting. All convergence proofs in Appendix B make intense use of formulae (like Itô Isometry or Doob's martingale inequality) that also have no deterministic equivalent.
- The change-of-variables optimized for the EDM framework is new (our Prop. 3.2) and is the one we use to achieve same performance as EDM solver but with twice less NFEs than the latter for ImageNet-64 (Fig. 1(c) in our manuscript).
To give an example of the difference in the obtained formulas for SEEDS in this case, we write below the SEEDS-2 iteration rule, and we can add it to the manuscript at the request of the reviewers:
    - $\lambda (t) := - \log \left( \frac{t}{\sigma_d \sqrt{t^2 + \sigma^2_d}}
\right) \qquad t_{\lambda} (\lambda) :=
\frac{\sigma_d}{\sqrt{\frac{1}{\sigma^2_d e^{- 2 \lambda}} - 1}}, \quad
\sigma_d = \sigma_{data}$
    - $h = \lambda_t - \lambda_s, \quad \lambda_{s_1} = \lambda_s +
rh , \quad s_1 = t_{\lambda} (\lambda_{s_1})$
    - $\epsilon_{\theta} \left( \widetilde{\mathbf{x}}_s, s \right)$ = $\left[ D\_{\theta}
\left( \frac{\widetilde{\mathbf{x}}_s}{\alpha_s} ; \sigma_s \right) -
\frac{\sigma_d^2}{s^2 + \sigma_d^2} \frac{\widetilde{\mathbf{x}}_s}{\alpha_s} \right]
\cdot \frac{\sqrt{s^2 + \sigma_d^2}}{s \sigma_d},$
    - $\widetilde{\mathbf{x}}_{s_1} = \frac{s_1^2 + \sigma_d^2}{s^2 + \sigma_d^2}
\widetilde{\mathbf{x}}_s + 2 \frac{s_1  \sqrt{s_1^2 + \sigma_d^2}}{\sigma_d} (e^{r h} - 1){\epsilon\_{\theta}}\left( \widetilde{\mathbf{x}}_s, s \right) + \sqrt{\frac{(s_1^2 + \sigma_d^2) (s^2 - s_1^2)}{(s^2 + \sigma_d^2)}} z^1$
    - $\widetilde{\mathbf{x}}_t = \frac{t^2 + \sigma_d^2}{s^2 + \sigma_d^2} \widetilde{\mathbf{x}}_s + 2 \frac{t  \sqrt{t^2 + \sigma_d^2}}{\sigma_d} (e^{ h} - 1) \epsilon\_{\theta}\left({\widetilde{\mathbf{x}}\_{s_1}}, s_1 \right) + \sqrt{\frac{(t^2 + \sigma_d^2) (s_1^2 - t^2)}{(s_1^2 + \sigma_d^2)}} (e^{r h} z^1 + z^2) $

We hope that this clarification both highlights: on the one hand, the building principles of SEEDS, and on the other hand, the original contributions of our work. Finally, we would like to emphasise that many available optimisation tools for sampling (for instance multi-stepping, dynamic thresholding,...) can be readily built on top of SEEDS to better address specific tasks like guided sampling.

References:

[DPM-Solver] Lu et al. DPM-solver: A Fast ODE Solver for Diffusion Probabilistic Model Sampling in Around 10 steps

---

### Comment · Area_Chair_MDun · 2023-08-18
**Respond to author rebuttals**

Dear Reviewers nfDR, MHs8, daVP, YvZd

Please respond to author rebuttals as soon as possible for timely communicating with authors until next Monday.

Thanks,
AC.

---

### Decision · Program_Chairs · 2023-09-21

**Decision:**

Accept (poster)

**Comment:**

This paper proposes a fast SDE solver while maintaining the optimal performance of the SDE in sampling from pre-trained diffusion probabilistic models. Especially, the proposed three SDE solvers (SEEDS) are derivative and training free with proven convergence guarantees and empirically demonstrates the optimal performances in terms of quality with small NFEs compared to the previous SDE solvers on image generation benchmarks.
Overall, the main contribution of this paper seems to the proposed analytic computation of the “stochastic” component and the corresponding “convergence guarantees”, which is different from the existing fast ODE solvers such as DPM-solver. I think this contribution is not marginal and the feasibility and the practicality have been sufficiently validated from experimental results. Therefore, I recommend this paper to be accepted.
Having said that, some reviewers claim that the derivation of the proposed solution for the stochastic term in the SDE seems to be based on common and general approaches and the resulting guaranteed convergence would be somewhat straightforward. Hence, the authors need to thoroughly and clearly address these points (more than the responses in the rebuttal) in the revision.